# APPROXIMATING NASH EQUILIBRIA IN NORMAL-FORM GAMES VIA STOCHASTIC OPTIMIZATION

**Ian Gemp**
DeepMind
London, UK
imgemp@google.com

**Luke Marris**
DeepMind
London, UK
marris@google.com

**Georgios Piliouras**
DeepMind
London, UK
gpil@google.com

## ABSTRACT

We propose the first loss function for approximate Nash equilibria of normal-form games that is amenable to unbiased Monte Carlo estimation. This construction allows us to deploy standard non-convex stochastic optimization techniques for approximating Nash equilibria, resulting in novel algorithms with provable guarantees. We complement our theoretical analysis with experiments demonstrating that stochastic gradient descent can outperform previous state-of-the-art approaches.

## 1 INTRODUCTION

Nash equilibrium (NE) famously encodes stable behavioral outcomes in multi-agent systems and is arguably the most influential solution concept in game theory. Formally speaking, if $n$ players independently choose $n$, possibly mixed, strategies ($x_k$ for $k \in [n]$) and their joint strategy ($\boldsymbol{x} = \prod_k x_k$) constitutes a *Nash equilibrium*, then no player has any incentive to unilaterally deviate from their strategy. This concept has sparked extensive research in various fields, ranging from economics (Milgrom and Weber, 1982) to machine learning (Goodfellow et al., 2014), and has even inspired behavioral theory generalizations such as quantal response equilibria (QREs) which allow for more realistic models of boundedly rational agents (McKelvey and Palfrey, 1995).

Unfortunately, when considering Nash equilibria beyond the 2-player, zero-sum scenario, two significant challenges arise. First, it becomes unclear how $n$ independent players would collectively identify a Nash equilibrium when multiple equilibria are possible, giving rise to the *equilibrium selection* problem (Harsanyi et al., 1988). Secondly, even approximating a single Nash equilibrium is known to be computationally intractable and specifically PPAD-complete (Daskalakis et al., 2009). Combining both problems together, e.g., testing for the existence of equilibria with welfare greater than some fixed threshold is NP-hard, and it is in fact even hard to approximate (Austrin et al., 2011).

From a machine learning practitioner's perspective, such computational complexity results hardly give pause for thought as collectively we have become all too familiar with the unreasonable effectiveness of heuristics in circumventing such obstacles. Famously, non-convex optimization is NP-hard, even if the goal is to compute a local minimizer (Murty and Kabadi, 1985), however, stochastic gradient descent (SGD) and variants succeed in training billion parameter models (Brown et al., 2020).

Unfortunately, computational techniques for Nash equilibrium have so far not achieved anywhere near the same level of success. In contrast, most modern NE solvers for $n$-player, $m$-action, general-sum, normal-form games (NFGs) are practically restricted to a handful of players and/or actions per player except in special cases, e.g., symmetric (Wiedenbeck and Brinkman, 2023) or mean-field games (Pérolat et al., 2022). For example, when running the suite of all 7 applicable methods from the hallmark `gambit` library (McKelvey et al., 2016) on a 4-player Blotto game, we find only brute-force pure-NE enumeration is able to return any NE within a 1 hour time limit. Scaling solvers to large games is difficult partially due to the fact that an NFG is represented by a tensor with an exponential $nm^n$ entries; even *reading* this description into memory can be computationally prohibitive. More to the point, any computational technique that presumes *exact* computation of the *expectation* of this tensor sampled according to $\boldsymbol{x}$ similarly does not have any hope of scaling beyond small instances.

This inefficiency arguably lies at the core of the differential success between machine learning (ML) optimization and equilibrium computation. For example, numerous techniques exist that reduce the

problem of Nash computation to the minimization of the expectation of a random variable (Section 3). Unfortunately, unlike the source of randomness in ML applications where batch learning suffices to produce unbiased estimators, these techniques do not extend easily to game theory which incorporates non-linear functions such as maximum and best-response. This raises our motivating goal:

**Can we solve for Nash equilibria via unbiased stochastic optimization?**

**Our results.** Following in the successful steps of the interplay between ML and stochastic optimization, we reformulate the approximation of Nash equilibria in a normal-form game as a stochastic non-convex optimization problem admitting unbiased Monte-Carlo estimation. This enables the use of powerful solvers and advances in parallel computing to efficiently enumerate Nash equilibria for $n$-player, general-sum games. Furthermore, this re-casting allows practitioners to incorporate other desirable objectives into the problem such as "find an approximate Nash equilibrium with welfare above $\omega$" or "find an approximate Nash equilibrium nearest the current observed joint strategy" resolving the equilibrium selection problem in an effectively ad-hoc and application tailored manner. Concretely, we make the following contributions by producing:

- A loss $\mathcal{L}^\tau(\boldsymbol{x})$ 1) whose global minima well approximate Nash equilibria in normal form games, 2) admits unbiased Monte-Carlo estimation, and 3) is Lipschitz and bounded.

- Efficient randomized algorithms for approximating Nash equilibria in a novel class of games. The algorithms emerge by employing the family of $\mathcal{X}$-armed bandit approaches to $\mathcal{L}^\tau(\boldsymbol{x})$ and connecting their global stochastic optimization guarantees to global approximate Nash guarantees.

- An empirical comparison of SGD against state-of-the-art baselines for approximating NEs in large games. In some games, vanilla SGD actually improves upon previous state-of-the-art; in others, SGD is slowed by saddle points, a familiar challenge in deep learning (Dauphin et al., 2014).

Overall, this perspective showcases a promising new route to approximating equilibria at scale in practice. We conclude the paper with discussion for future work.

## 2 PRELIMINARIES

In an $n$-player, normal-form game, each player $k \in \{1, \ldots, n\} = [n]$ has a strategy set $\mathcal{A}_k = \{a_{k1}, \ldots, a_{km_k}\}$ consisting of $m_k$ pure strategies. These strategies can be naturally indexed, so we redefine $\mathcal{A}_k = [m_k]$ as an abuse of notation. Each player $k$ also has a utility function, $u_k : \mathcal{A} = \prod_k \mathcal{A}_k \to [0, 1]$, (equiv. "payoff tensor") that maps joint actions to payoffs in the unit-interval. We denote the average cardinality of the players' action sets by $\bar{m} = \frac{1}{n} \sum_k m_k$ and maximum by $m^* = \max_k m_k$. Player $k$ may play a mixed strategy by sampling from a distribution over their pure strategies. Let player $k$'s mixed strategy be represented by a vector $x_k \in \Delta^{m_k-1}$ where $\Delta^{m_k-1}$ is the $(m_k - 1)$-dimensional probability simplex embedded in $\mathbb{R}^{m_k}$. Each function $u_k$ is then extended to this domain so that $u_k(\boldsymbol{x}) = \sum_{\boldsymbol{a} \in \mathcal{A}} u_k(\boldsymbol{a}) \prod_j x_{ja_j}$ where $\boldsymbol{x} = (x_1, \ldots, x_n)$ and $a_j \in \mathcal{A}_j$ denotes player $j$'s component of the joint action $\boldsymbol{a} \in \mathcal{A}$. For convenience, let $x_{-k}$ denote all components of $\boldsymbol{x}$ belonging to players other than player $k$.

The joint strategy $\boldsymbol{x} \in \prod_k \Delta^{m_k-1}$ is a Nash equilibrium if and only if, for all $k \in [n]$, $u_k(z_k, x_{-k}) \leq u_k(\boldsymbol{x})$ for all $z_k \in \Delta^{m_k-1}$, i.e., no player has any incentive to unilaterally deviate from $\boldsymbol{x}$. Nash is typically relaxed with $\epsilon$-Nash, our focus: $u_k(z_k, x_{-k}) \leq u_k(\boldsymbol{x}) + \epsilon$ for all $z_k \in \Delta^{m_k-1}$.

As an abuse of notation, let the atomic action $a_k = e_k$ also denote the $m_k$-dimensional "one-hot" vector with all zeros aside from a 1 at index $a_k$; its use should be clear from the context. We also introduce $\nabla_{x_k}^k$ as player $k$'s utility gradient. And for convenience, denote by $H_{kl}^k = \mathbb{E}_{x_{-kl}}[u_k(a_k, a_l, x_{-kl})]$ the bimatrix game approximation (Janovskaja, 1968) between players $k$ and $l$ with all other players marginalized out; $x_{-kl}$ denotes all strategies belonging to players other than $k$ and $l$ and $u_k(a_k, a_l, x_{-kl})$ separates out $l$'s strategy $x_l$ from the rest of the players $x_{-k}$. Similarly, denote by $T_{klq}^k = \mathbb{E}_{x_{-klq}}[u_k(a_k, a_l, a_q, x_{-klq})]$ the 3-player tensor approximation to the game. Note player $k$'s utility can now be written succinctly as $u_k(x_k, x_{-k}) = x_k^\top \nabla_{x_k}^k = x_k^\top H_{kl}^k x_l = T_{klq}^k x_k x_l x_q$ for any $l, q$ where we use Einstein notation for tensor arithmetic. For convenience, define $\texttt{diag}(z)$ as the function that places a vector $z$ on the diagonal of a square matrix, and $\texttt{diag3} : z \in \mathbb{R}^d \to \mathbb{R}^{d \times d \times d}$ as a 3-tensor of shape $(d, d, d)$ where $\texttt{diag3}(z)_{lll} = z_l$.

| Loss | Function | Obstacle |
|---|---|---|
| Exploitabilty ($\epsilon$) | $\max_k \epsilon_k(\boldsymbol{x})$ | max of r.v. |
| Nikaido-Isoda (NI) | $\sum_k \epsilon_k(\boldsymbol{x})$ | max of r.v. |
| Fully-Diff. Exp | $\sum_k \sum_{a_k \in \mathcal{A}_k} [\max(0, u_k(a_k, x_{-k}) - u_k(\boldsymbol{x}))]^2$ | max of r.v. |
| Gradient-based NI | NI w/ $\text{BR}_k \leftarrow \text{aBR}_k = \Pi_\Delta \left( x_k + \eta \nabla_{x_k} u_k(\boldsymbol{x}) \right)$ | $\Pi_\Delta$ of r.v. |
| Unconstrained | Loss + Simplex Deviation Penalty | sampling from $x_k \in \mathbb{R}^{m_k}$ |

Table 1: Previous loss functions for NFGs and their obstacles to unbiased estimation. Note that $\epsilon_k(\boldsymbol{x}) = \max_z u_k(z, x_{-k}) - u_k(\boldsymbol{x})$ contains a max operator (see equivalent definition in equation (1)).

Following convention from differential geometry, let $T_v \mathcal{M}$ be the tangent space of a manifold $\mathcal{M}$ at $v$. For the interior of the $d$-action simplex $\Delta^{d-1}$, the tangent space is the same at every point, so we drop the $v$ subscript, i.e., $T\Delta^{d-1}$. We denote the projection of a vector $z \in \mathbb{R}^d$ onto this tangent space as $\Pi_{T\Delta^{d-1}}(z) = [I - \frac{1}{d}\mathbf{1}\mathbf{1}^\top]z$ and call $\Pi_{T\Delta}(\nabla_{x_k}^k)$ a *projected-gradient*. We drop $d-1$ when the dimensionality is clear from the context. Finally, let $\mathcal{U}(S)$ denote a discrete uniform distribution over elements from set $S$.

## 3 RELATED WORK

Representing the problem of computing a Nash equilibrium as an optimization problem is not new. A variety of loss functions and pseudo-distance functions have been proposed. Most of them measure some function of how much each player can exploit the joint strategy by unilaterally deviating:

$$\epsilon_k(\boldsymbol{x}) \stackrel{\text{def}}{=} u_k(\text{BR}_k, x_{-k}) - u_k(\boldsymbol{x}) \text{ where } \text{BR}_k \in \arg\max_z u_k(z, x_{-k}). \quad (1)$$

As argued in the introduction, we believe it is important to be able to subsample payoff tensors of normal-form games in order to scale to large instances. As Nash equilibria can consist of mixed strategies, it is advantageous to be able to sample from an equilibrium to estimate its exploitability $\epsilon$. However none of these losses is amenable to unbiased estimation under sampled play. Each of the functions currently explored in the literature is biased under sampled play either because 1) a random variable appears as the argument of a complex, nonlinear (non-polynomial) function or because 2) how to sample play is unclear. Exploitability, Nikaido-Isoda (NI) (Nikaidô and Isoda, 1955) (also known by `NashConv` (Lanctot et al., 2017) and ADI (Gemp et al., 2022)), as well as fully-differentiable options (Shoham and Leyton-Brown, 2008, p. 106, Eqn 4.31) introduce bias when a `max` over payoffs is estimated using samples from $\boldsymbol{x}$. Gradient-based NI (Raghunathan et al., 2019) requires projecting the result of a gradient-ascent step onto the simplex; for the same reason as the `max`, this projection is prohibitive because it is a nonlinear operation which introduces bias. Lastly, unconstrained optimization approaches (Shoham and Leyton-Brown, 2008, p. 106) that instead penalize deviation from the simplex lose the ability to sample from strategies when each iterate $\boldsymbol{x}$ is no longer a distribution (i.e., $x_k \notin \Delta^{m_k-1}$). Table 1 summarizes these complications.

## 4 NASH EQUILIBRIUM AS STOCHASTIC OPTIMIZATION

We will now develop our proposed loss function which is amenable to unbiased estimation. Subsections 4.1-4.4 provide a warm-up in which we assume an interior (fully-mixed) Nash equilibrium exists. Subsection 4.5 then shows how to relax that assumption allowing us to approximate partially mixed equilibria as well (including pure equilibria). Our key technical insight is to pay special attention to the geometry of the simplex. To our knowledge, prior works have failed to recognize the role of the tangent space $T\Delta$. Proofs are in the appendix.

### 4.1 STATIONARITY ON THE SIMPLEX INTERIOR

**Lemma 1.** *Assuming player $k$'s utility, $u_k(x_k, x_{-k})$, is concave in its own strategy $x_k$, a strategy in the interior of the simplex is a best response $\text{BR}_k$ if and only if it has zero projected-gradient[1] norm.*

---

[1]Not to be confused with the nonlinear (biased) projected gradient operator in (Hazan et al., 2017).

In NFGs, each player's utility is linear in $x_k$, thereby satisfying the concavity condition of Lemma 1.

## 4.2 PROJECTED-GRADIENT NORM AS A LOSS

An equivalent description of a Nash equilibrium is a joint strategy $\boldsymbol{x}$ where every player's strategy is a best response to the equilibrium (i.e., $x_k = \texttt{BR}_k$ so that $\epsilon_k(\boldsymbol{x}) = 0$). Lemma 1 states that any interior best response has zero *projected-gradient* norm, which inspires the following loss function

$$\mathcal{L}(\boldsymbol{x}) = \sum_k \eta_k ||\Pi_{T\Delta}(\nabla_{x_k}^k)||^2 \tag{2}$$

where each $\eta_k > 0$ represents a scalar weight, or equivalently, a step size to be explained next.

**Proposition 1.** *The loss $\mathcal{L}$ is equivalent to* `NashConv`*, but where player $k$'s best response is approximated by a single step of projected-gradient ascent with step size $\eta_k$:* $\texttt{aBR}_k = x_k + \eta_k \Pi_{T\Delta}(\nabla_{x_k}^k)$.

This connection was already pointed out in prior work for unconstrained problems (Gemp et al., 2022; Raghunathan et al., 2019), but this result is the first for strategies constrained to the simplex.

## 4.3 CONNECTION TO TRUE EXPLOITABILITY

In general, we can bound exploitability in terms of the projected-gradient norm as long as each player's utility is concave (this result extends to subgradients of non-smooth functions).

**Lemma 2.** *The amount a player can gain by exploiting a joint strategy $\boldsymbol{x}$ is upper bounded by a quantity proportional to the norm of the projected-gradient:*

$$\epsilon_k(\boldsymbol{x}) \leq \sqrt{2} ||\Pi_{T\Delta}(\nabla_{x_k}^k)||. \tag{3}$$

This bound is not tight on the boundary of the simplex, which can be seen clearly by considering $x_k$ to be part of a pure strategy equilibrium. In that case, this analysis assumes $x_k$ can be improved upon by a projected-gradient ascent step (via the equivalence pointed out in Proposition 1). However, that is false because the probability of a pure strategy cannot be increased beyond 1. We mention this to provide further intuition for why our "warm-up" loss $\mathcal{L}(\boldsymbol{x})$ is only valid for interior equilibria.

Note that $||\Pi_{T\Delta}(\nabla_{x_k}^k)|| \leq ||\nabla_{x_k}^k||$ because $\Pi_{T\Delta}$ is a projection. Therefore, this improves the naive bounds on exploitability and distance to best responses given using the "raw" gradient $\nabla_{x_k}^k$.

**Lemma 3.** *The exploitability of a joint strategy $\boldsymbol{x}$, is upper bounded by a function of $\mathcal{L}(\boldsymbol{x})$:*

$$\epsilon \leq \sqrt{\frac{2n}{\min_k \eta_k}} \sqrt{\mathcal{L}(\boldsymbol{x})} \stackrel{\text{def}}{=} f(\mathcal{L}). \tag{4}$$

## 4.4 UNBIASED ESTIMATION

As discussed in Section 3, a primary obstacle to unbiased estimation of $\mathcal{L}(\boldsymbol{x})$ is the presence of complex, nonlinear functions of random variables, with the projection of a point onto the simplex being one such example (see $\Pi_\Delta$ in Table 1). However, $\Pi_{T\Delta}$, the projection onto the *tangent space of the simplex*, is linear! This is the insight that allows us to design an unbiased estimator (Lemma 5).

Our proposed loss requires computing the squared norm of the *expected value* of the projected-gradient under the players' mixed strategies, i.e., the $l$-th entry of player $k$'s gradient equals $\nabla_{x_{kl}}^k = \mathbb{E}_{a_{-k} \sim x_{-k}} u_k(a_{kl}, a_{-k})$. By analogy, consider a random variable $Y$. In general, $\mathbb{E}[Y]^2 \neq \mathbb{E}[Y^2]$. This means that we cannot just sample projected-gradients and then compute their average norm to estimate our loss. However, consider taking two independent samples from two corresponding identically distributed, independent random variables $Y^{(1)}$ and $Y^{(2)}$. Then $\mathbb{E}[Y^{(1)}]^2 = \mathbb{E}[Y^{(1)}]\mathbb{E}[Y^{(2)}] = \mathbb{E}[Y^{(1)}Y^{(2)}]$ by properties of expected value over products of independent random variables. This is a common technique to construct unbiased estimates of expectations over polynomial functions of random variables. Proceeding in this way, define $\hat{\nabla}_{x_k}^{k(1)}$ as a random, unbiased gradient estimate (see Table 2). Let $\hat{\nabla}_{x_k}^{k(2)}$ be independent and distributed identically to $\hat{\nabla}_{x_k}^{k(1)}$. Then Lemma 5 shows

$$\mathcal{L}(\boldsymbol{x}) = \mathbb{E}[\sum_k \eta_k \underbrace{(\hat{\nabla}_{x_k}^{k(1)} - \frac{1}{m_k}(\mathbf{1}^\top \hat{\nabla}_{x_k}^{k(1)})\mathbf{1})^\top}_{\text{projected-gradient 1}} \underbrace{(\hat{\nabla}_{x_k}^{k(2)} - \frac{1}{m_k}(\mathbf{1}^\top \hat{\nabla}_{x_k}^{k(2)})\mathbf{1})}_{\text{projected-gradient 2}}] \tag{5}$$

|  | Exact | Sample Others | Sample All |
|---|---|---|---|
| Estimator of $\nabla_{x_k}^{k(p)}$ | $[u_k(a_{kl}, x_{-k})]_l$ | $[u_k(a_{kl}, a_{-k} \sim x_{-k})]_l$ | $m_k u_k(a_{kl} \sim \mathcal{U}(\mathcal{A}_k), a_{-k} \sim x_{-k})e_l$ |
| $\hat{\nabla}_{x_k}^{k(p)}$ Bounds | $[0, 1]$ | $[0, 1]$ | $[0, m_k]$ |
| $\hat{\nabla}_{x_k}^{k(p)}$ Query Cost | $\prod_{k=1}^n m_k$ | $m_k$ | $1$ |
| $\hat{\mathcal{L}}$ Bounds | $\pm 1/4 \sum_k \eta_k m_k$ | $\pm 1/4 \sum_k \eta_k m_k$ | $\pm 1/4 \sum_k \eta_k m_k^3$ |
| $\hat{\mathcal{L}}$ Query Cost | $n \prod_{k=1}^n m_k$ | $2n\bar{m}$ | $2n$ |

Table 2: Examples and Properties of Unbiased Estimators of Loss and Player Gradients ($\hat{\nabla}_{x_k}^{k(p)}$).

where $\hat{\nabla}_{x_k}^{k(p)}$ is an unbiased estimator of player $k$'s gradient. This estimator can be constructed in several ways. The most expensive, an exact estimator, is constructed by marginalizing player $k$'s payoff tensor over all other players' strategies. However, a cheaper estimate can be obtained at the expense of higher variance by approximating this marginalization with a Monte Carlo (MC) estimate of the expectation. Specifically, if we sample a single action for each of the remaining players, we can construct an unbiased estimate of player $k$'s gradient by considering the payoff of each of its actions against the sampled background strategy. Lastly, we can consider constructing an estimate of player $k$'s gradient by sampling only a single action from player $k$ to represent their entire gradient. Each of these approaches is outlined in Table 2 along with the query complexity (Babichenko, 2016) of computing the estimator and bounds on the values it can take (Lemma 9).

We can extend Lemma 3 to one that holds under $T$ samples with probability $1 - \delta$ by applying, for example, a Hoeffding bound: $\epsilon \leq f\big(\hat{\mathcal{L}}(\boldsymbol{x}) + \mathcal{O}(\sqrt{\frac{1}{T} \ln(1/\delta)})\big)$ where $\hat{\mathcal{L}}$ is an MC estimate of $\mathcal{L}$.

## 4.5 INTERIOR EQUILIBRIA

We discussed earlier that $\mathcal{L}(\boldsymbol{x})$ captures interior equilibria. But some games may only have *partially mixed* equilibria, i.e., equilibria that lie on the boundary of the simplex. We show how to circumvent this shortcoming by considering quantal response equilibria (QREs), specifically, logit equilibria. By adding an entropy bonus to each player's utility, we can

- guarantee **all** equilibria are interior,

- still obtain unbiased estimates of our loss,

- maintain an upper bound on the exploitability $\epsilon$ of any approximate Nash equilibrium in the original game (i.e., the game without an entropy bonus).

Define $u_k^\tau(\boldsymbol{x}) = u_k(\boldsymbol{x}) + \tau S(x_k)$ where Shannon entropy $S(x_k) = -\sum_l x_{kl} \ln(x_{kl})$ is 1-strongly concave with respect to the 1-norm (Beck and Teboulle, 2003). It is known that Nash equilibria of entropy-regularized games satisfy the conditions for logit equilibria (Leonardos et al., 2021), which are solutions to the fixed point equation $x_k = \texttt{softmax}(\frac{1}{\tau}\nabla_{x_k}^k)$. The $\texttt{softmax}$ should make it clear to the reader that all probabilities have positive mass at positive temperature.

Recall that in order to construct an unbiased estimate of our loss, we simply needed to construct unbiased estimates of player gradients. The introduction of the entropy term to player $k$'s utility is special in that it depends entirely on known quantities, i.e., the player's own mixed strategy. We can directly and deterministically compute $\tau \frac{dS}{dx_k} = -\tau(\ln(x_k) + \boldsymbol{1})$ and add this to our estimator of $\nabla_{x_k}^{k(p)}$: $\hat{\nabla}_{x_k}^{k\tau(p)} = \hat{\nabla}_{x_k}^{k(p)} + \tau \frac{dS}{dx_k}$. Consider our loss function refined from (2) with changes in blue:

$$\mathcal{L}^\tau(\boldsymbol{x}) = \sum_k \eta_k ||\Pi_{T\Delta}(\nabla_{x_k}^{k\tau})||^2. \tag{6}$$

As mentioned above, the utilities with entropy bonuses are still concave, therefore, a similar bound to Lemma 2 applies. We use this to prove the QRE counterpart to Lemma 3 where $\epsilon_{QRE}$ is the exploitability of an approximate equilibrium in a game with entropy bonuses.

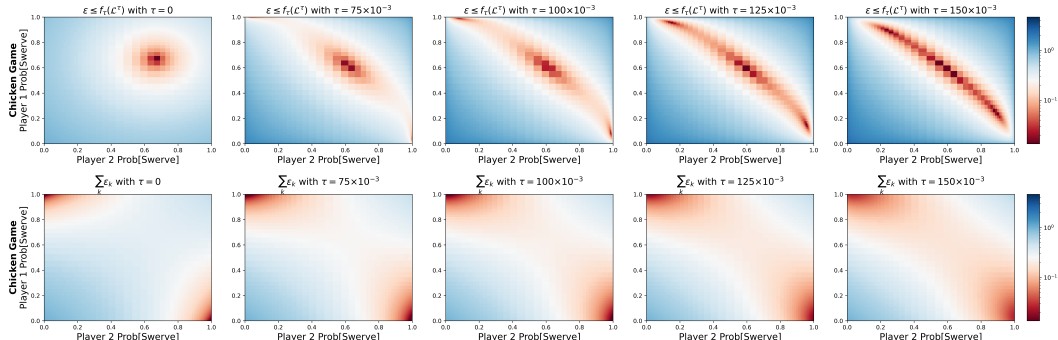

Figure 1: Effect of Sampled Play on a Biased Loss. The first row displays the expectation of the upper bound guaranteed by our proposed loss $\mathcal{L}^\tau$ with $\eta_k = 1$ for all $k$. The second row displays the expectation of NashConv under sampled play, i.e., $\sum_k \epsilon_k$ where $\epsilon_k = \mathbb{E}_{a_{-k} \sim x_{-k}}[\max_{a_k} u_k^\tau(a)] - \mathbb{E}_{a \sim x}[u_k^\tau(a)]$. To be consistent, we subtract the offset $\tau \log(m^2)$ from $f_\tau(\mathcal{L}^\tau)$ per Lemma 13, which relates the exploitability at positive temperature to that at zero temperature. The resulting loss surface clearly shows NashConv fails to recognize any interior Nash equilibrium due to its inherent bias.

**Lemma 4.** *The entropy regularized exploitability, $\epsilon_{QRE}$, of a joint strategy $x$, is upper bounded as:*

$$\epsilon_{QRE} \leq \sqrt{\frac{2n}{\min_k \eta_k}} \sqrt{\mathcal{L}^\tau(x)} \overset{\text{def}}{=} f(\mathcal{L}^\tau). \tag{7}$$

Lastly, we establish a connection between quantal response equilibria and Nash equilibria that allows us to approximate Nash equilibria in the original game via minimizing our modified loss $\mathcal{L}^\tau(x)$.

**Lemma 13** ($\mathcal{L}^\tau$ Scores Nash Equilibria). *Let $\mathcal{L}^\tau(x)$ be our proposed entropy regularized loss function and $x$ be an approximate QRE. Then it holds that*

$$\epsilon \leq \tau \log\left(\prod_k m_k\right) + \sqrt{\frac{2n}{\min_k \eta_k}} \sqrt{\mathcal{L}^\tau(x)} \overset{\text{def}}{=} f_\tau(\mathcal{L}^\tau). \tag{8}$$

This upper bound is plotted as a heatmap for a familiar Chicken game in the top row of Figure 1. First, notice how pure equilibria are not visible as minima for zero temperature, but appear for slightly warmer temperatures. Secondly, notice that NashConv in the bottom row is unable to capture the interior Nash equilibrium because of its high bias under sampled play. In contrast, our proposed loss $\mathcal{L}^\tau$ is guaranteed to capture all equilibria at low temperature $\tau$.

## 5 ANALYSIS

In the preceding section we established a loss function that upper bounds the exploitability of an approximate equilibrium. In addition, the zeros of this loss function have a one-to-one correspondence with quantal response equilibria (which approximate Nash equilibria at low temperature).

Here, we derive properties that suggest it is "easy" to optimize. While this function is generally non-convex and may suffer from a proliferation of saddle points (Figure 2), it is Lipschitz continuous (over the relevant subset of the interior) and bounded. These are two commonly made assumptions in the literature on non-convex optimization, which we leverage in Section 6. In addition, we can derive its gradient, its Hessian, and characterize its behavior around global minima.

**Lemma 14.** *The gradient of $\mathcal{L}^\tau(x)$ with respect to player $l$'s strategy $x_l$ is*

$$\nabla_{x_l} \mathcal{L}^\tau(x) = 2 \sum_k \eta_k B_{kl}^\top \Pi_T \Delta(\nabla_{x_k}^{k\tau}) \tag{9}$$

*where $B_{ll} = -\tau[I - \frac{1}{m_l}\mathbf{1}\mathbf{1}^\top]\texttt{diag}\left(\frac{1}{x_l}\right)$ and $B_{kl} = [I - \frac{1}{m_k}\mathbf{1}\mathbf{1}^\top]H_{kl}^k$ for $k \neq l$.*

Figure 2: Analysis of Loss Landscape. We reapply the analysis of (Dauphin et al., 2014), originally designed to understand the success of SGD in deep learning, to "slices" of several popular extensive form games. To construct a slice (or *meta-game*), we randomly sample 6 deterministic policies and then consider the corresponding $n$-player, 6-action normal-form game at $\tau = 0.1$ (with payoffs normalized to $[0, 1]$). The index of a critical point $\boldsymbol{x}_c$ ($\nabla_{\boldsymbol{x}}\mathcal{L}^\tau(\boldsymbol{x}_c) = \boldsymbol{0}$) indicates the fraction of negative eigenvalues in the Hessian of $\mathcal{L}^\tau$ at $\boldsymbol{x}_c$; $\alpha = 0$ indicates a local minimum, 1 a maximum, else a saddle point. We see a positive correlation between exploitability ($y$-axis), *projected*-gradient norm ($x$-axis), and $\alpha$ (color) indicating a lower prevalence of local minima at high exploitability.

**Lemma 16.** *The Hessian of $\mathcal{L}^\tau(\boldsymbol{x})$ can be written*

$$Hess(\mathcal{L}^\tau) = 2\big[\tilde{B}^\top \tilde{B} + T\Pi_{T\Delta}(\tilde{\nabla}^\tau)\big] \tag{10}$$

*where $\tilde{B}_{kl} = \sqrt{\eta_k}B_{kl}$, $\Pi_{T\Delta}(\tilde{\nabla}^\tau) = [\eta_1\Pi_{T\Delta}(\nabla^{1\tau}_{x_1}), \dots, \eta_n\Pi_{T\Delta}(\nabla^{n\tau}_{x_n})]$, and we augment $T$ (the 3-player approximation to the game, $T^k_{lqk}$) so that $T^l_{lll} = \tau\, diag3\big(\frac{1}{x_l^2}\big)$.*

At an NE, the latter term disappears because $\Pi_{T\Delta}(\nabla^{k\tau}_{x_k}) = \boldsymbol{0}$ for all $k$ (Lemma 1). If $\mathcal{X}$ was $\mathbb{R}^{n\bar{m}}$, then we could simply check if $\tilde{B}$ is full-rank to determine if $Hess \succ 0$, i.e., if $\mathcal{L}^\tau$ is locally strongly-convex. However, $\mathcal{X}$ is a simplex product, and we only care about curvature in directions toward which we can update our strategy profile $\boldsymbol{x}$. Toward that end, define $M$ to be the $n(\bar{m}+1) \times n\bar{m}$ matrix that stacks $\tilde{B}$ on top of a repeated identity matrix that encodes orthogonality to the simplex:

$$M(\boldsymbol{x}) = \begin{bmatrix} -\tau\sqrt{\eta_1}\Pi_{T\Delta}(\frac{1}{x_1}) & \sqrt{\eta_1}\Pi_{T\Delta}(H^1_{12}) & \cdots & \sqrt{\eta_1}\Pi_{T\Delta}(H^1_{1n}) \\ \vdots & \vdots & \vdots & \vdots \\ \sqrt{\eta_n}\Pi_{T\Delta}(H^n_{n1}) & \cdots & \sqrt{\eta_n}\Pi_{T\Delta}(H^n_{n,n-1}) & -\tau\sqrt{\eta_n}\Pi_{T\Delta}(\frac{1}{x_n}) \\ \boldsymbol{1}_1^\top & 0 & \cdots & 0 \\ \vdots & \vdots & \vdots & \vdots \\ 0 & \cdots & 0 & \boldsymbol{1}_n^\top \end{bmatrix} \tag{11}$$

where $\Pi_{T\Delta}(z \in \mathbb{R}^{a\times b}) = [I_a - \frac{1}{a}\boldsymbol{1}_a\boldsymbol{1}_a^\top]z$ subtracts the mean from each column of $z$ and $\frac{1}{x_k}$ is shorthand for $diag\big(\frac{1}{x_k}\big)$. If $M(\boldsymbol{x})z = \boldsymbol{0}$ for a nonzero vector $z \in \mathbb{R}^{n\bar{m}}$, this implies there exists a $z$ that 1) is orthogonal to the ones vectors of each simplex (i.e., is a valid equilibrium update direction) and 2) achieves zero curvature in the direction $z$, i.e., $z^\top(\tilde{B}^\top\tilde{B})z = z^\top(Hess)z = 0$, and so $Hess$ is not positive definite. Conversely, if $M(\boldsymbol{x})$ is of rank $n\bar{m}$ for a quantal response equilibrium $\boldsymbol{x}$, then the Hessian of $\mathcal{L}^\tau$ at $\boldsymbol{x}$ in the tangent space of the simplex product ($\mathcal{X} = \prod_k \mathcal{X}_k$) is positive definite. In this case, we call $\boldsymbol{x}$ *polymatrix*-isolated: **polymatrix** because we only require information of the local polymatrix approximation of the game (i.e., the $H^k_{kl}$ matrices) to construct $M$ and **isolated** because it implies $\boldsymbol{x}$ is not connected to any other equilibria.

**Definition 1** (*Polymatrix*-Isolated Equilibrium). *A Nash equilibrium $\boldsymbol{x}^*$ is polymatrix-isolated iff $\boldsymbol{x}^*$ is isolated according to its local polymatrix game approximation.*

By analyzing the rank of $M$, we can confirm that many classical matrix games including Rock-Paper-Scissors, Chicken, Matching Pennies, and Shapley's game all induce strongly convex $\mathcal{L}^\tau$'s at zero temperature (i.e., they have unique mixed Nash equilibria). In contrast, a game like Prisoner's Dilemma has a unique pure strategy that will not be captured by our loss at zero temperature.

## 6 ALGORITHMS

We have formally transformed the approximation of Nash equilibria in NFGs into a **stochastic** optimization problem. To our knowledge, this is the first such formulation that allows one-shot

Figure 3: Comparison of SGD on $\mathcal{L}^{\tau=0}$ against baselines on four games evaluated in (Gemp et al., 2022). The number of samples used to estimate each update iteration (i.e., minibatch size) is indicated by $s$. From left to right: 2-player, 3-action, nonsymmetric; 6-player, 5-action, nonsymmetric; 4-player, 66-action, symmetric; 3-player, 286-action, symmetric. SGD struggles at saddle points in Blotto.

unbiased Monte-Carlo estimation which is critical to introduce the use of powerful algorithms capable of solving high dimensional optimization problems. We explore two off-the-shelf approaches.

## 6.1 STOCHASTIC GRADIENT DESCENT

Stochastic gradient descent is the workhorse of high-dimensional stochastic optimization. It is guaranteed to converge to stationary points (Cutkosky et al., 2023), however, it may converge to local, rather than global minima. It also enjoys implicit gradient regularization (Barrett and Dherin, 2020), seeking "flat" minima and performs approximate Bayesian inference (Mandt et al., 2017). Despite the lack of global convergence guarantee, we find it performs well empirically in games previously examined by the literature: modified Shapley's (Ostrovski and van Strien, 2014), GAMUT D7 (Nudelman et al., 2004), Blotto (Arad and Rubinstein, 2012). Figure 3 shows SGD is competitive with scalable techniques to approximating NEs: FTRL (Shalev-Shwartz and Singer, 2006; Shalev-Shwartz et al., 2012), Regret Matching (Hart and Mas-Colell, 2000), ADIDAS/$^y QRE^{auto}$ (Gemp et al., 2022). Shapley's game induces a strongly convex $\mathcal{L}$ (see Section 5) leading to SGD's strong performance. Blotto reaches low, but nonzero $\epsilon$, demonstrating the challenges of saddle points.

## 6.2 HIGH PROBABILITY, GLOBAL POLYNOMIAL CONVERGENCE RATES VIA BANDITS

We explore one other algorithmic approach to non-convex optimization based on minimizing regret, which enjoys finite time **global** convergence rates. $\mathcal{X}$-armed bandits (Bubeck et al., 2011) systematically explore the space of solutions by refining a mesh over the joint strategy space, trading off exploration versus exploitation of promising regions. Several approaches exist (Bartlett et al., 2019; Valko et al., 2013) with open source implementations, e.g., (Li et al., 2023). Applying $\mathcal{X}$-armed bandits to our $\mathcal{L}^\tau$ can be thought of as a stochastic generalization of the *exclusion method* and other bandit approaches for Nash equilibria (Berg and Sandholm, 2017; Zhou et al., 2017).

Equipped with these techniques, we establish a high probability polynomial-time **global** convergence rate to Nash equilibria in $n$-player, general-sum games given all QREs($\tau$) are polymatrix-isolated. The quality of this approximation improves as $\tau \to 0$, at the same time increasing the constant on the convergence rate via the Lipschitz constant $\sqrt{\hat{L}}$ defined below. For clarity, we assume users provide a temperature in the form $\tau = \frac{1}{\ln(1/p)}$ with $p \in (0,1)$ which ensures all equilibria have probability mass greater than $\frac{p}{m^*}$ for all actions (Lemma 11). Lower $p$ corresponds with lower temperature.

**Theorem 4** (BLiN PAC Rate). *Assume $\eta_k = \eta = 2/\hat{L}$, $\tau = \frac{1}{\ln(1/p)}$, and a previously pulled arm is returned uniformly at random (i.e., $t \sim U([T])$). Then for any $w > 0$*

$$\epsilon_t \leq w\Big[\frac{1}{\ln(1/p)}\log\big(\prod_k m_k\big) + 2(1 + (4c^2 C_z)^{1/3})\sqrt{2n\hat{L}}\Big(\frac{\ln T}{T}\Big)^{\frac{1}{2(d_z+2)}}\Big] \qquad (12)$$

*with probability $(1 - w^{-1})(1 - 2T^{-2})$ where $m^* = \max_k m_k$, $2|\mathcal{L}^\tau| \leq c \leq \frac{1}{4}\Big(\frac{\ln(m^*)}{\ln(1/p)} + 2\Big)$ (Corollary 1), $\hat{L} = \Big(\frac{\ln(m^*)}{\ln(1/p)} + 2\Big)\Big(\frac{m^{*2}}{p\ln(1/p)} + n\bar{m}\Big)$ (Corollary 2), the zooming dimension $d_z = \frac{1}{2}n\bar{m}$, and the zooming constant $C_z = |\mathcal{X}^*|^{-1}(\frac{1}{4}r_\eta^2 \sigma_{-\infty})^{-n\bar{m}}$ (Corollary 30).*

The convergence rate for BLiN (Feng et al., 2022) depends on bounds on the exploitability in terms of the loss (Lemma 13), bounds on estimates of the loss (Corollary 1), Lipschitz bounds on the

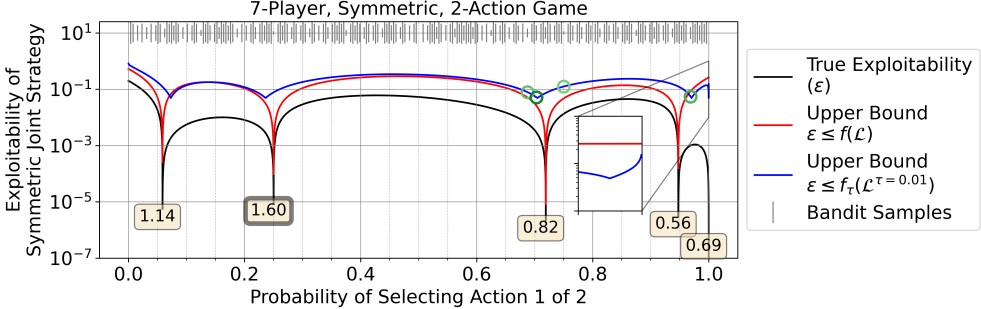

Figure 4: Bandit-based (BLiN) Nash solver applied to an artificial 7-player, symmetric, 2-action game. We search for a symmetric equilibrium, which is represented succinctly as the probability of selecting action 1. The plot shows the true exploitability $\epsilon$ of all symmetric strategies in black and indicates there exist potentially 5 NEs (the dips in the curve). Upper bounds on our unregularized loss $\mathcal{L}$ capture 4 of these equilibria, missing only the pure NE on the right. By considering our regularized loss, $\mathcal{L}^\tau$, we are able to capture this pure NE (see zoomed inset). The bandit algorithm selects strategies to evaluate, using 10 Monte-Carlo samples for each evaluation (arm pull) of $\mathcal{L}^\tau$. These samples are displayed as vertical bars above with the height of the vertical bar representing additional arm pulls. The best arms throughout search are denoted by green circles (darker indicates later in the search). The boxed numbers near equilibria display the welfare of the strategy profile.

infinity norm of the gradient (Corollary 2), and the number of distinct strategies ($n\bar{m} = \sum_k m_k$). This result further depends on the *near-optimality* or *zooming*-dimension $d_z$ and zooming constant $C_z$ which quantify the number of near optimal states. In particular, we assume $\mathcal{L}(s(z))$ is locally ($\sigma_{-\infty}$)-strongly convex with respect to $||\cdot||_\infty$ about each global optimum within a ball of radius $r_\eta$. Here, $s : [0,1]^{n(\bar{m}-1)} \to \prod_i \Delta^{m_i-1}$ is any function that maps from the unit hypercube to a product of simplices; we analyze two such maps in Appendix B.1. Next, we present an additional convergence rate result using an alternative $\mathcal{X}$-bandit approach, StoSOO (Valko et al., 2013).

**Theorem 5** (StoSOO Rate). *Corollary 1 of Valko et al. (2013) implies that with probability $(1 - w^{-1})(1-\delta)$ for any $w > 0$, a uniformly randomly drawn arm (i.e., $t \sim U([T])$) achieves*

$$\epsilon_t \le w\left[\frac{1}{\ln(1/p)}\log\left(\prod_k m_k\right) + \sqrt{n\hat{L}}\sqrt{\xi_1\sqrt{\frac{\log_b(Tk/\delta)}{2\log_b(e)k}} + \xi_2 b^{-\frac{1}{dC}\sqrt{T/k}}}\right] \qquad (13)$$

*where $d = n(\bar{m}-1)$, $\xi_1 = (2+2^{2/d})$, $\xi_2 = \frac{1}{4}db2^{(1+2/d)}$, $k = T\log_b(T)^{-3}$, $b$ is the branching factor for partitioning cells, and the near-optimality constant $C = |\mathcal{X}^*|^{-1}\sqrt{2\pi d}\left(\frac{b^2d^2}{5r_\eta^2\sigma_{-2}}\right)^{d/2}$ (Lemma 36).*

Here we assume $\mathcal{L}(s(z))$ is locally ($\sigma_{-2}$)-strongly convex with respect to $||\cdot||_2$ about each global optimum within a ball of radius $r_\eta$. Theorem 5 implies a $\tilde{\mathcal{O}}(T^{-1/4})$ global convergence rate (Proposition 2), however this is achieved only after an exponential number of burn-in iterations.

## 7 CONCLUSION

In this work, we proposed a stochastic loss for approximate Nash equilibria in normal-form games. An unbiased loss estimator of Nash equilibria is the "key" to the stochastic optimization "door" which holds a wealth of research innovations uncovered over several decades. Thus, it allows the development of new algorithmic techniques for computing equilibria. We consider bandit and vanilla SGD methods in this work, but these are only two of the many options now at our disposal (e.g, adaptive methods (Antonakopoulos et al., 2022), Gaussian processes (Calandriello et al., 2022), evolutionary algorithms (Hansen et al., 2003), etc.). Such approaches as well as generalizations of these techniques to extensive-form, imperfect-information games are promising directions for future work. Similarly to how deep learning research first balked at and then marched on to train neural networks via NP-hard non-convex optimization, we hope computational game theory can march ahead to make useful equilibrium predictions of large multiplayer systems.

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

APPENDIX: APPROXIMATING NASH EQUILIBRIA IN NORMAL-FORM GAMES VIA STOCHASTIC OPTIMIZATION

# A    LOSS WITH ITS PROPERTIES AND DERIVATIVES

In this section, we construct our loss function and derive many of its properties and derivatives (e.g., gradient and hessian) useful for analyzing and executing optimization algorithms.

## A.1    LOSS: CONNECTION TO EXPLOITABILITY, UNBIASED ESTIMATION, AND UPPER BOUNDS

### A.1.1    KKT CONDITIONS IMPLY FIXED POINT SUFFICIENCY

Consider the following constrained optimization problem:

$$\max_{\boldsymbol{x} \in \mathbb{R}^d} f(\boldsymbol{x}) \tag{14a}$$

$$s.t. \ g_i(\boldsymbol{x}) \leq 0 \ \forall i \tag{14b}$$

$$h_j(\boldsymbol{x}) = 0 \ \forall j \tag{14c}$$

where $f$ is concave and $g_i$ and $h_j$ represent inequality and equality constraints respectively. If $g_i$ and $h_i$ are affine functions, then any maximizer $\boldsymbol{x}^*$ of $f$ must satisfy the following necessary and sufficient KKT conditions (Ghojogh et al., 2021; Boyd and Vandenberghe, 2004):

- Stationarity: $\boldsymbol{0} \in \partial f(\boldsymbol{x}^*) - \sum_j \lambda_j \partial h_j(\boldsymbol{x}^*) - \sum_i \mu_i \partial g_i(\boldsymbol{x}^*)$
- Primal feasibility: $h_j(\boldsymbol{x}^*) = 0$ for all $j$ and $g_i(\boldsymbol{x}^*) \leq 0$ for all $i$
- Dual feasibility: $\mu_i \geq 0$ for all $i$
- Complementary slackness: $\mu_i g_i(\boldsymbol{x}^*) = 0$ for all $i$.

**Lemma 1.** *Assuming player $k$'s utility, $u_k(x_k, x_{-k})$, is concave in its own strategy $x_k$, a strategy in the interior of the simplex is a best response $BR_k$ if and only if it has zero projected-gradient[2] norm:*

$$BR_k \in \big( int\Delta \cap \arg\max_z u_k(z, x_{-k}) - u_k(x_k, x_{-k}) \big) \iff (BR_k \in int\Delta) \wedge (||\Pi_{T\Delta}[\nabla^k_{BR_k}]|| = 0).$$

*Proof.* Consider the problem of formally computing $\epsilon_k(\boldsymbol{x}) = \max_{z \in int\Delta} u_k(z, x_{-k}) - u_k(x_k, x_{-k})$:

$$\max_{z \in \mathbb{R}^d} u_k(z, x_{-k}) - u_k(x_k, x_{-k}) \tag{15a}$$

$$s.t. - z_k \leq 0 \ \forall k \tag{15b}$$

$$1 - \sum_k z_k = 0. \tag{15c}$$

Note that the objective is linear (concave) in $z$ and the constraints are affine, therefore the KKT conditions are necessary and sufficient for optimality. Recall that we assume that the solution $z^*$ lies in the interior of the simplex, i.e., $z_k^* > 0$ for each $k$. Also, let $e_k$ be a onehot vector, i.e., a zeros vector except with a 1 at index $k$. Mapping the KKT conditions onto this problem yields the following:

- Stationarity: $\boldsymbol{0} \in \partial u_k(z^*, x_{-k}) + \lambda \boldsymbol{1} + \sum_k \mu_k e_k$
- Primal feasibility: $\sum_k z_k^* = 1$ for all $k$
- Dual feasibility: $\mu_i \geq 0$ for all $k$
- Complementary slackness: $-\mu_k z_k^* = 0$ for all $k$.

For any point $z^* \in int\Delta$, primal feasibility will be satisfied. Given our assumption that $z_k^* > 0$, by complementary slackness and dual feasibility, each $\mu_k$ must be identically zero. This implies

---

[2]Not to be confused with the nonlinear (i.e., introduces bias) projected gradient operator introduced in (Hazan et al., 2017).

the stationarity condition can be simplified to $\mathbf{0} \in \partial u_k(z^*, x_{-k}) + \lambda\mathbf{1}$. Rearranging terms (and repurposing $\lambda$) we find that for any $z^*$, there exists a $\lambda$ such that

$$\lambda\mathbf{1} \in \partial u_k(z^*, x_{-k}). \tag{16}$$

Equivalently, $\partial u_k(z^*, x_{-k}) \propto \mathbf{1}$ at $z^* \in int\Delta$. Any vector proportional to the ones vector has zero projected-gradient norm, completing the claim: $[I - \frac{1}{m_k}\mathbf{1}\mathbf{1}^\top](\lambda\mathbf{1}) = \lambda(\mathbf{1} - \frac{m_k}{m_k}\mathbf{1}) = \mathbf{0}$. $\qquad \square$

### A.1.2 NORM OF PROJECTED-GRADIENT AND EQUIVALENCE TO NFG EXPLOITABILITY WITH APPROXIMATE BEST RESPONSES

**Proposition 1.** *The loss $\mathcal{L}$ is equivalent to* `NashConv`*, but where player $k$'s best response is approximated by a single step of projected-gradient ascent with step size $\eta_k$: $aBR_k = x_k + \eta_k\Pi_{T\Delta}(\nabla^k_{x_k})$.*

*Proof.* Define an approximate best response as the result of a player adjusting their strategy via a projected-gradient ascent step, i.e., $aBR_k = x_k + \eta_k\Pi_{T\Delta}(\nabla^k_{x_k})$ for player $k$.

In a normal form game, player $k$'s utility at this new strategy is

$$u_k(aBR_k, x_{-k}) = (\nabla^k_{x_k})^\top(x_k + \eta_k\Pi_{T\Delta}(\nabla^k_{x_k})) = u_k(\boldsymbol{x}) + \eta_k(\nabla^k_{x_k})^\top\Pi_{T\Delta}(\nabla^k_{x_k}). \tag{17}$$

Therefore, the amount player $k$ gains by playing `aBR` is

$$\hat{\epsilon}_k(\boldsymbol{x}) = u_k(aBR_k, x_{-k}) - u_k(\boldsymbol{x}) \tag{18a}$$

$$= \eta_k(\nabla^k_{x_k})^\top\Pi_{T\Delta}(\nabla^k_{x_k}) \tag{18b}$$

$$= \eta_k\big(\nabla^k_{x_k} - \frac{1}{m_k}(\mathbf{1}^\top\nabla^k_{x_k})\mathbf{1}\big)^\top\Pi_{T\Delta}(\nabla^k_{x_k}) \tag{18c}$$

$$= \eta_k||\Pi_{T\Delta}(\nabla^k_{x_k})||^2 \tag{18d}$$

where the third equality follows from the fact that the projected-gradient, $\Pi_{T\Delta}(\nabla^k_{x_k})$, is orthogonal to the ones vector. $\qquad \square$

### A.1.3 CONNECTION TO TRUE EXPLOITABILITY

**Lemma 2.** *The amount a player can gain by deviating is upper bounded by a quantity proportional to the norm of the projected-gradient:*

$$\epsilon_k(\boldsymbol{x}) \leq \sqrt{2}||\Pi_{T\Delta}(\nabla^k_{x_k})||. \tag{19}$$

*Proof.* Let $z$ be any point on the simplex. Then by concavity of $u_k$ with respect to $z$,

$$u_k(z, x_{-k}) - u_k(\boldsymbol{x}) \leq (\nabla^k_{x_k})^\top(z - x_k) \tag{20a}$$

$$= (\nabla^k_{x_k})^\top(z - x_k) - \frac{1}{m_k}(\mathbf{1}^\top\nabla^k_{x_k})\overbrace{\mathbf{1}^\top(z - x_k)}^{1-1=0} \tag{20b}$$

$$= (\Pi_{T\Delta}(\nabla^k_{x_k}))^\top\underbrace{(z - x_k)}_{\texttt{Diam}(\Delta)\leq\sqrt{2}} \tag{20c}$$

$$\leq \sqrt{2}||\Pi_{T\Delta}(\nabla^k_{x_k})||. \quad \text{(Cauchy-Schwarz)} \tag{20d}$$

$\qquad \square$

Continuing, we can prove a bound on $\epsilon$ in terms of the projected-gradient loss:

**Lemma 3.** *The exploitability, $\epsilon$, of a joint strategy $\boldsymbol{x}$, is upper bounded as a function of our proposed loss:*

$$\epsilon \leq \sqrt{\frac{2n}{\min_k \eta_k}}\sqrt{\mathcal{L}(\boldsymbol{x})}. \tag{21}$$

*Proof.*

$$\epsilon = \max_k \max_z u_k(z, x_{-k}) - u_k(\boldsymbol{x}) \tag{22a}$$

$$\leq \sum_k \max_z u_k(z, x_{-k}) - u_k(\boldsymbol{x}) \tag{22b}$$

$$\leq \sum_k \sqrt{2} ||\Pi_{T\Delta}(\nabla_{x_k}^k)||_2 \qquad \text{(Lemma 2)} \tag{22c}$$

$$= \sqrt{2} \Big|\Big| ||\Pi_{T\Delta}(\nabla_{x_1}^1)||_2, \ldots, ||\Pi_{T\Delta}(\nabla_{x_n}^n)||_2 \Big|\Big|_1 \tag{22d}$$

$$\leq \sqrt{2n} \Big|\Big| ||\Pi_{T\Delta}(\nabla_{x_1}^1)||_2, \ldots, ||\Pi_{T\Delta}(\nabla_{x_n}^n)||_2 \Big|\Big|_2 \tag{22e}$$

$$= \sqrt{2n} \sqrt{\sum_k ||\Pi_{T\Delta}(\nabla_{x_k}^k)||_2^2} \tag{22f}$$

$$= \sqrt{2n} \sqrt{\sum_k \Big(\frac{1}{\eta_k}\Big) \eta_k ||\Pi_{T\Delta}(\nabla_{x_k}^k)||_2^2} \tag{22g}$$

$$\leq \sqrt{\frac{2n}{\min_k \eta_k}} \sqrt{\sum_k \eta_k ||\Pi_{T\Delta}(\nabla_{x_k}^k)||_2^2} \tag{22h}$$

$$= \sqrt{\frac{2n}{\min_k \eta_k}} \sqrt{\mathcal{L}(\boldsymbol{x})}. \tag{22i}$$

$\square$

**Lemma 4.** *The entropy regularized exploitability, $\epsilon_{QRE}$, of a joint strategy $\boldsymbol{x}$, is upper bounded as a function of our proposed loss:*

$$\epsilon_{QRE} \leq \sqrt{\frac{2n}{\min_k \eta_k}} \sqrt{\mathcal{L}^\tau(\boldsymbol{x})}. \tag{23}$$

*Proof.* Recall that $u_k^\tau(x_k, x_{-k})$ is also concave with respect to $x_k$. Then

$$\epsilon_{QRE} = \max_k \max_z u_k^\tau(z, x_{-k}) - u_k^\tau(\boldsymbol{x}) \tag{24a}$$

$$\leq \sum_k \max_z u_k^\tau(z, x_{-k}) - u_k^\tau(\boldsymbol{x}) \tag{24b}$$

$$\leq \sum_k \sqrt{2} ||\Pi_{T\Delta}(\nabla_{x_k}^{k\tau})||_2 \qquad \text{(Lemma 2)} \tag{24c}$$

$$= \sqrt{2} \Big|\Big| ||\Pi_{T\Delta}(\nabla_{x_1}^{1\tau})||_2, \ldots, ||\Pi_{T\Delta}(\nabla_{x_n}^{n\tau})||_2 \Big|\Big|_1 \tag{24d}$$

$$\leq \sqrt{2n} \Big|\Big| ||\Pi_{T\Delta}(\nabla_{x_1}^{1\tau})||_2, \ldots, ||\Pi_{T\Delta}(\nabla_{x_n}^{n\tau})||_2 \Big|\Big|_2 \tag{24e}$$

$$= \sqrt{2n} \sqrt{\sum_k ||\Pi_{T\Delta}(\nabla_{x_k}^{k\tau})||_2^2} \tag{24f}$$

$$\leq \sqrt{2n} \sqrt{\sum_k \Big(\frac{1}{\eta_k}\Big) \eta_k ||\Pi_{T\Delta}(\nabla_{x_k}^{k\tau})||_2^2} \tag{24g}$$

$$\leq \sqrt{\frac{2n}{\min_k \eta_k}} \sqrt{\sum_k \eta_k ||\Pi_{T\Delta}(\nabla_{x_k}^{k\tau})||_2^2} \tag{24h}$$

$$= \sqrt{\frac{2n}{\min_k \eta_k}} \sqrt{\mathcal{L}^\tau(\boldsymbol{x})}. \tag{24i}$$

$\square$

### A.1.4 UNBIASED ESTIMATION

**Lemma 5.** *An unbiased estimate of $\mathcal{L}(\boldsymbol{x})$ can be obtained by drawing two samples (pure strategies) from each players' mixed strategy and observing payoffs.*

*Proof.* Define $\hat{\nabla}_{x_k}^{k(1)}$ as a random, unbiased gradient estimate (see Table 2). Let $\hat{\nabla}_{x_k}^{k(2)}$ be independent and distributed identically to $\hat{\nabla}_{x_k}^{k(1)}$. Then,

$$\mathbb{E}_{\boldsymbol{a}^{(1)}\sim\boldsymbol{x},\boldsymbol{a}^{(2)}\sim\boldsymbol{x}}[\sum_k \eta_k \underbrace{(\hat{\nabla}_{x_k}^{k(1)} - \frac{1}{m_k}(\mathbf{1}^\top\hat{\nabla}_{x_k}^{k(1)})\mathbf{1})^\top}_{\text{projected-gradient 1}} \underbrace{(\hat{\nabla}_{x_k}^{k(2)} - \frac{1}{m_k}(\mathbf{1}^\top\hat{\nabla}_{x_k}^{k(2)})\mathbf{1})}_{\text{projected-gradient 2}}] \tag{25a}$$

$$= \sum_k \eta_k \mathbb{E}_{\boldsymbol{a}^{(1)}\sim\boldsymbol{x},\boldsymbol{a}^{(2)}\sim\boldsymbol{x}}[(\hat{\nabla}_{x_k}^{k(1)} - \frac{1}{m_k}(\mathbf{1}^\top\hat{\nabla}_{x_k}^{k(1)})\mathbf{1})^\top (\hat{\nabla}_{x_k}^{k(2)} - \frac{1}{m_k}(\mathbf{1}^\top\hat{\nabla}_{x_k}^{k(2)})\mathbf{1})] \tag{25b}$$

$$= \sum_k \eta_k \mathbb{E}_{\boldsymbol{a}^{(1)}\sim\boldsymbol{x}}[(\hat{\nabla}_{x_k}^{k(1)} - \frac{1}{m_k}(\mathbf{1}^\top\hat{\nabla}_{x_k}^{k(1)})\mathbf{1})]^\top \mathbb{E}_{\boldsymbol{a}^{(2)}\sim\boldsymbol{x}}[(\hat{\nabla}_{x_k}^{k(2)} - \frac{1}{m_k}(\mathbf{1}^\top\hat{\nabla}_{x_k}^{k(2)})\mathbf{1})] \tag{25c}$$

$$= \sum_k \eta_k (\nabla_{x_k}^k - \frac{1}{m_k}(\mathbf{1}^\top\nabla_{x_k}^k)\mathbf{1})^\top (\nabla_{x_k}^k - \frac{1}{m_k}(\mathbf{1}^\top\nabla_{x_k}^k)\mathbf{1}) \tag{25d}$$

$$= \sum_k \eta_k ||\Pi_{T\Delta}(\nabla_{x_k}^k)||^2 \tag{25e}$$

$$= \mathcal{L}(\boldsymbol{x}) \tag{25f}$$

where the first equality follows from linearity of expectation, the second from independence of random variables, and the third from $\mathbb{E}_{\boldsymbol{a}^{(p)}\sim\boldsymbol{x}}[\hat{\nabla}_{x_k}^{k(p)}] = \nabla_{x_k}^k$, i.e., $\hat{\nabla}_{x_k}^{k(p)}$ is an unbiased estimator of player $k$'s gradient. Therefore, equation (25a) comprises an unbiased estimate of $\mathcal{L}(\boldsymbol{x})$ proving the claim.

$\square$

**Lemma 6.** *The loss formed as the sum of the squared norms of the projected-gradients, $\mathcal{L}^\tau$, can be decomposed into three terms as follows:*

$$\mathcal{L}^\tau(\boldsymbol{x}) = \underbrace{\sum_k \eta_k x_q^\top B_{kq}^\top B_{kq} x_q}_{(A)} + \underbrace{2\sum_k \eta_k E_k^\top B_{kq} x_q}_{(B)} + \underbrace{\sum_k \eta_k E_k^\top E_k}_{(C)} \tag{26}$$

*where $q$ is any player other than $k$.*

*Proof.* Let $S^\tau = -\tau \sum_l x_{kl} \ln(x_{kl})$ so that $\frac{\partial S^\tau}{\partial x_k} = -\tau(\ln(x_k) + \mathbf{1})$. Note that $\Pi_{T\Delta}[\frac{\partial S^\tau}{\partial x_k}] = -\tau\Pi_{T\Delta}[\ln(x_k)]$.

$$\mathcal{L}^\tau(\boldsymbol{x}) = \sum_k \eta_k (\Pi_{T\Delta}(\nabla_{x_k}^{k\tau}))^\top \Pi_{T\Delta}(\nabla_{x_k}^{k\tau}) \tag{27a}$$

$$= \sum_k \eta_k [H_{kq}^k x_q + \frac{\partial S^\tau}{\partial x_k}]^\top [I - \frac{1}{m_k}\mathbf{1}\mathbf{1}^\top][I - \frac{1}{m_k}\mathbf{1}\mathbf{1}^\top][H_{kq}^k x_q + \frac{\partial S^\tau}{\partial x_k}] \tag{27b}$$

$$= \sum_k \eta_k \Big( x_q^\top [H_{kq}^k]^\top [I - \frac{1}{m_k}\mathbf{1}\mathbf{1}^\top]^2 [H_{kq}^k] x_q + 2[\frac{\partial S^\tau}{\partial x_k}]^\top [I - \frac{1}{m_k}\mathbf{1}\mathbf{1}^\top]^2 [H_{kq}^k x_q] \tag{27c}$$

$$+ [\frac{\partial S^\tau}{\partial x_k}]^\top [I - \frac{1}{m_k}\mathbf{1}\mathbf{1}^\top]^2 [\frac{\partial S^\tau}{\partial x_k}] \Big) \tag{27d}$$

$$= \underbrace{\sum_k \eta_k x_q^\top B_{kq}^\top B_{kq} x_q}_{(A)} + \underbrace{2\sum_k \eta_k E_k^\top B_{kq} x_q}_{(B)} + \underbrace{\sum_k \eta_k E_k^\top E_k}_{(C)} \tag{27e}$$

where $B_{kq} = [I - \frac{1}{m_k}\mathbf{1}\mathbf{1}^\top]H_{kq}^k$ and $E_k = [I - \frac{1}{m_k}\mathbf{1}\mathbf{1}^\top][\frac{\partial S^\tau}{\partial x_k}] = -\tau[I - \frac{1}{m_k}\mathbf{1}\mathbf{1}^\top]\ln(x_k)$.

$\square$

A.1.5 BOUND ON LOSS

By Proposition 1, Equation (18d), we can also rewrite this loss as a weighted sum of 2-norms, $\mathcal{L}(\boldsymbol{x}) = \sum_k \eta_k ||\nabla_{x_k}^k - \mu_k||_2^2$ where $\mu_k = \frac{1}{m_k}(\mathbf{1}^\top \nabla_{x_k}^k) \in [0,1]$ for brevity. This will allow us to more easily analyze our loss.

**Lemma 7.** *Assume payoffs are bounded in $[0,1]$, then setting $\eta_k \leq \frac{4}{nm_k}$ or $\eta_k \leq \frac{4}{n\bar{m}}$ or $\sum_k \eta_k \leq \frac{4}{\bar{m}}$ ensures $0 \leq \mathcal{L}(x) \leq 1$ for all $x \in \mathcal{X}$.*

*Proof.*

$$0 \leq \mathcal{L}(\boldsymbol{x}) = \sum_k \eta_k ||\nabla_{x_k}^k - \mu_k||_2^2 \tag{28a}$$

$$= \sum_k \eta_k m_k \Big[\frac{1}{m_k} \sum_l (\nabla_{x_{kl}}^k - \mu_k)^2\Big] \tag{28b}$$

$$= \sum_k \eta_k m_k Var[\nabla_{x_k}^k] \tag{28c}$$

$$\leq \frac{1}{4} \sum_k \eta_k m_k \tag{28d}$$

$$\leq \frac{1}{4}(\max_k \eta_k)\Big(\sum_k m_k\Big) \tag{28e}$$

$$= \frac{1}{4}(\max_k \eta_k)n\bar{m} \leq 1 \tag{28f}$$

$$\implies (\max_k \eta_k) \leq \frac{4}{n\bar{m}} \tag{28g}$$

where the first inequality follows from Popoviciu's inequality (Popoviciu, 1935): the variance of a bounded random variable $X$ is upper bounded by $Var[X] \leq \frac{1}{4}(\max_X - \min_X)^2$. $\square$

Next, we establish the following useful lemmas.

**Lemma 8.** *The matrix $I - \frac{1}{m_k}\mathbf{1}\mathbf{1}^\top$ is a projection matrix and therefore idempotent. It is also symmetric, which implies it is its own square root.*

*Proof.*

$$[I - \frac{1}{m_k}\mathbf{1}\mathbf{1}^\top]^\top[I - \frac{1}{m_k}\mathbf{1}\mathbf{1}^\top] = I - \frac{2}{m_k}\mathbf{1}\mathbf{1}^\top + \frac{1}{m_k^2}\mathbf{1}(\mathbf{1}^\top\mathbf{1})\mathbf{1}^\top \tag{29a}$$

$$= I - \frac{2}{m_k}\mathbf{1}\mathbf{1}^\top + \frac{1}{m_k}\mathbf{1}\mathbf{1}^\top \tag{29b}$$

$$= [I - \frac{1}{m_k}\mathbf{1}\mathbf{1}^\top]. \tag{29c}$$

$\square$

**Lemma 9.** *The product $A[I_m - \frac{1}{m}\mathbf{1}_m\mathbf{1}_m^\top]^p B$ for any integer $p > 0$ has entries whose absolute value is bounded by $\frac{m}{4}(A_{\max} - A_{\min})(B_{\max} - B_{\min})$ where $A_{\min}, A_{\max}, B_{\min}, B_{\max}$ represent the minima and maxima of the matrices respectively.*

*Proof.* The matrix $[I - \frac{1}{m}\mathbf{1}\mathbf{1}^\top]$ is idempotent (Lemma 8) so we can rewrite the product for any $p$ as

$$A[I - \frac{1}{m}\mathbf{1}\mathbf{1}^\top][I - \frac{1}{m}\mathbf{1}\mathbf{1}^\top]B. \tag{30}$$

The matrix $[I - \frac{1}{m}\mathbf{1}\mathbf{1}^\top]$ has the property that it removes the mean from every row of a matrix when right multiplied against it, i.e., $A[I - \frac{1}{m}\mathbf{1}\mathbf{1}^\top]$ removes the means from the rows of $A$. Similarly, left multiplying it removes the means from the columns. Let $\tilde{A}$ and $\tilde{B}$ represent these mean-centered

results respectively. The absolute value of the $ij$th entry in the resulting product can then be recognized as

$$\Big| \sum_k \tilde{A}_{ik} \tilde{B}_{kj} \Big| = \Big| \sum_k \Big( A_{ik} - \frac{1}{m} \sum_{k'} A_{ik'} \Big) \Big( B_{kj} - \frac{1}{m} \sum_{k'} B_{k'j} \Big) \Big| \tag{31a}$$

$$= |m \cdot Corr(A_{i,\cdot}, B_{\cdot,j}) \cdot \sigma_{A_{i,\cdot}} \sigma_{B_{\cdot,j}}| \tag{31b}$$

$$\leq m \sigma_{A_{i,\cdot}} \sigma_{B_{\cdot,j}}. \tag{31c}$$

By Popoviciu's inequality (Popoviciu, 1935), we know the variance of a bounded random variable $X$ is upper bounded by $Var[X] \leq \frac{1}{4}(\max_X - \min_X)^2$. Hence its standard deviation is bounded by $Std[X] \leq \frac{1}{2}(\max_X - \min_X)$. Plugging these bounds for $A$ and $B$ into equation (31c) completes the claim. $\qquad \square$

**Lemma 10.** *Assume payoffs are bounded in* $[0, 1]$*, then*

$$|\mathcal{L}^\tau(\boldsymbol{x})| \leq \frac{1}{4} \big( \max_k \eta_k \big) n \bar{m} \big( \tau \ln \big( \frac{1}{x_{\min}} \big) + 1 \big)^2 \tag{32}$$

*for any* $\boldsymbol{x}$ *such that* $x_{kl} \geq x_{\min} \; \forall k, l$.

*Proof.* First note that for payoffs in $[0, 1]$, the entries in

$$\Pi_{T\Delta}(\nabla_{x_k}^{k\tau}) = \Pi_{T\Delta}\big( \nabla_{x_k}^k - \tau(\ln(x_k) + \mathbf{1}) \big) = \Pi_{T\Delta}\big( \nabla_{x_k}^k - \tau \ln(x_k) \big) \tag{33}$$

are bounded within $[0, \tau \ln(\frac{1}{x_{\min}}) + 1]$ with a range of $\tau \ln(\frac{1}{x_{\min}}) + 1$. Then starting from the definition of $\mathcal{L}^\tau$ and applying Lemma 9, we find

$$|\mathcal{L}^\tau(\boldsymbol{x})| = \Big| \sum_k \eta_k ||\Pi_{T\Delta}(\nabla_{x_k}^{k\tau})||^2 \Big| \tag{34a}$$

$$\leq \frac{1}{4} \sum_k \eta_k m_k \big( \tau \ln \big( \frac{1}{x_{\min}} \big) + 1 \big)^2 \tag{34b}$$

$$\leq \frac{1}{4} \big( \tau \ln \big( \frac{1}{x_{\min}} \big) + 1 \big)^2 \big( \max_k \eta_k \big) \sum_k m_k \tag{34c}$$

$$= \frac{1}{4} \big( \max_k \eta_k \big) n \bar{m} \big( \tau \ln \big( \frac{1}{x_{\min}} \big) + 1 \big)^2. \tag{34d}$$

$\qquad \square$

## A.2 QREs Approximate NEs at Low Temperature

**Lemma 11.** *Setting* $\tau = \ln(1/p)^{-1}$ *with* $p \in (0, 1)$ *and payoffs in* $[0, 1]$ *ensures that any QRE contains probabilities greater than* $\frac{p}{\max_k m_k}$.

*Proof.* Let $m^* = \max_k m_k$ and $\nabla_{x_k}^k$ be player $k$'s gradient. Recall that a QRE($\tau$), specifically a logit equilibrium, satisfies the fixed point equation $x_k = \texttt{softmax}(\frac{\nabla_{x_k}^k}{\tau})$ for all $k$. Then the probability for any action in a QRE strategy profile, $\boldsymbol{x}$, is lower bounded as

$$x_{kl} \geq \min_k \min_l \{x_{kl}\} \geq \min_{\boldsymbol{x}' | \boldsymbol{x}' \text{ is QRE}} \min_k \min_l \{x'_{kl}\} \tag{35a}$$

$$= \min_{\nabla_{x_k}^k} \min_k \min_l \Big[ \texttt{softmax}\Big( \frac{\nabla_{x_k}^k}{\tau} \Big) \Big]_l \tag{35b}$$

$$= \frac{e^0}{(m^* - 1)e^{\frac{1}{\tau}} + e^0} \tag{35c}$$

$$\overset{\text{def}}{=} x_{\min} \tag{35d}$$

where equation (35c) follows from minimizing the numerator and maximizing the denominator of the softmax formula subject to the payoff constraints.

Rearranging terms, we find

$$e^{\frac{1}{\tau}} = \frac{1}{m^* - 1}\Big(\frac{1}{x_{\min}} - 1\Big) \implies \tau = \frac{1}{\ln\Big(\frac{1}{m^*-1}\big(\frac{1}{x_{\min}} - 1\big)\Big)}. \tag{36}$$

Let $p \in (0, 1)$ such that $x_{\min} = \frac{p}{\max_k m_k} = \frac{p}{m^*}$, then

$$\tau = \frac{1}{\ln\Big(\frac{1}{m^*-1}\big(\frac{1}{x_{\min}} - 1\big)\Big)} \tag{37a}$$

$$= \frac{1}{\ln\Big(\frac{1}{m^*-1}\big(\frac{m^*}{p} - 1\big)\Big)} \tag{37b}$$

$$= \frac{1}{\ln\Big(\frac{m^*-p}{m^*-1}\frac{1}{p}\Big)} \tag{37c}$$

$$\leq \frac{1}{\ln(\frac{1}{p})}. \tag{37d}$$

This implies if we set $\tau = \ln(1/p)^{-1}$, then we are guaranteed that all QREs contain probabilities greater than $x_{\min} = \frac{p}{\max_k m_k}$. $\qquad\square$

**Corollary 1.** *Assume payoffs are bounded in $[0, 1]$ and $\tau = \ln(1/p)^{-1}$, then*

$$|\mathcal{L}^\tau(\boldsymbol{x})| \leq \frac{1}{4}(\max_k \eta_k)n\bar{m}\Big(\frac{\ln(m^*)}{\ln(1/p)} + 2\Big)^2. \tag{38}$$

*Proof.* Starting with Lemma 10 and applying Lemma 11 , we find

$$|\mathcal{L}^\tau(\boldsymbol{x})| \leq \frac{1}{4}(\max_k \eta_k)n\bar{m}\Big(\tau \ln\Big(\frac{1}{x_{\min}}\Big) + 1\Big)^2 \tag{39a}$$

$$= \frac{1}{4}(\max_k \eta_k)n\bar{m}\Big(\frac{1}{\ln(1/p)} \ln\Big(\frac{m^*}{p}\Big) + 1\Big)^2 \tag{39b}$$

$$= \frac{1}{4}(\max_k \eta_k)n\bar{m}\Big(\frac{1}{\ln(1/p)}\big(\ln(m^*) + \ln(1/p)\big) + 1\Big)^2 \tag{39c}$$

$$= \frac{1}{4}(\max_k \eta_k)n\bar{m}\Big(\frac{\ln(m^*)}{\ln(1/p)} + 2\Big)^2. \tag{39d}$$

$$\square$$

**Lemma 12** (Low Temperature Approximate QREs are Approximate Nash Equilibria)**.** *Let $\nabla_{x_k}^{k\tau}$ be player $k$'s entropy regularized gradient and $\boldsymbol{x}$ be an approximate QRE. Then it holds that*

$$u_k(BR_k, x_{-k}) - u_k(\boldsymbol{x}) \leq \tau \log(m_k) + \sqrt{2}||\Pi_{T\Delta}(\nabla_{x_k}^{k\tau})||. \tag{40}$$

*Proof.* Beginning with the definition of exploitability, we find

$$u_k(\mathrm{BR}_k, x_{-k}) - u_k(\boldsymbol{x}) = \big(u_k(\mathrm{BR}_k, x_{-k}) + \tau S(\mathrm{BR}_k) - \tau S(\mathrm{BR}_k)\big) \tag{41a}$$
$$- \big(u_k(\boldsymbol{x}) + \tau S(x_k) - \tau S(x_k)\big)$$

$$= u_k^\tau(\mathrm{BR}_k, x_{-k}) - u_k^\tau(\boldsymbol{x}) + \tau\big(S(x_k) - S(\mathrm{BR}_k)\big) \tag{41b}$$

$$\leq \max_{z \in \Delta^{m_k-1}} u_k^\tau(z, x_{-k}) - u_k^\tau(\boldsymbol{x}) + \tau \max_{z' \in \Delta^{m_k-1}} S(z') \tag{41c}$$

$$\leq \sqrt{2}||\Pi_{T\Delta}(\nabla_{x_k}^{k\tau})|| + \tau \max_{z' \in \Delta^{m_k-1}} S(z') \tag{41d}$$

$$\leq \sqrt{2}||\Pi_{T\Delta}(\nabla_{x_k}^{k\tau})|| + \tau \log(m_k) \tag{41e}$$

where the second equality follows from the definition of player $k$'s entropy regularized utility $u_k^\tau$, the first inequality from nonnegativity of entropy $S$, the second inequality from concavity of $u_k^\tau$ with respect to its first argument (Lemma 2), and the last from the maximum possible value of Shannon entropy over distributions on $m_k$ actions. $\qquad\square$

**Lemma 13** ($\mathcal{L}^\tau$ *Scores Nash Equilibria*). *Let $\mathcal{L}^\tau(\boldsymbol{x})$ be our proposed entropy regularized loss function and $\boldsymbol{x}$ be an approximate QRE. Then it holds that*

$$\epsilon \leq \tau \log \Big( \prod_k m_k \Big) + \sqrt{\frac{2n}{\min_k \eta_k}} \sqrt{\mathcal{L}^\tau(\boldsymbol{x})}. \tag{42}$$

*Proof.* Beginning with the definition of exploitability and applying Lemma 12, we find

$$\epsilon = \max_k u_k(\text{BR}_k, x_{-k}) - u_k(\boldsymbol{x}) \qquad (\text{recall each } \epsilon_k \geq 0) \tag{43a}$$

$$\leq \sum_k u_k(\text{BR}_k, x_{-k}) - u_k(\boldsymbol{x}) \tag{43b}$$

$$\leq \sum_k \Big[ \tau \log(m_k) + \sqrt{2} ||\Pi_{T\Delta}(\nabla_{x_k}^{k\tau})|| \Big] \tag{43c}$$

$$= \tau \sum_k \log(m_k) + \sum_k \sqrt{2} ||\Pi_{T\Delta}(\nabla_{x_k}^{k\tau})|| \tag{43d}$$

$$\leq \tau \log \Big( \prod_k m_k \Big) + \sqrt{\frac{2n}{\min_k \eta_k}} \sqrt{\mathcal{L}^\tau(\boldsymbol{x})}. \tag{43e}$$

where the last inequality follows from the same steps (24c)-(24i) outlined in Lemma 4, which established the relationship between $\mathcal{L}(\boldsymbol{x})$ and $\epsilon$. $\qquad\square$

### A.3 GRADIENT OF LOSS

**Lemma 14.** *The gradient of $\mathcal{L}^\tau(\boldsymbol{x})$ with respect to player $l$'s strategy $x_l$ is*

$$\nabla_{x_l} \mathcal{L}(\boldsymbol{x}) = 2 \sum_k \eta_k B_{kl}^\top \Pi_{T\Delta}(\nabla_{x_k}^{k\tau}) \tag{44}$$

*where $B_{ll} = -\tau[I - \frac{1}{m_l}\mathbf{1}\mathbf{1}^\top]\texttt{diag}\big(\frac{1}{x_l}\big)$ and $B_{kl} = [I - \frac{1}{m_k}\mathbf{1}\mathbf{1}^\top]H_{kl}^k$ for $k \neq l$.*

*Proof.* Recall from Lemma 6 that the loss can be decomposed as $\mathcal{L}^\tau(\boldsymbol{x}) = (A) + (B) + (C)$. Then

$$D_{x_l}[(A)] = D_{x_l}\Big[\sum_k \eta_k x_q^\top B_{kq}^\top B_{kq} x_q\Big] = 2 \sum_{k \neq l} \eta_k B_{kl}^\top B_{kl} x_l \tag{45}$$

where $q \neq k$ and $B_{kq} = [I - \frac{1}{m_k}\mathbf{1}\mathbf{1}^\top][H_{kq}^k]$ does not depend on $x_k$.

Also, letting $B_{ll} = -\tau[I - \frac{1}{m_l}\mathbf{1}\mathbf{1}^\top]\texttt{diag}\big(\frac{1}{x_l}\big)$,

$$D_{x_l}[(B)] = D_{x_l}[-2\tau \sum_k \eta_k \ln(x_k)^\top B_{kq} x_q] \tag{46a}$$

$$= -2\tau\Big[\eta_l D_{x_l}[\ln(x_l)^\top B_{lq} x_q] + \sum_{k \neq l} \eta_k D_{x_l}[\ln(x_k)^\top B_{kl} x_l]\Big] \tag{46b}$$

$$= -2\tau\Big[\eta_l \texttt{diag}\big(\frac{1}{x_l}\big) B_{lq} x_q + \sum_{k \neq l} \eta_k B_{kl}^\top \ln(x_k)\Big] \tag{46c}$$

$$= -2\tau\Big[\eta_l \Big([I - \frac{1}{m_l}\mathbf{1}\mathbf{1}^\top]\texttt{diag}\big(\frac{1}{x_l}\big)\Big)^\top \Pi_{T\Delta}(\nabla^l) + \sum_{k \neq l} \eta_k B_{kl}^\top \ln(x_k)\Big] \tag{46d}$$

$$= 2\Big[\eta_l B_{ll}^\top \Pi_{T\Delta}(\nabla^l) - \tau \sum_{k \neq l} \eta_k B_{kl}^\top \ln(x_k)\Big]. \tag{46e}$$

And

$$D_{x_l}[(C)] = D_{x_l}\big[\sum_k \eta_k \tau^2 \ln(x_k)^\top [I - \frac{1}{m_k}\mathbf{1}\mathbf{1}^\top]\ln(x_k)\big] \tag{47a}$$

$$= 2\tau^2\Big[\eta_l \mathtt{diag}\big(\frac{1}{x_l}\big)[I - \frac{1}{m_l}\mathbf{1}\mathbf{1}^\top]\ln(x_l)\Big] \tag{47b}$$

$$= -2\tau\eta_l\Big([I - \frac{1}{m_l}\mathbf{1}\mathbf{1}^\top]\mathtt{diag}\big(\frac{1}{x_l}\big)\Big)^\top \Pi_{T\Delta}(-\tau \ln(x_l)) \tag{47c}$$

$$= 2\eta_l B_{ll}^\top \Pi_{T\Delta}(-\tau \ln(x_l)). \tag{47d}$$

Putting these together, we find

$$\nabla_{x_l}\mathcal{L}(\boldsymbol{x}) = 2\sum_{k\neq l}\eta_k B_{kl}^\top(B_{kl}x_l - \tau\ln(x_k)) + 2\eta_l B_{ll}^\top\big[\Pi_{T\Delta}(\nabla^l) + \Pi_{T\Delta}(-\tau\ln(x_l))\big] \tag{48a}$$

$$= 2\eta_l B_{ll}^\top \Pi_{T\Delta}(\nabla_{x_k}^{k\tau}) + 2\sum_{k\neq l}\eta_k B_{kl}^\top \Pi_{T\Delta}(\nabla_{x_k}^{k\tau}) \tag{48b}$$

$$= 2\sum_k \eta_k B_{kl}^\top \Pi_{T\Delta}(\nabla_{x_k}^{k\tau}). \tag{48c}$$

$\square$

### A.3.1 Unbiased Estimation

In order to construct an unbiased estimate for each $B_{kl}$, we will need to form an independent unbiased estimate of $H_{kl}^k$. Recall that $H_{kl}^k$ is simply the expected bimatrix game between players $k$ and $l$ when all other players sample their actions according to their current strategies.

### A.4 Bound on Gradient / Lipschitz Property

**Lemma 15.** *Assume payoffs are upper bounded by* 1, *then the infinity norm of the gradient is bounded as*

$$||\nabla_{\boldsymbol{x}}\mathcal{L}^\tau(\boldsymbol{x})||_\infty \leq \frac{1}{2}(\max_k \eta_k)\big(\tau\ln\big(\frac{1}{x_{\min}}\big) + 1\big)\Big[\tau m^*\big(\frac{1}{x_{\min}} - 1\big) + n\bar{m}\Big]. \tag{49}$$

*Proof.* Recall from Lemma 14 that the gradient of $\mathcal{L}(\boldsymbol{x})$ with respect to player $l$'s strategy $x_l$ is

$$\nabla_{x_l}\mathcal{L}(\boldsymbol{x}) = 2\sum_k \eta_k B_{kl}^\top \Pi_{T\Delta}(\nabla_{x_k}^{k\tau}) \tag{50}$$

where $B_{ll} = -\tau[I - \frac{1}{m_l}\mathbf{1}\mathbf{1}^\top]\mathtt{diag}\big(\frac{1}{x_l}\big)$ and $B_{kl} = [I - \frac{1}{m_k}\mathbf{1}\mathbf{1}^\top]H_{kl}^k$ for $k \neq l$.

As noted before in the derivation of Lemma 10, for payoffs in $[0, 1]$, the entries in $\nabla_{x_k}^{k\tau} = \nabla_{x_k}^k - \tau\ln(x_k)$ are bounded within $[0, \tau\ln(\frac{1}{x_{\min}}) + 1]$ with a range $\tau\ln(\frac{1}{x_{\min}}) + 1$. Similarly, the entries in $-\tau\mathtt{diag}\big(\frac{1}{x_l}\big)$ are bounded within $[-\tau\frac{1}{x_{\min}}, -\tau]$ with a range of $\tau(\frac{1}{x_{\min}} - 1)$.

The infinity norm of the gradient can then be bounded as

$$||\nabla_{\boldsymbol{x}}\mathcal{L}^{\tau}(\boldsymbol{x})||_{\infty} = \max_{l} ||\nabla_{x_l}\mathcal{L}(\boldsymbol{x})||_{\infty} \tag{51a}$$

$$= \max_{l} ||2\sum_{k} \eta_k B_{kl}^{\top}\Pi_{T\Delta}(\nabla_{x_k}^{k\tau})||_{\infty} \tag{51b}$$

$$\leq 2\sum_{k} \eta_k \max_{l} ||B_{kl}^{\top}\Pi_{T\Delta}(\nabla_{x_k}^{k\tau})||_{\infty} \tag{51c}$$

$$\leq \frac{1}{2}\sum_{k \neq l^*} \eta_k m_k\left(\tau\ln\left(\frac{1}{x_{\min}}\right) + 1\right) + \frac{1}{2}\eta_{l^*}m_{l^*}\tau\left(\frac{1}{x_{\min}} - 1\right)\left(\tau\ln\left(\frac{1}{x_{\min}}\right) + 1\right) \tag{51d}$$

$$= \frac{1}{2}\left(\tau\ln\left(\frac{1}{x_{\min}}\right) + 1\right)\left[\eta_{l^*}m_{l^*}\tau\left(\frac{1}{x_{\min}} - 1\right) + \sum_{k \neq l^*}\eta_k m_k\right] \tag{51e}$$

$$\leq \frac{1}{2}(\max_{k}\eta_k)\left(\tau\ln\left(\frac{1}{x_{\min}}\right) + 1\right)\left[\tau m_{l^*}\left(\frac{1}{x_{\min}} - 1\right) + \sum_{k \neq l^*}m_k\right] \tag{51f}$$

$$\leq \frac{1}{2}(\max_{k}\eta_k)\left(\tau\ln\left(\frac{1}{x_{\min}}\right) + 1\right)\left[\tau m^*\left(\frac{1}{x_{\min}} - 1\right) + n\bar{m}\right] \tag{51g}$$

where the second inequality follows from Lemma 9.

$\square$

**Corollary 2.** *If $\tau$ is set according to Lemma 11 as $\tau = \ln(1/p)^{-1}$ and payoffs are in $[0, 1]$, then the infinity norm of the gradient is bounded as*

$$||\nabla_{\boldsymbol{x}}\mathcal{L}^{\tau}(\boldsymbol{x})||_{\infty} \leq \frac{1}{2}(\max_{k}\eta_k)\left[\frac{\ln(m^*)}{\ln(1/p)} + 2\right]\left[\frac{m^{*2}}{p\ln(1/p)} + n\bar{m}\right] = \frac{1}{2}(\max_{k}\eta_k)\hat{L} \tag{52}$$

*where $m^* = \max_k m_k$ and $\hat{L}$ is defined implicitly for convenience in other derivations.*

*Proof.* Starting with Lemma 15 and applying Lemma 11 (i.e., $\tau = \ln(1/p)^{-1}$ and $x_{\min} = \frac{p}{m^*}$ with payoffs in $[0, 1]$ and where $m^* = \max_k m_k$), we find

$$||\nabla_{\boldsymbol{x}}\mathcal{L}^{\tau}(\boldsymbol{x})||_{\infty} \leq \frac{1}{2}(\max_{k}\eta_k)\left(\tau\ln\left(\frac{1}{x_{\min}}\right) + 1\right)\left[\tau m^*\left(\frac{1}{x_{\min}} - 1\right) + n\bar{m}\right] \tag{53a}$$

$$= \frac{1}{2}(\max_{k}\eta_k)\left[\frac{\ln(m^*/p)}{\ln(1/p)} + 1\right]\left[\frac{m^*}{\ln(1/p)}\left(\frac{m^*}{p} - 1\right) + n\bar{m}\right] \tag{53b}$$

$$\leq \frac{1}{2}(\max_{k}\eta_k)\left[\frac{\ln(m^*)}{\ln(1/p)} + 2\right]\left[\frac{m^{*2}}{p\ln(1/p)} + n\bar{m}\right]. \tag{53c}$$

As $p \to 0^+$, the norm of the gradient blows up because the gradient of Shannon entropy blows up for small probabilities. As $p \to 1$, the norm of the gradient blows up because we require infinite temperature $\tau$ to guarantee all QREs are nearly uniform; recall $\tau$ is the regularization coefficient on the entropy bonus terms which means our modified utilities blow up for large $\tau$. $\square$

### A.5 HESSIAN OF LOSS

We will now derive the Hessian of our loss. This will be useful in establishing properties about global minima that enable the application of tailored minimization algorithms. Let $D_z[f(z)]$ denote the differential operator applied to (possibly multivalued) function $f$ with respect to $z$. For example, $D_{x_q}[H_{lk}^k] = D_{x_q}[x_q T_{qlk}^k] = T_{qlk}^k$ where $T_{qlk}^k$ is player $k$'s payoff tensor according to the three-way approximation between players $k$, $l$, and $q$ to the game at $\boldsymbol{x}$.

**Lemma 16.** *The Hessian of $\mathcal{L}^{\tau}(\boldsymbol{x})$ can be written*

$$Hess(\mathcal{L}^{\tau}) = 2\tilde{B}^{\top}\tilde{B} + T\Pi_{T\Delta}(\tilde{\nabla}^{\tau}) \tag{54}$$

*where $\tilde{B}_{kl} = \sqrt{\eta_k}B_{kl}$, $\Pi_{T\Delta}(\tilde{\nabla}^{\tau}) = [\eta_1\Pi_{T\Delta}(\nabla_{x_1}^{1\tau}), \ldots, \eta_n\Pi_{T\Delta}(\nabla_{x_n}^{n\tau})]$, and we augment $T$ (the 3-player tensor approximation to the game, $T_{lqk}^k$) so that $T_{lll}^l = \tau \text{diag3}\left(\frac{1}{x_l^2}\right)$ and otherwise 0.*

*Proof.* Recall the gradient of our proposed loss:

$$\nabla_{x_l}\mathcal{L}(\boldsymbol{x}) = 2\sum_k \eta_k B_{kl}^\top \Pi_{T\Delta}(\nabla_{x_k}^{k\tau}) \tag{55}$$

where $B_{ll} = -\tau[I - \frac{1}{m_l}\mathbf{1}\mathbf{1}^\top]\texttt{diag}\big(\frac{1}{x_l}\big)$ and $B_{kl} = [I - \frac{1}{m_k}\mathbf{1}\mathbf{1}^\top]H_{kl}^k$ for $k \neq l$.

Consider the following Jacobians, which will play an auxiliary role in our derivation of the Hessian:

$$D_l[B_{ll}] = \tau[I - \frac{1}{m_l}\mathbf{1}\mathbf{1}^\top]\texttt{diag3}\big(\frac{1}{x_l^2}\big) \tag{56a}$$

$$D_q[B_{ll}] = \mathbf{0} \tag{56b}$$

$$D_l[B_{kl}] = \mathbf{0} \tag{56c}$$

$$D_q[B_{kl}] = [I - \frac{1}{m_k}\mathbf{1}\mathbf{1}^\top]T_{klq}^k \tag{56d}$$

$$D_k[\Pi_{T\Delta}(\nabla_{x_k}^{k\tau})] = [I - \frac{1}{m_k}\mathbf{1}\mathbf{1}^\top]D_k[\nabla_{x_k}^{k\tau}] \tag{56e}$$

$$= [I - \frac{1}{m_k}\mathbf{1}\mathbf{1}^\top]D_k[\nabla_{x_k}^k - \tau\ln(x_k)] \tag{56f}$$

$$= [I - \frac{1}{m_k}\mathbf{1}\mathbf{1}^\top][-\tau\texttt{diag}\big(\frac{1}{x_k}\big)] \tag{56g}$$

$$= B_{kk} \tag{56h}$$

$$D_l[\Pi_{T\Delta}(\nabla_{x_k}^{k\tau})] = [I - \frac{1}{m_k}\mathbf{1}\mathbf{1}^\top]D_l[\nabla_{x_k}^{k\tau}] \tag{56i}$$

$$= [I - \frac{1}{m_k}\mathbf{1}\mathbf{1}^\top]D_l[\nabla_{x_k}^k - \tau\ln(x_k)] \tag{56j}$$

$$= [I - \frac{1}{m_k}\mathbf{1}\mathbf{1}^\top][H_{kl}^k] \tag{56k}$$

$$= B_{kl}. \tag{56l}$$

We can derive the diagonal blocks of the Hessian as

$$D_{ll}[\mathcal{L}(\boldsymbol{x})] = D_l[\nabla_{x_l}\mathcal{L}(\boldsymbol{x})] \tag{57a}$$

$$= 2D_l[\sum_k \eta_k B_{kl}^\top \Pi_{T\Delta}(\nabla_{x_k}^{k\tau})] \tag{57b}$$

$$= 2\Big[\eta_l D_l\big[B_{ll}^\top \Pi_{T\Delta}(\nabla_{x_l}^{l\tau})\big] + \sum_{k \neq l} \eta_k D_l\big[B_{kl}^\top \Pi_{T\Delta}(\nabla_{x_k}^{k\tau})\big]\Big] \tag{57c}$$

$$= 2\Big[\eta_l \big[D_l[B_{ll}]^\top \Pi_{T\Delta}(\nabla_{x_l}^{l\tau}) + B_{ll}^\top D_l[\Pi_{T\Delta}(\nabla_{x_l}^{l\tau})]\big] \tag{57d}$$

$$+ \sum_{k \neq l} \eta_k \big[D_l[B_{kl}]^\top \Pi_{T\Delta}(\nabla_{x_k}^{k\tau}) + B_{kl}^\top D_l[\Pi_{T\Delta}(\nabla_{x_k}^{k\tau})]\big]\Big] \tag{57e}$$

$$= 2\Big[\eta_l \big[\tau\texttt{diag3}\big(\frac{1}{x_l^2}\big)[I - \frac{1}{m_l}\mathbf{1}\mathbf{1}^\top]\Pi_{T\Delta}(\nabla_{x_l}^{l\tau}) + B_{ll}^\top B_{ll}\big] + \sum_{k \neq l} \eta_k B_{kl}^\top B_{kl}\Big] \tag{57f}$$

$$= 2\Big[\tau\eta_l\texttt{diag}\big([\frac{1}{x_l^2}] \odot \Pi_{T\Delta}(\nabla_{x_l}^{l\tau})\big) + \sum_k \eta_k B_{kl}^\top B_{kl}\Big] \tag{57g}$$

and the off-diagonal blocks as

$$D_{lq}[\mathcal{L}(\boldsymbol{x})] = D_q[\nabla_{x_l}\mathcal{L}(\boldsymbol{x})] \tag{58a}$$

$$= 2D_q\left[\sum_k \eta_k B_{kl}^\top \Pi_{T\Delta}(\nabla_{x_k}^{k\tau})\right] \tag{58b}$$

$$= 2\left[\eta_l D_q\left[B_{ll}^\top \Pi_{T\Delta}(\nabla_{x_l}^{l\tau})\right] + \sum_{k\neq l} \eta_k D_q\left[B_{kl}^\top \Pi_{T\Delta}(\nabla_{x_k}^{k\tau})\right]\right] \tag{58c}$$

$$= 2\left[\eta_l\left[D_q[\overline{B_{ll}}]^\top \Pi_{T\Delta}(\nabla_{x_l}^{l\tau}) + B_{ll}^\top D_q[\Pi_{T\Delta}(\nabla_{x_l}^{l\tau})]\right]\right.$$

$$\left. + \sum_{k\neq l} \eta_k\left[D_q[B_{kl}]^\top \Pi_{T\Delta}(\nabla_{x_k}^{k\tau}) + B_{kl}^\top D_q[\Pi_{T\Delta}(\nabla_{x_k}^{k\tau})]\right]\right] \tag{58e}$$

$$= 2\left[\eta_l B_{ll}^\top B_{lq} + \sum_{k\neq l} \eta_k\left[T_{lqk}^k[I - \frac{1}{m_k}\mathbf{1}\mathbf{1}^\top]\Pi_{T\Delta}(\nabla_{x_k}^{k\tau}) + B_{kl}^\top B_{kq}\right]\right] \tag{58f}$$

$$= 2\left[\sum_k \eta_k B_{kl}^\top B_{kq} + \sum_{k\neq l} \eta_k T_{lqk}^k \Pi_{T\Delta}(\nabla_{x_k}^{k\tau})\right]. \tag{58g}$$

Therefore, the Hessian can be written concisely as

$$2\left[\tilde{B}^\top \tilde{B} + T\Pi_{T\Delta}(\tilde{\nabla}^\tau)\right] \tag{59}$$

where $\tilde{B}_{kl} = \sqrt{\eta_k}B_{kl}$, $\Pi_{T\Delta}(\tilde{\nabla}^\tau) = [\eta_1\Pi_{T\Delta}(\nabla_{x_1}^{1\tau}), \ldots, \eta_n\Pi_{T\Delta}(\nabla_{x_n}^{n\tau})]$, and we augment $T$ (the 3-player tensor approximation to the game, $T_{lqk}^k$) so that $T_{lll}^l = \tau\mathtt{diag3}\left(\frac{1}{x_l^2}\right)$ and otherwise 0.

$\square$

# B GLOBAL CONVERGENCE GUARANTEES

In this section, we analyze the application of optimization techniques such as bandit algorithms and gradient descent to our loss function.

## B.1 MAPS FROM HYPERCUBE TO SIMPLEX PRODUCT

In this subsection, we derive properties of a map $s$ from the unit-hypercube to the simplex product. This map is necessary to to adapt our proposed loss $\mathcal{L}^\tau$ to the commonly assumed setting in the $\mathcal{X}$-armed bandit literature (Bubeck et al., 2011) where the feasible set is a hypercube. We derive relevant properties of two such maps: the `softmax` and a mapping that interprets dimensions of the hypercube as angles on a unit-sphere that are then $\ell_1$-normalized.

**Lemma 17.** *Let $f(x) = -\mathcal{L}(s(x))$. Then $||\nabla f(x)||_\infty \leq ||J(s(x))^\top||_\infty ||\nabla \mathcal{L}(s(x))||_\infty$.*

*Proof.*

$$||\nabla f(x)||_\infty = ||J(s(x))^\top \nabla \mathcal{L}(s(x))||_\infty \leq ||J(s(x))^\top||_\infty ||\nabla \mathcal{L}(s(x))||_\infty. \tag{60}$$

$\square$

**Lemma 18.** *The $\infty$-norm of the Jacobian-transpose of a transformation $s(x)$ applied elementwise to a product space is bounded by the $\infty$-norm of the Jacobian-transpose of a single transformation from that product space, i.e., $||J(s(\boldsymbol{x}))^\top||_\infty \leq \max_{x_i \in \mathcal{X}_i} ||J(s(x_i))^\top||_\infty$ for any $i$.*

*Proof.* Let $\boldsymbol{x} \in \mathcal{X} = \prod_{i=1}^n \mathcal{X}_i$, $\mathcal{Z} = \prod_{i=1}^n \mathcal{Z}_i$ and $S : \mathcal{X} \to \mathcal{Z} = [s(x_1); \cdots ; s(x_n)]^\top$ where ; denotes column-wise stacking, $x_i \in \mathcal{X}_i$. Also, $\mathcal{X}_i = \mathcal{X}_j$ and $\mathcal{Z}_i = \mathcal{Z}_j$ for all $i$ and $j$. Then the Jacobian of $S(\boldsymbol{x})$ is

$$J(S(\boldsymbol{x}))^\top = \begin{bmatrix} J(s(x_1))^\top & 0 \ldots & 0 \\ 0 & J(s(x_2))^\top \ldots & 0 \\ 0 & 0 \ddots & 0 \\ 0 & 0 \ldots & J(s(x_n))^\top \end{bmatrix}. \tag{61}$$

The $\infty$-norm of this matrix is the max 1-norm of any row. This matrix is diagonal, therefore, the $\infty$-norm of each elementwise Jacobian-transpose represents the max 1-norm of the rows spanned by its block. Given that the domains, ranges, and transformations $s$ for all blocks are the same, their $\infty$-norms are also the same. The max $\infty$ over the blocks is then equal to the $\infty$-norm of any individual $J(s(x_i))^\top$. $\square$

### B.1.1 HESSIAN OF BANDIT REWARD FUNCTION

**Lemma 19.** *Let $s(x)$ be a function that maps the unit hypercube to the simplex product (mixed strategy space). Then the objective function $f(x) = -\mathcal{L}(s(x))$. The Hessian of $-f(x)$ at an optimum $x^*$ in direction $\Delta$ is $\Delta x^\top [Ds(x)^\top H_\mathcal{L}(x) Ds(x)]\big|_{x^*} \Delta x$ where $H_\mathcal{L}$ is the Hessian of $\mathcal{L}$ and $Ds(x)$ is the Jacobian of $s(x)$.*

*Proof.*

$$(D^2(\mathcal{L} \circ s)(x^*))(\Delta x, \Delta x) = \Delta x^\top \Big[ \sum_i \overbrace{\partial_i \mathcal{L}(s(x))}^{=0 \text{ at } x=x^*} D^2 h_i(x)\Big]\Big|_{x^*} \Delta x \tag{62a}$$

$$+ \Delta x^\top [Ds(x)^\top H_\mathcal{L}(x) Ds(x)]\big|_{x^*} \Delta x$$

$$= \Delta x^\top [Ds(x)^\top H_\mathcal{L}(x) Ds(x)]\big|_{x^*} \Delta x. \tag{62b}$$

$\square$

**Lemma 20.** *Let $s(x) : \mathcal{X} \to \prod_k \Delta^{m_k-1}$ be an injective function, i.e., $x \neq y \implies s(x) \neq s(y)$. Also let $J = J(s(x))$ be the Jacobian of $s$ with respect to $x$ and $\Delta x$ be a nonzero vector in the tangent space of $\mathcal{X}$. Then*

$$J\Delta x \neq \mathbf{0}. \tag{63}$$

*Proof.* Recall that the $ij$th entry of the Jacobian represents $\frac{\partial s_i}{\partial x_j}$ so that the $i$th entry of $J\Delta x$ is

$$[J\Delta x]_i = \sum_j \frac{\partial s_i}{\partial x_j} \Delta x_j = ds_i. \tag{64}$$

Assume $J\Delta x = \mathbf{0}$. This would imply a change in $x \in \mathcal{X}$ results in no change in $s$ ($ds = \mathbf{0}$), contradicting the fact that $s$ is injective. Therefore, we must conclude the claim that $J\Delta x \neq \mathbf{0}$. $\square$

**Lemma 21.** *Let $J = J(s(x))$ be the Jacobian of any composition of transformations $s = s_t \circ \ldots s_1$ where $s_t(z) = [z_i / \sum_j z_j]_i$. Then $J\Delta x$ lies in the tangent space of the simplex.*

*Proof.* We aim to show $\mathbf{1}^\top J\Delta x = \mathbf{0}$ for any $\Delta x$ and $x$. By chain rule, the Jacobian of $s$ is $J = J(s) = \prod_{t'=t}^{t'=1} J(s'_t)$. Therefore, $\mathbf{1}^\top J\Delta x = \mathbf{1}^\top (\prod_{t'=t}^{t'=1} J(s'_t))\Delta x$. Consider the first product:

$$\mathbf{1}^\top J(s_t) = \mathbf{0} \tag{65}$$

by Lemma 24. Therefore $\mathbf{1}^\top J\Delta x = \mathbf{1}^\top J(s_t)(\prod_{t'=t-1}^{t'=1} J(s'_t))\Delta x = \mathbf{0}^\top (\prod_{t'=t-1}^{t'=1} J(s'_t))\Delta x = 0$. This implies $J\Delta x$ is orthogonal to $\mathbf{1}$ for any $x \in \mathcal{X}$ and $\Delta x$, therefore $J\Delta x$ lies in the tangent space of the simplex for any $x \in \mathcal{X}$ and $\Delta x$. $\square$

### B.1.2 SOFTMAX MAP

Let $s : [0,1]^{d-1} \to \Delta^{d-1} \in \mathbb{R}^d$ be the `softmax` function. See (Gao and Pavel, 2017) for an analysis of many of its properties and in the context of game theory. Note that $s$ maps a $(d-1)$ dimensional variable to a $d$ dimensional distribution. This can be practically handled by always appending a 0 to the $(d-1)$-dimensional input prior to applying the standard `softmax`. We perform our analysis below in terms of the standard `softmax`, but note the norms we derive apply to our modified (invertible) `softmax`.

Standard:

$$s(x) = \frac{1}{\sum_{j=1}^d e^{x_j}} \left[ e^{x_1}, \ldots e^{x_d} \right]. \tag{66}$$

Modified:

$$s(x) = \frac{1}{1 + \sum_{j=1}^{d-1} e^{x_j}} \left[ e^{x_1}, \ldots e^{x_{d-1}}, 1 \right]. \tag{67}$$

**Lemma 22.** *Let $J$ be the Jacobian of the softmax operator. Then $||J||_\infty \leq 2$ and $||J^\top||_\infty \leq 2$.*

*Proof.* Let $S_i$ represent the $i$th entry of $S = \mathtt{softmax}(z)$ for any $z \in \mathbb{R}^m$. Then the 1-norm of row $i$ is upper bounded as

$$D_j S_i = S_i(\delta_{ij} - S_j) \tag{68a}$$

$$\implies \sum_j |D_j S_i| = \sum_j |S_i(\delta_{ij} - S_j)| \tag{68b}$$

$$\leq \sum_j |\delta_{ij} S_i| + |S_i S_j| \tag{68c}$$

$$= S_i + \sum_j S_i S_j \tag{68d}$$

$$= S_i + S_i \sum_j S_j \tag{68e}$$

$$= 2S_i \tag{68f}$$

$$\leq 2 \; \forall i. \tag{68g}$$

Also, the 1-norm of column $j$ is upper bounded similarly as

$$
\text{(69a)}
$$

$$
\sum_i |D_j S_i| = \sum_i |S_i(\delta_{ij} - S_j)| \tag{69b}
$$

$$
\leq \sum_i |\delta_{ij} S_i| + |S_i S_j| \tag{69c}
$$

$$
= S_j + \sum_i S_i S_j \tag{69d}
$$

$$
= S_i + S_j \sum_i S_i \tag{69e}
$$

$$
= 2S_j \tag{69f}
$$

$$
\leq 2 \ \forall j. \tag{69g}
$$

The $\infty$-norm of a matrix is the maximum 1-norm of any row. Therefore, $||J||_\infty$ and $||J^\top||_\infty$ are both upper bounded by 2. $\qquad\square$

**Corollary 3.** *Let $J$ be the Jacobian of the softmax operator. Then $||J||_2 \leq 2$ and $||J^\top||_2 \leq 2$.*

*Proof.* The Gershgorin circle theorem states that every eigenvalue of $J$ lies within one of the discs centered at the diagonal of $J$ with radius equal to the 1-norm of the row (excluding the diagonal term). The Jacobian of the softmax operator contains only nonnegative diagonal terms, therefore, the 1-norm of any entire row also represents the maximum magnitude of any eigenvalue allowed by any Gershgorin disc. In addition, $J$ is symmetric and therefore, the magnitude of its eigenvalues are equivalent to its singular values. Therefore by Lemma 22, $||J||_2 \leq 2$. $\qquad\square$

### B.1.3 SPHERICAL MAP

For spherical coordinates, let $s(x) = n(l(c(x)))$ where $c(x) = \pi/2x$, $l(\psi)$ maps angles to the unit sphere, and $n(z) = [z_i/\sum_j z_j]_i$.

**Definition 2.** *Define $l(\psi)$ as the transformation to the unit-sphere using spherical coordinates:*

$$
l_1(\psi) = \cos(\psi_1) \tag{70a}
$$
$$
l_2(\psi) = \sin(\psi_1)\cos(\psi_2) \tag{70b}
$$
$$
l_3(\psi) = \sin(\psi_1)\sin(\psi_2)\cos(\psi_3) \tag{70c}
$$
$$
\vdots = \vdots \tag{70d}
$$
$$
l_{m-1}(\psi) = \sin(\psi_1)\sin(\psi_2)\ldots\cos(\psi_{m-1}) \tag{70e}
$$
$$
l_m(\psi) = \sin(\psi_1)\sin(\psi_2)\ldots\sin(\psi_{m-1}). \tag{70f}
$$

**Lemma 23.** *Let $J$ be the Jacobian of the transformation to the unit-sphere using spherical coordinates, i.e. $z = l(\psi)$ where $||l||^2 = 1$ and $\psi_i \in [0, \frac{\pi}{2}]$ represents an angle for each $i$. Then $||J||_F \leq \sqrt{m}$.*

*Proof.* The Jacobian of the transformation is

$$
J(l) =
\begin{bmatrix}
-\sin(\psi_1) & 0 & \cdots & 0 \\
\cos(\psi_1)\cos(\psi_2) & -\sin(\psi_1)\sin(\psi_2) & \cdots & 0 \\
\vdots & \vdots & \ddots & \vdots \\
\cos(\psi_1)\sin(\psi_2)\ldots\cos(\psi_{m-1}) & \cdots & \cdots & -\sin(\psi_1)\ldots\sin(\psi_{m-2})\sin(\psi_{m-1}) \\
\cos(\psi_1)\sin(\psi_2)\ldots\sin(\psi_{m-1}) & \cdots & \cdots & \sin(\psi_1)\ldots\sin(\psi_{m-2})\cos(\psi_{m-1})
\end{bmatrix}
\tag{71}
$$

and it square is

$$
J(l) = \begin{bmatrix}
t_1 & 0 & \cdots & 0 \\
\cos(\psi_1)^2 \cos(\psi_2)^2 & \sin(\psi_1)^2 t_2 & \cdots & 0 \\
\vdots & \vdots & \ddots & \vdots \\
\cos(\psi_1)^2 \sin(\psi_2)^2 \ldots \cos(\psi_{m-1})^2 & \cdots & \cdots & \sin(\psi_1)^2 \ldots \sin(\psi_{m-2})^2 t_{m-1} \\
\cos(\psi_1)^2 \sin(\psi_2)^2 \ldots \sin(\psi_{m-1})^2 & \cdots & \cdots & \sin(\psi_1)^2 \ldots \sin(\psi_{m-2})^2 t_m
\end{bmatrix}
\tag{72}
$$

where

$$
\delta_{im} = 1 \text{ if } i = m, 0 \text{ else} \tag{73a}
$$
$$
t_i = \delta_{im} \cos^2(\psi_{i-1}) + (1 - \delta_{im}) \sin^2(\psi_i) \leq 1. \tag{73b}
$$

To compute the Frobenius norm, we will need the sum of the squares of all entries. We will consider the sum of each row individually using the following auxiliary variable $R_{i,k \leq i}$ where $\sum_j J_{ij}^2 = R_{i,1}$ and apply a recursive inequality.

$$
\tag{74a}
$$

$$
R_{i,k \leq i} = \sum_{k'=k}^{i-1} \cos^2(\psi_{k'}) \Big[ \prod_{l=k, l \neq k'}^{i-1} \sin^2(\psi_l) \Big] \cos^2(\psi_i) + t_i \prod_{l=k}^{i-1} \sin^2(\psi_l) \tag{74b}
$$

$$
= \cos^2(\psi_k) \underbrace{\Big[ \prod_{l=k+1}^{i-1} \sin^2(\psi_l) \Big] \cos^2(\psi_i)}_{\leq 1} \tag{74c}
$$

$$
+ \sin^2(\psi_k) \sum_{k'=k+1}^{i-1} \cos^2(\psi_{k'}) \Big[ \prod_{l=k+1, l \neq k'}^{i-1} \sin^2(\psi_l) \Big] \cos^2(\psi_i) \tag{74d}
$$

$$
+ \sin^2(\psi_k) t_i \prod_{l=k+1}^{i-1} \sin^2(\psi_l) \tag{74e}
$$

$$
\leq \cos^2(\psi_k) \tag{74f}
$$

$$
+ \sin^2(\psi_k) \Big( \sum_{k'=k+1}^{i-1} \cos^2(\psi_{k'}) \Big[ \prod_{l=k+1, l \neq k'}^{i-1} \sin^2(\psi_l) \Big] \cos^2(\psi_i) + t_i \prod_{l=k+1}^{i-1} \sin^2(\psi_l) \Big) \tag{74g}
$$

$$
= \cos^2(\psi_k) + \sin^2(\psi_k) R_{i,k+1}. \tag{74h}
$$

Note then that $R_{i,k+1} \leq 1 \implies R_{i,k} \leq 1$. We know $R_{i,i} = t_i \leq 1$, therefore, $R_{i,1} \leq 1$ by applying the inequality recursively. Finally, $\sum_j J_{ij}^2 = R_{i,1} \leq 1$ implies the claim $||J||_F^2 = \sum_i R_{i,1} \leq m$. $\quad\square$

**Lemma 24.** *Let $J$ be the Jacobian of $n(z) = z/Z$ where $Z = \sum_k z_k$. Then $\mathbf{1}^\top J = \mathbf{0}^\top$.*

*Proof.* The $ij$th entry of the Jacobian of $n(z)$ is

$$
J(n)_{ij} = \frac{1}{Z^2}(-z_i + \delta_{ij} Z). \tag{75}
$$

Therefore $[\mathbf{1}^\top J]_j = \sum_i J(n)_{ij} = \frac{1}{Z^2}(-Z + Z) = 0$ where $z$ is a point on the unit-sphere in the positive orthant. $\quad\square$

## B.2 NEAR OPTIMALITY & ZOOMING DIMENSION

In this subsection, we derive bounds on the near-optimality dimension and zooming dimension of the associated bandit optimization problems we use to derive global convergence guarantees to Nash equilibria. This is written in terms of maximizing a function $f$ rather than minimizing a loss to better match the bandit literature (Bubeck et al., 2011; Valko et al., 2013; Feng et al., 2022).

**Assumption 1.** *Locally around each interior $x^* \in \mathcal{X}^*$, $-f(x)$ is lower bounded by $-f(x^*) + \sigma_- ||x - x^*||^{\alpha_{hi}}$. In other words, for all $f(x) \geq f(x^*) - \eta$:*

$$f(x^*) - f(x) \geq \sigma_- ||x - x^*||^{\alpha_{hi}} \tag{76}$$

*where we have left the precise norm unspecified for generality. Let $\ell(x, x^*) = \sigma_+ ||x - x^*||^{\alpha_{lo}}$.*

**Definition 3.** $\mathcal{X}_\epsilon \stackrel{\text{def}}{=} \{x \in \mathcal{X} \mid \exists x^* \in \mathcal{X}^* \ s.t. \ f(x) \geq f(x^*) - \epsilon\}$

**Definition 4.** $\mathcal{X}_\epsilon^{lower} \stackrel{\text{def}}{=} \{x \in \mathcal{X} \mid \exists x^* \in \mathcal{X}^* \ s.t. \ f(x^*) - \sigma_- ||x - x^*||^{\alpha_{hi}} \geq f(x^*) - \epsilon\}$

**Corollary 4.** $\mathcal{X}_\epsilon \subseteq \mathcal{X}_\epsilon^{lower}$.

*Proof.* By Assumption 1, $f(x^*) - \sigma_- ||x - x^*||^{\alpha_{hi}} \geq f(x)$. Therefore, any $x \in \mathcal{X}$ that satisfies the requirement for an element of $\mathcal{X}_\epsilon$, $f(x) \geq f(x^*) - \epsilon$, will also satisfy the requirement for an element of $\mathcal{X}_\epsilon^{lower}$. $\square$

**Definition 5** ($\psi$-near Optimality Dimension)**.** *The $\psi$-near optimality dimension is the smallest $d' > 0$ such that there exists $C > 0$ such that for any $\epsilon > 0$, the maximum number of disjoint $\ell$-balls of radius $\psi\epsilon$ and center in $\mathcal{X}_\epsilon$ is less than $C\epsilon^{-d'}$.*

**Definition 6** (Zooming Dimension)**.** *The zooming dimension is the smallest $d_z > 0$ such that there exists $C_z > 0$ such that for any $r > 0$, the maximum number of disjoint $\ell$-balls of radius $\frac{r}{2}$ and center in $\mathcal{X}_{16r}$ is less than $C_z r^{-d_z}$.*

**Corollary 5** (Zooming Dimension from $\psi$-near Optimality Dimension)**.** *The zooming dimension of $f : x \in [0,1]^d \to [-1, 1]$ under $\ell(x, y) = \sigma_+ ||x - y||^{\alpha_{lo}}$ is $d_z = d(\frac{\alpha_{hi} - \alpha_{lo}}{\alpha_{lo}\alpha_{hi}})$ with constant $C_z = 16^{-d'} C$ with $\psi = \frac{1}{32}$ ($\psi$ is needed to compute $C$).*

*Proof.* Mapping the definition of $\psi$-near optimality onto zooming dimension, we find $\psi\epsilon = r/2$ and $\epsilon = 16r$. Then we can infer $\psi = 1/32$; this is used to compute $C$ (see Theorem 1). Rewriting the bound from $\psi$-near optimality in terms of $r$, we find

$$N_\epsilon = N_{16r} \leq C(16r)^{-d'} = C16^{-d'} r^{-d'} = C_z r^{-d_z} \tag{77}$$

where $N_{16r}$ denotes the number of $\ell$-balls of radius $r/2$ with center in $\mathcal{X}_{16r}$, $d_z = d'$, $C_z = C16^{-d'} = C16^{-d_z}$. Therefore, this translation only effects the constant $C_z$, not the zooming dimension. $\square$

**Lemma 25** ($N_{\epsilon \leq \eta} \leq C_{\epsilon \leq \eta} \epsilon^{-d'}$)**.** *The maximum number of disjoint $\ell$-balls with radius $\psi\epsilon$ and center in $\mathcal{X}_{\epsilon \leq \eta}$, $N_{\epsilon \leq \eta}$, is upper bounded by $C_{\epsilon \leq \eta} \epsilon^{-d'}$ where $C_{\epsilon \leq \eta} = |\mathcal{X}^*| \left( \frac{\sigma_+}{\psi \sigma_-^{\alpha_{lo}/\alpha_{hi}}} \right)^{d/\alpha_{lo}}$ and $d' = d(\frac{\alpha_{hi} - \alpha_{lo}}{\alpha_{lo}\alpha_{hi}})$.*

*Proof.* The number of disjoint $\ell$-balls of radius $\psi\epsilon$ and center in $\mathcal{X}_{\epsilon \leq \eta}$ can be upper bounded as follows.

Rewrite $\mathcal{X}_\epsilon^{lower}$ by rearranging terms as

$$\mathcal{X}_\epsilon^{lower} = \{x \in \mathcal{X} | \exists x^* \in \mathcal{X}^* \ s.t. \ ||x - x^*|| \leq \left( \frac{\epsilon}{\sigma_-} \right)^{1/\alpha_{hi}} \stackrel{\text{def}}{=} r_\epsilon \} \tag{78}$$

and recall that from Corollary 4 that $\mathcal{X}_\epsilon \subseteq \mathcal{X}_\epsilon^{lower}$. Furthermore, an $\ell$-ball of radius $\psi\epsilon$ implies

$$\ell(x, y) = \sigma_+ ||x - y||^{\alpha_{lo}} \leq \psi\epsilon \implies ||x - y|| \leq \left( \frac{\psi\epsilon}{\sigma_+} \right)^{1/\alpha_{lo}} \stackrel{\text{def}}{=} r_\ell. \tag{79}$$

The number of disjoint $\ell$-balls that can pack into a set $\mathcal{X}_\epsilon$, $N_{\epsilon \leq \eta}$, is upper bounded by the ratio of the volumes of the two sets:

$$N_{\epsilon \leq \eta} \leq \frac{Vol(\mathcal{X}_\epsilon)}{Vol(\mathcal{B}_\ell)} \tag{80a}$$

$$\leq \frac{Vol(\mathcal{X}_\epsilon^{lower})}{Vol(\mathcal{B}_\ell)} \tag{80b}$$

$$= \frac{|\mathcal{X}^*| S_d r_\epsilon^d}{S_d r_\ell^d} \tag{80c}$$

$$\leq \frac{|\mathcal{X}^*| \left(\frac{\epsilon}{\sigma_-}\right)^{d/\alpha_{hi}}}{\left(\frac{\psi\epsilon}{\sigma_+}\right)^{d/\alpha_{lo}}} \tag{80d}$$

$$= |\mathcal{X}^*| \left(\frac{\sigma_+^{1/\alpha_{lo}} \psi^{-1/\alpha_{lo}}}{\sigma_-^{1/\alpha_{hi}}}\right)^d \epsilon^{d(1/\alpha_{hi} - 1/\alpha_{lo})} \tag{80e}$$

$$= |\mathcal{X}^*| \left(\frac{\sigma_+}{\psi \sigma_-^{\alpha_{lo}/\alpha_{hi}}}\right)^{d/\alpha_{lo}} \epsilon^{-d\left(\frac{\alpha_{hi} - \alpha_{lo}}{\alpha_{lo} \alpha_{hi}}\right)} \tag{80f}$$

$$= C_{\epsilon \leq \eta} \epsilon^{-d'} \tag{80g}$$

where $C_{\epsilon \leq \eta} = |\mathcal{X}^*| \left(\frac{\sigma_+}{\psi \sigma_-^{\alpha_{lo}/\alpha_{hi}}}\right)^{d/\alpha_{lo}}$, $d' = d\left(\frac{\alpha_{hi} - \alpha_{lo}}{\alpha_{lo} \alpha_{hi}}\right)$, $|\mathcal{X}^*|$ is the number of distinct global optima, and $S_d$ is the volume constant for a $d$-sphere under the given norm $|| \cdot ||$. $\qquad \square$

Recall, these results apply when $f(x) \geq f(x^*) - \eta$, i.e., when $\epsilon \leq \eta$. Otherwise, we can upper bound the number of $\ell$-balls by considering the entire set $\mathcal{X}$ which has volume 1. First, we will bound the constant associated with the volume of a $d$-sphere.

**Lemma 26** ($N_{\epsilon \geq \eta} \leq C_{\epsilon \geq \eta}$). *The maximum number of disjoint $\ell$-balls with radius $\psi\epsilon$ and center in $\mathcal{X}_{\epsilon \geq \eta}$, $N_{\epsilon \geq \eta}$, is upper bounded by $C_{\epsilon \geq \eta}$ where $C_{\epsilon \geq \eta} = S_d^{-1} \left(\frac{\sigma_+}{\psi\eta}\right)^{d/\alpha_{lo}}$ and $S_d$ is the volume constant for a $d$-sphere under a given norm.*

*Proof.* We can upper bound the number of $\ell$-balls needed to pack the entire space by considering the smallest possible radius $\psi\eta$:

$$N_{\epsilon \geq \eta} \leq \frac{Vol(\mathcal{X})}{Vol(\mathcal{B}_\ell)} \tag{81a}$$

$$= \frac{1}{S_d r_\ell^d} \tag{81b}$$

$$\leq \frac{1}{S_d \left(\frac{\psi\eta}{\sigma_+}\right)^{d/\alpha_{lo}}} \tag{81c}$$

$$= S_d^{-1} \left(\frac{\sigma_+}{\psi\eta}\right)^{d/\alpha_{lo}} \tag{81d}$$

$$= C_{\epsilon \geq \eta} \tag{81e}$$

where $r_l$ was defined in equation (79). $\qquad \square$

**Theorem 1.** *The $\psi$-near optimality dimension of $f : x \in [0,1]^d \to [-1,1]$ under $\ell$ is $d' = d\left(\frac{\alpha_{hi} - \alpha_{lo}}{\alpha_{lo} \alpha_{hi}}\right)$ with constant*

$$C = \max \left\{ 1, S_d^{-1} \left(r_\eta^{\frac{\alpha_{hi}}{\alpha_{lo}}} \sigma_-^{-\left(\frac{\alpha_{hi} - \alpha_{lo}}{\alpha_{lo} \alpha_{hi}}\right)}\right)^{-d} \right\} \left(\frac{\sigma_+}{\psi \sigma_-^{\alpha_{lo}/\alpha_{hi}}}\right)^{d/\alpha_{lo}} \tag{82}$$

*where $S_d$ is the volume constant for a $d$-sphere under the same norm as $\ell$.*

*Proof.* First, let us define $r_\eta = \left(\frac{\eta}{\sigma_-}\right)^{1/\alpha_{hi}}$ as in equation (78) which implies $\eta = \sigma_- r_\eta^{\alpha_{hi}}$. Then apply Lemmas 25 ($N_{\epsilon \leq \eta} \leq C_{\epsilon \leq \eta} \epsilon^{-d'}$) and 26 ($N_{\epsilon \geq \eta} \leq C_{\epsilon \geq \eta}$) which bound the number of $\ell$-balls required to pack $\mathcal{X}_\epsilon$ when $\epsilon$ is less than and greater than $\eta$ respectively:

$$C_{\epsilon \leq \eta} = |\mathcal{X}^*| \left( \frac{\sigma_+}{\psi \sigma_-^{\alpha_{lo}/\alpha_{hi}}} \right)^{d/\alpha_{lo}} \tag{83}$$

$$d' = d \left( \frac{\alpha_{hi} - \alpha_{lo}}{\alpha_{lo}\alpha_{hi}} \right) \tag{84}$$

and

$$C_{\epsilon \geq \eta} = S_d^{-1} \left( \frac{\sigma_+}{\psi \eta} \right)^{d/\alpha_{lo}} \tag{85a}$$

$$= |\mathcal{X}^*|^{-1} S_d^{-1} \eta^{-d/\alpha_{lo}} \sigma_-^{d/\alpha_{hi}} |\mathcal{X}^*| \left( \frac{\sigma_+}{\psi \sigma_-^{\alpha_{lo}/\alpha_{hi}}} \right)^{d/\alpha_{lo}} \tag{85b}$$

$$= |\mathcal{X}^*|^{-1} S_d^{-1} \eta^{-d/\alpha_{lo}} \sigma_-^{d/\alpha_{hi}} C_{\epsilon \leq \eta} \tag{85c}$$

$$= |\mathcal{X}^*|^{-1} S_d^{-1} r_\eta^{-d\alpha_{hi}/\alpha_{lo}} \sigma_-^{-d/\alpha_{lo}} \sigma_-^{d/\alpha_{hi}} C_{\epsilon \leq \eta} \tag{85d}$$

$$= |\mathcal{X}^*|^{-1} S_d^{-1} r_\eta^{-d\frac{\alpha_{hi}}{\alpha_{lo}}} \sigma_-^{-d\left(\frac{\alpha_{hi}-\alpha_{lo}}{\alpha_{lo}\alpha_{hi}}\right)} C_{\epsilon \leq \eta} \tag{85e}$$

$$= |\mathcal{X}^*|^{-1} S_d^{-1} \left( r_\eta^{\frac{\alpha_{hi}}{\alpha_{lo}}} \sigma_-^{\left(\frac{\alpha_{hi}-\alpha_{lo}}{\alpha_{lo}\alpha_{hi}}\right)} \right)^{-d} C_{\epsilon \leq \eta} \tag{85f}$$

where $S_d$ is the volume constant for a $d$-sphere under the given norm. $S_d^{-1}$ has been upper bounded for the 2-norm in Lemma 35. For the $\infty$-norm, $S_d^{-1} = 2^{-d}$. We have written $C_{\epsilon \geq \eta}$ in terms of $C_{\epsilon \leq \eta}$ to clarify which is larger.

Therefore,

$$C = \max \left\{ 1, |\mathcal{X}^*|^{-1} S_d^{-1} \left( r_\eta^{\frac{\alpha_{hi}}{\alpha_{lo}}} \sigma_-^{\left(\frac{\alpha_{hi}-\alpha_{lo}}{\alpha_{lo}\alpha_{hi}}\right)} \right)^{-d} \right\} C_{\epsilon \leq \eta} \tag{86a}$$

$$= \max \left\{ 1, |\mathcal{X}^*|^{-1} S_d^{-1} \left( r_\eta^{\frac{\alpha_{hi}}{\alpha_{lo}}} \sigma_-^{\left(\frac{\alpha_{hi}-\alpha_{lo}}{\alpha_{lo}\alpha_{hi}}\right)} \right)^{-d} \right\} \left( \frac{\sigma_+}{\psi \sigma_-^{\alpha_{lo}/\alpha_{hi}}} \right)^{d/\alpha_{lo}}. \tag{86b}$$

Intuitively, if the radius for which the polynomial bounds hold ($r_\eta$) is large and the minimum curvature constant $\sigma_-$ is also large, then the bound $C_{\epsilon \leq \eta}$ holds for large deviations from optimality $\eta$. The number of $\eta$-radius $\ell$-balls required to cover the remaining space, $C_{\epsilon \geq \eta}$, will be comparatively small. $\qquad \square$

## B.3   D-BLIN

The regret bound for Doubling BLiN (Feng et al., 2022) was originally proved assuming a standard normal distribution, however, the authors state their proof can be easily adapted to any sub-Gaussian distribution, which includes bounded random variables. This matches our setting with bounded payoffs, so we repeat their analysis here for that setting.

In what follows, in an effort to match notation in (Feng et al., 2022), let $\mu(x) = -\mathcal{L}^\tau(s(x))$ denote the expected *reward* for evaluating $x \in [0,1]^d$; $\hat{\mu}(x)$ is an unbiased estimate of $\mu(x)$. Here, $s : [0,1]^{n(\bar{m}-1)} \to \prod_i \Delta^{m_i-1}$ is any function that maps from the unit hypercube to a product of simplices; we analyze two such maps in Appendix B.1. Also let $\mathcal{A}_m$ denote the collection of hypercubes to be investigated during the $m$th batch of arm pulls, $C \in \mathcal{A}_m$ denote a hypercube partition in $\mathcal{X}$, $n_m$ indicates the number of times each cube in $\mathcal{A}_m$ is played in batch $m$, and $B_{stop}$ denote the last batch.

**Definition 7** (Global Arm Accuracy). $\mathcal{E} \overset{\text{def}}{=} \left\{ |\mu(x) - \hat{\mu}_m(C)| \leq r_m + \sqrt{c_1 \frac{\ln T}{n_m}}, \ \forall 1 \leq m \leq B_{stop} - 1, \ \forall C \in \mathcal{A}_m, \ \forall x \in C \right\}.$

Define: $n_m = c_2 \frac{\ln T}{r_m^2} \implies r_m = \sqrt{c_2 \frac{\ln T}{n_m}}$.

**Definition 8** (Elimination Rule). *Eliminate* $C \in \mathcal{A}_m$ *if* $\hat{\mu}_m^{\max} - \hat{\mu}_m(C) \geq 2(1 + \sqrt{c_1/c_2})r_m = 2(\sqrt{c_2} + \sqrt{c_1})\sqrt{\frac{\ln T}{n_m}}$ *where* $\hat{\mu}_m^{\max} \stackrel{\text{def}}{=} \max_{C \in \mathcal{A}_m} \hat{\mu}_m(C)$.

**Lemma 27.** $Pr[\mathcal{E}] \geq 1 - 2T^{-2(c_1/c^2 - 1)}$.

*Proof.* Let $\hat{\mu}_m(C) = \frac{1}{n_m} \sum_{i=1}^{n_m} y_{C,i}$ be the unbiased estimate of $\mu(x \in C)$ using $n_m$ samples. Assume each $y_{C,i} \in [a,b]$ with $c = b - a$ and $\hat{\mu}(C) = \frac{1}{n_m} \sum_{i=1}^{n_m} y_{C,i}$. Applying a Hoeffding inequality gives

$$Pr\left[|\hat{\mu}(C) - \mathbb{E}[\hat{\mu}(C)]| \geq \sqrt{c_1 \frac{\ln T}{n_m}}\right] \leq 2e^{-2c_1 \ln T / c^2} \tag{87a}$$

$$= 2(e^{\ln T})^{-2c_1/c^2} \tag{87b}$$

$$= 2T^{-2c_1/c^2} \ \forall C. \tag{87c}$$

By Lipschitzness of $\mu$ we also have

$$|\mathbb{E}[\hat{\mu}(C)] - \mu(x)| \leq r_m, \ \forall x \in C. \tag{88}$$

Then consider

$$\sup_{x \in C} |\mu(x) - \hat{\mu}(C)| = \sup_{x \in C} |\mu(x) - \mathbb{E}[\hat{\mu}(C)] + \mathbb{E}[\hat{\mu}(C)] - \hat{\mu}(C)| \tag{89a}$$

$$\leq \sup_{x \in C} \left(|\mu(x) - \mathbb{E}[\hat{\mu}(C)]| + |\mathbb{E}[\hat{\mu}(C)] - \hat{\mu}(C)|\right) \tag{89b}$$

$$= \sup_{x \in C} |\mu(x) - \mathbb{E}[\hat{\mu}(C)]| + |\mathbb{E}[\hat{\mu}(C)] - \hat{\mu}(C)| \tag{89c}$$

$$\leq \sqrt{c_1 \frac{\ln T}{n_m}} + r_m \tag{89d}$$

with probability $1 - 2T^{-2c_1/c^2}$. The first inequality follows by triangle inequality and the second follows from equation (88) and considering the complement of equation (87c).

The complement of this result occurs with probability

$$Pr\left[\sup_{x \in C} |\mu(x) - \hat{\mu}(C)| \geq r_m + \sqrt{c_1 \frac{\ln T}{n_m}}\right] \leq 2T^{-2c_1/c^2}. \tag{90}$$

At least 1 arm is played in each cube $C \in \mathcal{A}_m$ for $1 \leq m \leq B_{stop} - 1$, therefore, $|\mathcal{A}_m| \leq T$ must be true given the exit condition of the algorithm. In addition, assume $B_{stop} \leq T$ ($B_{stop}$ will be defined such that this is true). Then a union bound over all $T^2$ events gives

$$Pr\left[\exists m \in [1, B_{stop} - 1], C \in \mathcal{A}_m \ s.t. \ \sup_{x \in C} |\mu(x) - \hat{\mu}(C)| \geq r_m + \sqrt{c_1 \frac{\ln T}{n_m}}\right] \tag{91a}$$

$$\leq \sum_{m=1}^{B_{stop}-1} \sum_{C \in \mathcal{A}_m} Pr\left[\sup_{x \in C} |\mu(x) - \hat{\mu}(C)| \geq r_m + \sqrt{c_1 \frac{\ln T}{n_m}}\right] \tag{91b}$$

$$\leq \sum_{m=1}^{B_{stop}-1} \sum_{C \in \mathcal{A}_m} 2T^{-2c_1/c^2} \tag{91c}$$

$$\leq 2T^{-2c_1/c^2} T^2. \tag{91d}$$

Taking the complement of this event and noting that $\sup_{x \in C} |\mu(x) - \hat{\mu}(C)| \leq r_m + \sqrt{c_1 \frac{\ln T}{n_m}} \implies |\mu(x) - \hat{\mu}(C)| \leq r_m + \sqrt{c_1 \frac{\ln T}{n_m}} \ \forall x \in C$ gives the desired result. $\qquad \square$

**Lemma 28** (Optimal Arm Survives). *Under event $\mathcal{E}$, the optimal arm $x^* = \arg\max \mu(x)$ is not eliminated after the first $B_{stop} - 1$ batches.*

*Proof.* Let $C_m^*$ denote the cube containing $x^*$ in $\mathcal{A}_m$. Under event $\mathcal{E}$, for any cube $C \in \mathcal{A}_m$ and $x \in C$, the following relation shows that $C_m^*$ avoids the elimination rule in round $m$:

$$\hat{\mu}(C) - \hat{\mu}(C_m^*) \leq \left( \mu(x) + r_m + \sqrt{c_1 \frac{\ln T}{n_m}} \right) + \left( -\mu(x^*) + r_m + \sqrt{c_1 \frac{\ln T}{n_m}} \right) \tag{92a}$$

$$= \underbrace{(\mu(x) - \mu(x^*))}_{\leq 0} + 2r_m + 2\sqrt{c_1 \frac{\ln T}{n_m}} \tag{92b}$$

$$\leq 2\sqrt{c_2 \frac{\ln T}{n_m}} + 2\sqrt{c_1 \frac{\ln T}{n_m}} \tag{92c}$$

$$= 2(\sqrt{c_1} + \sqrt{c_2})\sqrt{\frac{\ln T}{n_m}} \tag{92d}$$

where the first inequality follows from applying Lemma 27 to upper bound $\hat{\mu}(C)$ and $\hat{\mu}(C_m^*)$ individually. The remaining steps use the optimality of $x^*$, the definition of $r_m$, and the elimination rule. $\qquad\square$

**Lemma 29.** *Under event $\mathcal{E}$, for any $1 \leq m \leq B_{stop}$, any $C \in A_m$ and any $x \in C$, $\Delta_x$ satisfies*

$$\Delta_x \leq 4(1 + \sqrt{c_1/c_2})r_{m-1} \tag{93}$$

*Proof.* For $m = 1$, recall that $r_m$ is the side length of a cube $C \in \mathcal{A}_m$, therefore, $\Delta_x \leq r_{m-1} \leq 4(1 + \sqrt{c_1/c_2})r_{m-1}$ holds directly from the Lipschitzness of $\mu$.

For $m > 1$, let $C_{m-1}^* \in \mathcal{A}_{m-1}$ be the cube containing $x^*$. From Lemma 28, this cube has not been eliminated under event $\mathcal{E}$. For any cube $C \in \mathcal{A}_m$ and $x \in C$, it is clear that $x$ is also in the parent of $C$, denoted $C_{par}$ ($x \in C \subset C_{par}$). Then for any $x \in C$, it holds that

$$\Delta_x = \mu(x^*) - \mu(x) \leq \left( \hat{\mu}_{m-1}(C_{m-1}^*) + r_{m-1} + \sqrt{c_1 \frac{\ln T}{n_{m-1}}} \right) \tag{94a}$$

$$+ \left( -\hat{\mu}_{m-1}(C_{par}) + r_{m-1} + \sqrt{c_1 \frac{\ln T}{n_{m-1}}} \right)$$

$$= (\hat{\mu}_{m-1}(C_{m-1}^*) - \hat{\mu}_{m-1}(C_{par})) + 2(\sqrt{c_1} + \sqrt{c_2})\sqrt{\frac{\ln T}{n_{m-1}}} \tag{94b}$$

$$\leq (\hat{\mu}_{m-1}^{\max} - \hat{\mu}_{m-1}(C_{par})) + 2(\sqrt{c_1} + \sqrt{c_2})\sqrt{\frac{\ln T}{n_{m-1}}} \tag{94c}$$

$$\leq 2(\sqrt{c_1} + \sqrt{c_2})\sqrt{\frac{\ln T}{n_{m-1}}} + 2(\sqrt{c_1} + \sqrt{c_2})\sqrt{\frac{\ln T}{n_{m-1}}} \tag{94d}$$

$$= 4(\sqrt{c_1} + \sqrt{c_2})\sqrt{\frac{\ln T}{n_{m-1}}} \tag{94e}$$

$$= 4(1 + \sqrt{c_1/c_2})r_{m-1} \tag{94f}$$

where we have applied Lemma 27 similarly as in Lemma 28 and also used the definition of $r_{m-1}$. The last two inequalities use the fact that $\hat{\mu}_{m-1}(C_{m-1}^*) \leq \hat{\mu}_{m-1}^{\max}$ and $C_{par}$ was not eliminated. $\quad\square$

**Theorem 2** (BLiN Regret Rate). *With probability exceeding $1 - 2T^{-2(c_1/c^2 - 1)}$, the $T$-step total regret $R(T)$ of BLiN with Doubling Edge-length Sequence (D-BLiN) Feng et al. (2022) satisfies*

$$R(T) \leq 8(1 + \sqrt{c_1/c_2})(2c_2 + 1)\ln(T)^{\frac{1}{d_z + 2}} T^{\frac{d_z + 1}{d_z + 2}} \tag{95}$$

*where $d_z$ is the zooming dimension of the problem instance. In addition, D-BLiN only needs no more than $B^* = \frac{\log 2(T) - \log 2(\ln(T))}{d_z + 2} + 2$ rounds of communications to achieve this regret rate.*

*Proof.* Since $r_m = \frac{r_{m-1}}{2} \implies r_{m-1} = 2r_m$ for the Doubling Edge-length Sequence, Lemma 29 implies that every cube $C \in A_m$ is a subset of $S(8(1 + \sqrt{c_1/c_2})r_m)$. Thus from the definition of zooming dimension (Corollary 5 with appropriate condition), we have

$$|\mathcal{A}_m| \leq N_{r_m} \leq C_z r_m^{-d_z}. \tag{96}$$

Fix any positive number $B$. Also by Lemma 29, we know that any arm played after batch $B$ incurs a regret bounded by $8(1 + \sqrt{c_1/c_2})r_B$, since the cubes played after batch $B$ have edge length no larger than $r_B$. Then the total regret that occurs after batch $B$ is bounded by $8(1 + \sqrt{c_1/c_2})r_B T$ (where $T$ is an upper bound on the number of arms).

Thus the regret can be bounded as

$$R(T) \leq \sum_{m=1}^{B} \sum_{C \in \mathcal{A}_m} \sum_{i=1}^{n_m} \Delta_{x_{C,i}} + 8(1 + \sqrt{c_1/c_2})r_B T \tag{97}$$

where the first term bounds the regret in the first $B$ batches of D-BLiN, and the second term bounds the regret after the first $B$ batches. If the algorithm stops at batch $\tilde{B} < B$, we define $\mathcal{A}_m = $ for any $\tilde{B} < m \leq B$ and inequality equation (97) still holds.

By Lemma 29, we have $\Delta_{x_{C,i}} \leq 8(1 + \sqrt{c_1/c_2})r_m$ for all $C \in \mathcal{A}_m$. We can thus bound equation (97) by

$$R(T) \leq \sum_{m=1}^{B} |\mathcal{A}_m| \cdot n_m \cdot 8(1 + \sqrt{c_1/c_2})r_m + 8(1 + \sqrt{c_1/c_2})r_B T \tag{98a}$$

$$\leq \sum_{m=1}^{B} N_{r_m} \cdot n_m \cdot 8(1 + \sqrt{c_1/c_2})r_m + 8(1 + \sqrt{c_1/c_2})r_B T \tag{98b}$$

$$= \sum_{m=1}^{B} N_{r_m} \cdot c_2 \frac{\ln T}{r_m^2} \cdot 8(1 + \sqrt{c_1/c_2})r_m + 8(1 + \sqrt{c_1/c_2})r_B T \tag{98c}$$

$$= \sum_{m=1}^{B} N_{r_m} \cdot \frac{\ln T}{r_m} \cdot 8c_2(1 + \sqrt{c_1/c_2}) + 8(1 + \sqrt{c_1/c_2})r_B T \tag{98d}$$

where equation (98b) uses equation (96), and equation (98c) uses equality $n_m = c_2 \frac{\ln T}{r_m^2}$. Since $r_m = 2^{-m+1}$ and $N_{r_m} \leq C_z r_m^{-d_z} \leq C_z 2^{(m-1)d_z}$, we have

$$R(T) \leq \sum_{m=1}^{B} C_z 2^{(m-1)d_z} \cdot \frac{\ln T}{2^{-m+1}} \cdot 8c_2(1 + \sqrt{c_1/c_2}) + 8(1 + \sqrt{c_1/c_2})2^{-B+1}T \tag{99a}$$

$$= 8(1 + \sqrt{c_1/c_2})\left[ c_2 C_z \ln T \sum_{m=1}^{B} 2^{(m-1)(d_z+1)} + 2^{-B+1}T \right]. \tag{99b}$$

Continuing we find

$$R(T) \le 8(1 + \sqrt{c_1/c_2}) \Big[ c_2 C_z \ln T \sum_{m=1}^{B} 2^{(m-1)(d_z+1)} + 2^{-B+1} T \Big] \tag{100a}$$

$$= 8(1 + \sqrt{c_1/c_2}) \Big[ c_2 C_z \ln T \sum_{m=1}^{B} \big( 2^{d_z+1} \big)^{m-1} + 2^{-B+1} T \Big] \tag{100b}$$

$$= 8(1 + \sqrt{c_1/c_2}) \Big[ c_2 C_z \ln T \sum_{m=0}^{B-1} \big( 2^{d_z+1} \big)^{m} + 2^{-B+1} T \Big] \tag{100c}$$

$$= 8(1 + \sqrt{c_1/c_2}) \Big[ c_2 C_z \ln T \Big( \frac{2^{B(d_z+1)} - 1}{2^{d_z+1} - 1} \Big) + 2^{-B+1} T \Big] \quad \text{via geometric series} \tag{100d}$$

$$\le 8(1 + \sqrt{c_1/c_2}) \Big[ c_2 C_z \ln T \Big( \frac{2^{B(d_z+1)}}{2^{d_z+1} - 1} \Big) + 2^{-B+1} T \Big] \tag{100e}$$

$$\le 8(1 + \sqrt{c_1/c_2}) \Big[ c_2 C_z \ln T \Big( 2 \cdot \frac{2^{B(d_z+1)}}{2^{d_z+1}} \Big) + 2^{-B+1} T \Big] \tag{100f}$$

$$= 8(1 + \sqrt{c_1/c_2}) \Big[ 2 c_2 C_z 2^{(B-1)(d_z+1)} \ln T + 2^{-(B-1)} T \Big]. \tag{100g}$$

This inequality holds for any positive $B$. By choosing $B^* = 1 + \frac{\log_2(\frac{T}{\ln T})}{d_z+2}$, we have

$$R(T) \le 8(1 + \sqrt{c_1/c_2}) \Big[ 2 c_2 C_z \Big( \frac{T}{\ln T} \Big)^{\frac{(d_z+1)}{(d_z+2)}} \ln T + \Big( \frac{\ln T}{T} \Big)^{\frac{1}{(d_z+2)}} T \Big] \tag{101a}$$

$$= 8(1 + \sqrt{c_1/c_2}) \Big[ 2 c_2 C_z T^{\frac{(d_z+1)}{(d_z+2)}} \ln T^{1 - \frac{(d_z+1)}{(d_z+2)}} + T^{1 - \frac{1}{(d_z+2)}} \ln T^{\frac{1}{(d_z+2)}} \Big] \tag{101b}$$

$$= 8(1 + \sqrt{c_1/c_2}) \Big[ 2 c_2 C_z T^{\frac{(d_z+1)}{(d_z+2)}} \ln T^{\frac{1}{(d_z+2)}} + T^{\frac{(d_z+1)}{(d_z+2)}} \ln T^{\frac{1}{(d_z+2)}} \Big] \tag{101c}$$

$$= 8(1 + \sqrt{c_1/c_2})(2 c_2 C_z + 1) T^{\frac{(d_z+1)}{(d_z+2)}} \ln T^{\frac{1}{(d_z+2)}}. \tag{101d}$$

$\square$

**Corollary 6** (BLiN Regret Rate Refined). *Setting $c_1 = 2c^2$ and $c_2 = 2\Big( \frac{c}{4C_z} \Big)^{2/3}$ simplifies Theorem 2 such that*

$$R(T) \le 8(1 + (4c^2 C_z)^{1/3})^2 T^{\frac{(d_z+1)}{(d_z+2)}} \ln T^{\frac{1}{(d_z+2)}}. \tag{102}$$

*with probability $1 - 2T^{-2}$.*

*Proof.* If we set $c_1 = 2c^2$ and $c_2 = 2\Big( \frac{c}{4C_z} \Big)^{2/3}$, then $\sqrt{c_1/c_2} = c\Big( \frac{4C_z}{c} \Big)^{1/3} = \Big( 4c^2 C_z \Big)^{1/3} = 2c_2 C_z$. $\square$

**Lemma 30.** *The zooming dimension and zooming constant under the $\ell(x, y) = ||x - y||_\infty$ norm are*

$$d_z = \frac{1}{2} n \bar{m} \tag{103}$$

$$C_z = |\mathcal{X}^*|^{-1} \Big( \frac{4}{r_\eta^2 \sigma_{-\infty}} \Big)^{n\bar{m}} \tag{104}$$

*where $\sigma_{-\infty} = ||\mathrm{Hess}(-f(x))^{-1}||_\infty$ is an upper bound on the infinity norm of the inverse Hessian matrix of the function at every equilibrium.*

*Proof.* Recall from Theorem 1 that $d_z = d\big( \frac{\alpha_{hi} - \alpha_{lo}}{\alpha_{lo} \alpha_{hi}} \big)$ with constant $C_z = 16^{-d'} C$ where $C$ is defined below. In addition, BLiN assumes $\ell(x, y) = ||x - y||_\infty$. Matching to Assumption 1, we see that $\sigma_+ = \alpha_{lo} = 1$. Under the infinity norm, the volume constant $S_d = 2^d$.

We will define the other constants with respect to properties of the Hessian of $f(x)$ about each equilibrium, specifically the infinity norm of the inverse Hessian so that $\sigma_- = \sigma_{-\infty}$. This means

we will bound the function locally with a quadratic, i.e., $\alpha_{hi} = 2$. Lastly, recall from Corollary 5 that $\psi = \frac{1}{32}$ and the dimension of our search space (the product space of player mixed strategies) is $d = n(\bar{m} - 1) \le n\bar{m}$ for simplicity.

Plugging this information into Theorem 1, we find

$$d_z = d' = \frac{1}{2}n\bar{m} \tag{105}$$

and

$$C = \max\left\{1, |\mathcal{X}^*|^{-1}S_d^{-1}\left(r_\eta^{\frac{\alpha_{hi}}{\alpha_{lo}}}\sigma_-^{-\left(\frac{\alpha_{hi}-\alpha_{lo}}{\alpha_{lo}\alpha_{hi}}\right)}\right)^{-d}\right\}\left(\frac{\sigma_+}{\psi\sigma_-^{\alpha_{lo}/\alpha_{hi}}}\right)^{d/\alpha_{lo}} \tag{106a}$$

$$= \max\left\{1, |\mathcal{X}^*|^{-1}2^{-d}\left(r_\eta^2\sigma_{-\infty}^{\frac{1}{2}}\right)^{-d}\right\}\left(\frac{32}{\sigma_{-\infty}^{1/2}}\right)^d \tag{106b}$$

$$= \max\left\{1, |\mathcal{X}^*|^{-1}\left(2r_\eta^2\sigma_{-\infty}^{\frac{1}{2}}\right)^{-d}\right\}\left(\frac{32}{\sigma_{-\infty}^{1/2}}\right)^d \tag{106c}$$

$$\stackrel{hard}{=} |\mathcal{X}^*|^{-1}\left(2r_\eta^2\sigma_{-\infty}^{\frac{1}{2}}\right)^{-d}\left(\frac{32}{\sigma_{-\infty}^{1/2}}\right)^d \tag{106d}$$

$$\stackrel{hard}{=} |\mathcal{X}^*|^{-1}\left(\frac{16}{r_\eta^2\sigma_{-\infty}}\right)^{n\bar{m}} \tag{106e}$$

where $hard$ indicates we are assuming $r_\eta$ and $\sigma_{-\infty}$ are small enough to dominate the other operand of the $\max$.

Finally, converting the near optimality constant to a zooming constant, we find

$$C_z = 16^{-d_z}|\mathcal{X}^*|^{-1}\left(\frac{16}{r_\eta^2\sigma_{-\infty}}\right)^{n\bar{m}} \tag{107a}$$

$$= |\mathcal{X}^*|^{-1}\left(\frac{4}{r_\eta^2\sigma_{-\infty}}\right)^{n\bar{m}}. \tag{107b}$$

$\square$

## B.4  BOUNDED DIAMETERS AND WELL-SHAPED CELLS

We assume the feasible set is a unit-hypercube of dimensionality $d$ where cells are evenly split along the longest edge to give $b$ new partitions and $x_{h,i}$ represents the center of each cell.

There exists a decreasing sequence $w(h) > 0$, such that for any depth $h \ge 0$ and for any cell $\mathcal{X}_{h,i}$ of depth $h$, we have $\sup_{x \in \mathcal{X}_{h,i}} \ell(x_{h,i}, x) \le w(h)$. Moreover, there exists $\nu > 0$ such that for any depth $h \ge 0$, any cell $\mathcal{X}_{h,i}$ contains an $\ell$-ball of radius $\nu w(h)$ centered at $x_{h,i}$.

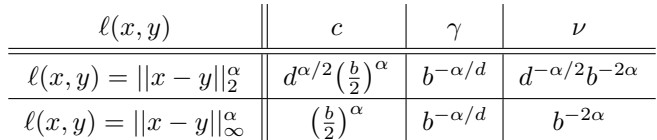

| $\ell(x,y)$ | $c$ | $\gamma$ | $\nu$ |
|---|---|---|---|
| $\ell(x,y) = \|x - y\|_2^\alpha$ | $d^{\alpha/2}\left(\frac{b}{2}\right)^\alpha$ | $b^{-\alpha/d}$ | $d^{-\alpha/2}b^{-2\alpha}$ |
| $\ell(x,y) = \|x - y\|_\infty^\alpha$ | $\left(\frac{b}{2}\right)^\alpha$ | $b^{-\alpha/d}$ | $b^{-2\alpha}$ |

Table 3: Bounding Constants: $\sup_{x \in \mathcal{X}_{h,i}} \ell(x_{h,i}, x) \le w(h) = c\gamma^h$.

### B.4.1  $L_2$-NORM

**Lemma 31** ($L_2$-Norm Bounding Ball). *Let $\ell(x,y) = \|x - y\|_2^\alpha$. Then $\sup_{x \in \mathcal{X}_{h,i}} \ell(x_{h,i}, x) \le w_2(h) = c\gamma^h$ where $c = \left(\frac{db^2}{4}\right)^{\alpha/2}$ and $\gamma = b^{-\alpha/d}$.*

*Proof.*

$$w(0) = \left[\sum_{i=1}^{d} (1/2)^2\right]^{\alpha/2} = \left(\frac{d}{4}\right)^{\alpha/2} \tag{108a}$$

$$w(1) = \left[(1/b \cdot 1/2)^2 + \sum_{i=2}^{d} (1/2)^2\right]^{\alpha/2} = \left[(1/b^2)(1/4) + (d-1)(1/4)\right]^{\alpha/2} \tag{108b}$$

$$= \left(\frac{d-1+1/b^2}{4}\right)^{\alpha/2} \tag{108c}$$

$$w(d) = \left[\sum_{i=1}^{d} (1/b \cdot 1/2)^2\right]^{\alpha/2} = \left(\frac{d}{4 \cdot b^2}\right)^{\alpha/2} \tag{108d}$$

$$w(h) = \left[r(1/b)^{2(q+1)}(1/2)^2 + \sum_{i=r}^{d}(1/b)^{2q}(1/2)^2\right]^{\alpha/2} \tag{108e}$$

$$= \left[(1/b)^{2q}(1/2)^2\left(r(1/b)^2 + (d-r)\right)\right]^{\alpha/2} \tag{108f}$$

$$= \left[(1/b^2)^q(1/4)\left(d - r(1 - \frac{1}{b^2})\right)\right]^{\alpha/2} \tag{108g}$$

$$\leq \left[(1/b^2)^q(1/4)d\right]^{\alpha/2} \tag{108h}$$

$$\leq \left[(1/b^2)^{h/d-1}(1/4)d\right]^{\alpha/2} \tag{108i}$$

$$= \left[(1/b^2)^{h/d}(b^2/4)d\right]^{\alpha/2} \tag{108j}$$

$$= \left(\frac{db^2}{4}\right)^{\alpha/2}(1/b)^{\frac{\alpha}{d}h} \tag{108k}$$

$$= c\gamma^h \tag{108l}$$

where $q, r = divmod(h, d) \implies q \geq h/d - 1$, $c = \left(\frac{db^2}{4}\right)^{\alpha/2}$, and $\gamma = (1/b)^{\alpha/d} = b^{-\alpha/d}$. $\qquad\square$

**Lemma 32** ($L_2$-Norm Inner Ball). *Let* $\ell(x, y) = ||x - y||_2^\alpha$. *Any cell* $\mathcal{X}_{h,i}$ *contains an* $\ell$-*ball of radius* $\nu w_2(h)$ *where* $\nu = (db^4)^{-\alpha/2}$.

*Proof.* Any cell $\mathcal{X}_{h,i}$ contains an $\ell$-ball of radius equal to its shortest axis:

$$r_{\min} = \left[(1/4)(1/b^2)^{\lceil h/d \rceil}\right]^{\alpha/2} \tag{109a}$$

$$\geq \left[(1/4)(1/b^2)^{h/d+1}\right]^{\alpha/2} \tag{109b}$$

$$= \left(\frac{1}{b^2 \cdot 4}\right)^{\alpha/2}(1/b)^{\frac{\alpha}{d}h} \tag{109c}$$

$$= w(h) \cdot \left(\frac{1}{db^4}\right)^{\alpha/2}. \tag{109d}$$

$$\square$$

### B.4.2 $L_\infty$-NORM

**Lemma 33** ($L_\infty$-Norm Bounding Ball). *Let* $\ell(x, y) = ||x - y||_\infty^\alpha$. *Then* $\sup_{x \in \mathcal{X}_{h,i}} \ell(x_{h,i}, x) \leq w_\infty(h) = c\gamma^h$ *where* $c = \left(\frac{b}{2}\right)^\alpha$ *and* $\gamma = b^{-\alpha/d}$.

*Proof.* Any cell $\mathcal{X}_{h,i}$ is contained by an $\ell$-ball of radius equal to its longest axis:

$$r_{\max} = \left[(1/4)(1/b^2)^{\lfloor h/d \rfloor}\right]^{\alpha/2} \tag{110a}$$

$$\leq \left[(1/4)(1/b^2)^{h/d-1}\right]^{\alpha/2} \tag{110b}$$

$$= \left(\frac{b^2}{4}\right)^{\alpha/2}(1/b)^{\frac{\alpha}{d}h} \tag{110c}$$

$$= c\gamma^h \tag{110d}$$

where $c = \left(\frac{b^2}{4}\right)^{\alpha/2}$, and $\gamma = (1/b)^{\alpha/d} = b^{-\alpha/d}$. □

**Lemma 34** ($L_\infty$-Norm Inner Ball). *Let $\ell(x,y) = ||x - y||_\infty^\alpha$. Any cell $\mathcal{X}_{h,i}$ contains an $\ell$-ball of radius $\nu w_\infty(h)$ where $\nu = b^{-2\alpha}$.*

*Proof.* Any cell $\mathcal{X}_{h,i}$ contains an $\ell$-ball of radius equal to its shortest axis:

$$r_{\min} = \left[(1/4)(1/b^2)^{\lceil h/d \rceil}\right]^{\alpha/2} \tag{111a}$$

$$\geq \left[(1/4)(1/b^2)^{h/d+1}\right]^{\alpha/2} \tag{111b}$$

$$= \left(\frac{1}{b^2 \cdot 4}\right)^{\alpha/2}(1/b)^{\frac{\alpha}{d}h} \tag{111c}$$

$$= w(h) \cdot \left(\frac{1}{b^4}\right)^{\alpha/2}. \tag{111d}$$

□

### B.5 STOCHASTIC SIMULTANEOUS OPTIMISTIC OPTIMIZATION

StoSOO is flexible in its choice of metric and partitioning structure. In StoSOO, we may choose $\ell(x,y) = ||x - y||_2^2$.

**Lemma 35.** *The volume of a $d$-sphere with radius $r$ and $d$ even is given by $S_d r^d$ where $S_d^{-1} \leq \sqrt{2\pi d}\left(\frac{d}{2\pi e}\right)^{d/2}$.*

*Proof.* First, we recall Stirling's bounds on the factorial: $\sqrt{2\pi n}(\frac{n}{e})^n e^{\frac{1}{12n+1}} < n! < \sqrt{2\pi n}(\frac{n}{e})^n e^{\frac{1}{12n}}$. This will be useful for bounding the Gamma function: $\Gamma(d) = (d-1)!$ for even $d$.

Given $d$ is even, we start with the exact formula for $S_d$:

$$S_d^{-1} = \frac{\Gamma(d/2+1)}{\pi^{d/2}} \tag{112a}$$

$$= \frac{(d/2)!}{\pi^{d/2}} \tag{112b}$$

$$< \frac{\sqrt{2\pi(d/2)}(\frac{d/2}{e})^{d/2}e^{\frac{1}{12(d/2)}}}{\pi^{d/2}} \tag{112c}$$

$$= \frac{\pi^{1/2}d^{1/2}(\frac{d}{2e})^{d/2}e^{\frac{1}{6d}}}{\pi^{d/2}} \tag{112d}$$

$$= \frac{\pi^{1/2}d^{(d+1)/2}e^{\frac{1}{6d}}}{(2\pi e)^{d/2}} \tag{112e}$$

$$\leq \sqrt{2\pi d}\left(\frac{d}{2\pi e}\right)^{d/2}. \tag{112f}$$

□

**Lemma 36.** *The near optimality dimension and constant under the $\ell(x,y) = ||x - y||_2^2$ norm are*

$$d' = 0 \tag{113}$$

$$C = |\mathcal{X}^*|^{-1}\sqrt{2\pi n\bar{m}}\left(\frac{n\bar{m}}{5\nu r_\eta^2\sigma_{-2}}\right)^{n\bar{m}/2} \tag{114}$$

*where $\sigma_{-2}$ is a lower bound on the singular value of the Hessian matrix of the function at every equilibrium and $\nu$ is defined in Table 3 for the corresponding $\ell(x,y)$-ball.*

*Proof.* Recall from Theorem 1 that $d' = d(\frac{\alpha_{hi}-\alpha_{lo}}{\alpha_{lo}\alpha_{hi}})$ with constant $C$ defined below. In addition, we will analyze StoSOO with the choice $\ell(x,y) = ||x - y||_2^2$. Matching to Assumption 1, we see

that $\sigma_+ = 1$ and $\alpha_{lo} = 2$. Under the 2-norm, the volume constant $S_d^{-1} \leq \sqrt{2\pi d}\left(\frac{d}{2\pi e}\right)^{d/2}$ (see Lemma 35).

We will bound the function locally with a quadratic, i.e., $\alpha_{hi} = 2$. Lastly, from (Valko et al., 2013, Corollary 1), $\psi = \frac{\nu}{3}$ with $\nu = \frac{1}{db^2}$ defined in Table 3 and the dimension of our search space (the product space of player mixed strategies) is $d = n(\bar{m} - 1) \leq n\bar{m}$ for simplicity.

Plugging this information into Theorem 1, we find $d' = 0$ and

$$C = \max\left\{1, |\mathcal{X}^*|^{-1} S_d^{-1}\left(r_\eta^{\frac{\alpha_{hi}}{\alpha_{lo}}} \sigma_-^{\left(\frac{\alpha_{hi}-\alpha_{lo}}{\alpha_{lo}\alpha_{hi}}\right)}\right)^{-d}\right\}\left(\frac{\sigma_+}{\psi\sigma_-^{\alpha_{lo}/\alpha_{hi}}}\right)^{d/\alpha_{lo}} \tag{115a}$$

$$= \max\left\{1, |\mathcal{X}^*|^{-1} S_d^{-1} r_\eta^{-d}\right\}\left(\frac{1}{\psi\sigma_{-2}}\right)^{d/2} \tag{115b}$$

$$\overset{hard}{=} |\mathcal{X}^*|^{-1} S_d^{-1} r_\eta^{-d}\left(\frac{3}{\nu\sigma_{-2}}\right)^{d/2} \tag{115c}$$

$$\overset{hard}{\leq} |\mathcal{X}^*|^{-1} \sqrt{2\pi d}\left(\frac{d}{2\pi e}\right)^{d/2}\left(\frac{3}{\nu r_\eta^2 \sigma_{-2}}\right)^{d/2} \tag{115d}$$

$$\overset{hard}{\leq} |\mathcal{X}^*|^{-1} \sqrt{2\pi d}\left(\frac{b^2 d^2}{5 r_\eta^2 \sigma_{-2}}\right)^{d/2} \tag{115e}$$

$$\overset{hard}{=} |\mathcal{X}^*|^{-1} \sqrt{2\pi n\bar{m}}\left(\frac{b^2 n^2 \bar{m}^2}{5 r_\eta^2 \sigma_{-2}}\right)^{n\bar{m}/2} \tag{115f}$$

where $hard$ indicates we are assuming $r_\eta$, the radius of the ball under which the local polynomial bounds are accurate, is small enough to dominate the other operand of the $\max$. $\square$

**Theorem 3** (StoSOO Regret Rate). *Corollary 1 of Valko et al. (2013) implies that with probability $1 - \delta$, the regret, $R_t$, of StoSOO after $t$ pulls is upper bounded as*

$$(2 + b^{2/d})\sqrt{\frac{\log_b(tk/\delta)}{2\log_b(e)k}} + \frac{1}{4}db^{2(1+2/d)}b^{-\frac{1}{dC}}\sqrt{t/k} \tag{116}$$

*where $d = n(\bar{m}-1)$, $b$ is the branching factor for partitioning cells, $C$ is the near-optimality constant, and $k$ is the maximum number of evaluations per node.*

*Proof.* Plugging the constants $c$, $\gamma$, and $\nu$ defined in Table 3 for the 2-norm into Corollary 1 of Valko et al. (2013), we find with probability $1 - \delta$:

$$R_t \leq (2 + 1/\gamma)\epsilon + c\gamma^{\frac{1}{2C}\sqrt{t/k}-2} \tag{117a}$$

$$= (2 + 1/\gamma)\sqrt{\frac{\log(tk/\delta)}{2k}} + c\gamma^{\frac{1}{2C}\sqrt{t/k}-2} \tag{117b}$$

$$= (2 + b^{2/d})\sqrt{\frac{\log(tk/\delta)}{2k}} + \frac{1}{4}db^2(b^{-2/d})^{\frac{1}{2C}\sqrt{t/k}-2} \tag{117c}$$

$$= (2 + b^{2/d})\sqrt{\frac{\log_b(tk/\delta)}{2\log_b(e)k}} + \frac{1}{4}db^{2(1+2/d)}(b^{-2/d})^{\frac{1}{2C}\sqrt{t/k}} \tag{117d}$$

$$= (2 + b^{2/d})\sqrt{\frac{\log_b(tk/\delta)}{2\log_b(e)k}} + \frac{1}{4}db^{2(1+2/d)}b^{-\frac{1}{dC}\sqrt{t/k}}. \tag{117e}$$

$\square$

**Proposition 2.** *Assume the same conditions as Theorem 3 and let $k = t\log_b(t)^{-\rho}$ where $\rho \geq 3$. Then the StoSOO (Algorithm 1 of Valko et al. (2013)) bound requires $t = b^{\left(\frac{dC}{2}\right)^{\frac{2}{\rho-2}}}$ pulls before exhibiting a $\tilde{\mathcal{O}}(T^{-1/2})$ decay in regret.*

*Proof.* We know from the given definition of $k$ that $\sqrt{t/k} = \log_b(t)^{\rho/2}$. We will analyze the second term in the regret formula of Corollary 3. Note that the second term, $b^{-\frac{1}{dC}\sqrt{t/k}} = (b^{\sqrt{t/k}})^{-\frac{1}{dC}} = (b^{\log_b(t)^{\rho/2}})^{-\frac{1}{dC}} = (b^{\log_b(t)\cdot\log_b(t)^{(\rho-2)/2}})^{-\frac{1}{dC}} = \left(t^{\log_b(t)^{(\rho-2)/2}}\right)^{-\frac{1}{dC}} = t^{-\frac{\log_b(t)^{(\rho-2)/2}}{dC}}$. To achieve a $t^{-1/2}$ convergence rate (or better), we need $t \geq b^{(\frac{dC}{2})^{\frac{2}{\rho-2}}}$. For convenience, we report the entire simplification of the bound below:

$$R_t \leq (2 + b^{2/d})\sqrt{\frac{\log_b(tk/\delta)}{2\log_b(e)k}} + \frac{1}{4}db^{2(1+2/d)}b^{-\frac{1}{dC}\sqrt{t/k}} \tag{118a}$$

$$= (2 + b^{2/d})\sqrt{\frac{\log_b(t^2/(\delta\log_b(t)^\rho))\log_b(t)^\rho}{2\log_b(e)t}} + \frac{1}{4}db^{2(1+2/d)}t^{-\frac{\log_b(t)^{(\rho-2)/2}}{dC}} \tag{118b}$$

$$\leq (2 + b^{2/d})\sqrt{\frac{\log_b(t^2/\delta)\log_b(t/\delta)^\rho}{2\log_b(e)t}} + \frac{1}{4}db^{2(1+2/d)}t^{-\frac{\log_b(t)^{(\rho-2)/2}}{dC}} \tag{118c}$$

$$= (2 + b^{2/d})\sqrt{\frac{2\log_b(t/\delta)^{\rho+1}}{2\log_b(e)t}} + \frac{1}{4}db^{2(1+2/d)}t^{-\frac{\log_b(t)^{(\rho-2)/2}}{dC}} \tag{118d}$$

$$= \frac{(2 + b^{2/d})}{\sqrt{\log_b(e)}}\frac{\log_b(t/\delta)^{(\rho+1)/2}}{\sqrt{t}} + \frac{1}{4}db^{2(1+2/d)}t^{-\frac{\log_b(t)^{(\rho-2)/2}}{dC}} \tag{118e}$$

$$= \frac{(2 + b^{2/d})}{\sqrt{\log_b(e)}}\frac{[\log_b(t) - \log_b(\delta)]^{(\rho+1)/2}}{\sqrt{t}} + \frac{1}{4}db^{2(1+2/d)}t^{-\frac{\log_b(t)^{(\rho-2)/2}}{dC}}. \tag{118f}$$

$\square$

## B.6  REGRET TO PAC BOUNDS

**Lemma 37.** *[Loss Regret to Exploitability Regret] Assume exploitability of a joint strategy $x$ is upper bounded by $f(\mathcal{L}^\tau(x))$ where $f$ is a concave function and $\mathcal{L}^\tau$ is a loss function. Let $x_t$ be a joint strategy randomly drawn from the set of predictions made by an online learning algorithm $\mathcal{A}$ over $T$ steps. Then the expected exploitability of $x_t$ is bounded by the average regret of $\mathcal{A}$:*

$$\mathbb{E}_t[\epsilon_t] \leq f(\frac{1}{T}R(T)). \tag{119}$$

*Proof.*

$$\mathbb{E}_t[\epsilon_t] = \mathbb{E}_t[f(\mathcal{L}(x_t))] \leq f(\mathbb{E}_t[\mathcal{L}(x_t)]) = f(\frac{1}{T}\sum_t \mathcal{L}(x_t)) = f(\frac{1}{T}R(T)) \tag{120}$$

where the inequality follows from Jensen's inequality. $\square$

**Theorem 4** (BLiN PAC Rate). *Assume $\eta_k = \eta = 2/\hat{L}$ as defined in Lemma 2, $\tau = \frac{1}{\ln(1/p)}$ so that all equilibria place at least $\frac{p}{m^*}$ mass on each strategy, and a previously pulled arm is returned uniformly at random (i.e., $t \sim U(T)$). Then for any $w > 0$,*

$$\epsilon_t \leq w\left[\tau\log\left(\prod_k m_k\right) + 2(1 + (4c^2C_z)^{1/3})\sqrt{2n\hat{L}}\left(\frac{\ln T}{T}\right)^{\frac{1}{2(d_z+2)}}\right] \tag{121}$$

*with probability $(1 - w^{-1})(1 - 2T^{-2})$ where $m^* = \max_k m_k$, and $c \leq \frac{n\bar{m}}{\hat{L}}\left(\frac{\ln(m^*)}{\ln(1/p)} + 2\right)^2$ is an upper bound on the range of $\hat{\mathcal{L}}^\tau$ (Corollary 1), $\hat{L} = \left(\frac{\ln(m^*)}{\ln(1/p)} + 2\right)\left(\frac{m^{*2}}{p\ln(1/p)} + n\bar{m}\right)$ (Corollary 2), the zooming dimension $d_z = \frac{1}{2}n\bar{m}$, and the zooming constant $C_z = |\mathcal{X}^*|^{-1}\left(\frac{4}{r_\eta^2\sigma_{-\infty}}\right)^{n\bar{m}}$ (Corollary 30).*

*Proof.* Assume $\eta_k = \eta = \frac{2}{\hat{L}}$ as defined in Lemma 2 so that $\mathcal{L}^\tau$ is 1-Lipschitz with respect to $|| \cdot ||_\infty$. Also assume a previously pulled arm is returned uniformly at random. Starting with Lemma 13 and applying Corollary 6, we find

$$\mathbb{E}[\epsilon_t] \leq \tau \log \Big( \prod_k m_k \Big) + \sqrt{\frac{2n}{\min_k \eta_k}} \sqrt{\frac{1}{T} \sum_t \mathcal{L}^\tau(\boldsymbol{x}_t)} \tag{122a}$$

$$\leq \frac{1}{\ln(1/p)} \log \Big( \prod_k m_k \Big) + \sqrt{n\hat{L}} \sqrt{8(1 + (4c^2)^{1/3})^2 T^{\frac{-1}{(d_z+2)}} \ln T^{\frac{1}{(d_z+2)}}} \tag{122b}$$

$$= \frac{1}{\ln(1/p)} \log \Big( \prod_k m_k \Big) + 2(1 + (4c^2 C_z)^{1/3}) \sqrt{2n\hat{L}} \Big( \frac{\ln T}{T} \Big)^{\frac{1}{2(d_z+2)}} \tag{122c}$$

with probability $1 - 2T^{-2}$ where $m^* = \max_k m_k$, and $c \leq \frac{n\bar{m}}{\hat{L}} \Big( \frac{\ln(m^*)}{\ln(1/p)} + 2 \Big)^2$ is an upper bound on the range of sampled values from $\hat{\mathcal{L}}^\tau$ (see Corollary 1).

Markov's inequality then allows us to bound the pointwise exploitability of any arm returned by the algorithm as

$$\epsilon_t \leq w \Big[ \frac{1}{\ln(1/p)} \log \Big( \prod_k m_k \Big) + 2(1 + (4c^2 C_z)^{1/3}) \sqrt{2n\hat{L}} \Big( \frac{\ln T}{T} \Big)^{\frac{1}{2(d_z+2)}} \Big] \tag{123}$$

with probability $(1 - w^{-1})(1 - 2T^{-2})$ for any $w > 0$. $\qquad\square$

**Theorem 5** (StoSOO PAC Rate). *Corollary 1 of Valko et al. (2013) implies that with probability $(1 - w^{-1})(1 - \delta)$ for any $w > 0$, a uniformly randomly drawn arm (i.e., $t \sim U([T])$) achieves*

$$\epsilon_t \leq w \Big[ \frac{1}{\ln(1/p)} \log \Big( \prod_k m_k \Big) + \sqrt{n\hat{L}} \sqrt{\xi_1 \sqrt{\frac{\log_b(Tk/\delta)}{2\log_b(e)k}} + \xi_2 b^{-\frac{1}{dC}\sqrt{T/k}}} \Big] \tag{124}$$

*where $d = n(\bar{m}-1)$, $\xi_1 = (2+2^{2/d})$, $\xi_2 = \frac{1}{4}db^{2(1+2/d)}$, $k = T\log_b(T)^{-3}$, $b$ is the branching factor for partitioning cells, and the near-optimality constant $C = |\mathcal{X}^*|^{-1}\sqrt{2\pi d}\Big( \frac{b^2 d^2}{5r_\eta^2 \sigma_{-2}} \Big)^{d/2}$ (Lemma 36).*

*Proof.* Assume $\eta_k = \eta = \frac{2}{\hat{L}}$ as defined in Lemma 2. Also assume a previously pulled arm is returned uniformly at random. Starting with Lemma 13 and applying Theorem 3 and Lemma 37, we find

$$\mathbb{E}[\epsilon_t] \leq \tau \log \Big( \prod_k m_k \Big) + \sqrt{\frac{2n}{\min_k \eta_k}} \sqrt{\frac{1}{T} \sum_t \mathcal{L}^\tau(\boldsymbol{x}_t)} \tag{125a}$$

$$\leq \frac{1}{\ln(1/p)} \log \Big( \prod_k m_k \Big) + \sqrt{n\hat{L}} \sqrt{\xi_1 \sqrt{\frac{\log_b(Tk/\delta)}{2\log_b(e)k}} + \xi_2 b^{-\frac{1}{dC}\sqrt{T/k}}} \tag{125b}$$

with probability $1 - \delta$ where $m^* = \max_k m_k$, $d = n(\bar{m}-1)$, $\xi_1 = (2+2^{2/d})$, $\xi_2 = \frac{1}{4}db^{2(1+2/d)}$, $k = T\log_b(T)^{-3}$, $b$ is the branching factor for partitioning cells, and the near-optimality constant $C = |\mathcal{X}^*|^{-1}\sqrt{2\pi d}\Big( \frac{b^2 d^2}{5r_\eta^2 \sigma_{-2}} \Big)^{d/2}$ (Lemma 36).

Markov's inequality then allows us to bound the pointwise exploitability of any arm returned by the algorithm as

$$\epsilon_t \leq w \Big[ \frac{1}{\ln(1/p)} \log \Big( \prod_k m_k \Big) + \sqrt{n\hat{L}} \sqrt{\xi_1 \sqrt{\frac{\log_b(Tk/\delta)}{2\log_b(e)k}} + \xi_2 b^{-\frac{1}{dC}\sqrt{T/k}}} \Big] \tag{126}$$

with probability $(1 - w^{-1})(1 - \delta)$ for any $w > 0$. $\qquad\square$

## B.7 COMPLEXITY OF POLYMATRIX GAMES

**Lemma 38.** *For a polymatrix game defined by the set of bimatrix games with payoff matrix $P_{kl}^k$ for every player $k$ and $l \neq k$, the rank of the matrix $M(x)$ defined in equation (11) can be equivalently studied by replacing all instances of $H_{kl}^k$ with $P_{kl}^k$.*

*Proof.* Consider the polymatrix game given by the set of matrices $P_{kl}^k$ for every $k$ and $l \neq k$. The polymatrix game can be equivalently written in normal form, albeit, less concisely. Note that the polymatrix approximation, $H_{kl}^k$, as we have defined it (see Section 2 Preliminaries) of this normal-form representation between players $k$ and $l$ with all other players' strategies marginalized out is related to the true underlying bimatrix game between them as follows:

$$H_{kl}^k[a_k, a_l] = \mathbb{E}_{x_{-kl}}[u_k(a_k, a_l, x_{-kl})] \ \forall a_k, a_l \tag{127a}$$

$$= a_k^\top \left( P_{kl}^k a_l + \sum_{j \notin \{k,l\}} P_{kj}^k x_j \right) \forall a_k, a_l \tag{127b}$$

$$= a_k^\top P_{kl}^k a_l + \underbrace{a_k^\top \left( \sum_{j \notin \{k,l\}} P_{kj}^k x_j \right)}_{p_k} \ \forall a_k, a_l \tag{127c}$$

where $P_{kl}^k$ is player $k$'s payoff matrix for the bimatrix game between players $k$ and $l$ in a polymatrix game and $p_k$ does not depend on player $l$'s strategy.

This implies that $H_{kl}^k$ is equal to $P_{kl}^k$ up to a constant offset of the rows, i.e.,

$$H_{kl}^k = P_{kl}^k + C_k \tag{128}$$

where $C_k$ is a matrix with constant rows.

Consider the matrix $M(x)$ which contains $H_{kl}^k$ blocks. Recall that the bottom rows of $M(x)$ contain rows of 1's matching each column of $H_{kl}^k$ blocks. Consider multiplying the $l$th row of 1's (which contains 0's on all columns not in the $l$th block) by $\sqrt{\eta_k}[I - \frac{1}{m_k}\mathbf{1}_k\mathbf{1}_k^\top]C_k$ and subtracting it from the block containing $\sqrt{\eta_k}[I - \frac{1}{m_k}\mathbf{1}_k\mathbf{1}_k^\top]H_{kl}^k$,

$$\sqrt{\eta_k}[I - \frac{1}{m_k}\mathbf{1}_k\mathbf{1}_k^\top][H_{kl}^k - C_k] = \sqrt{\eta_k}[I - \frac{1}{m_k}\mathbf{1}_k\mathbf{1}_k^\top]P_{kl}^k. \tag{129}$$

Note that $\sqrt{\eta_k}[I - \frac{1}{m_k}\mathbf{1}_k\mathbf{1}_k^\top]C_k$ still remains a matrix with constant rows (the preconditioner effectively subtracts a constant matrix from $C_k$). This multiplying and subtracting a row from another is an elementary operation on the matrix, meaning it does not change its row rank. Therefore, for a polymatrix game, we can reason about the positive definiteness of the Hessian at equilibria by examining the matrix $M(x)$ with all $H_{kl}^k$'s swapped for $P_{kl}^k$'s. □

Interestingly, at zero temperature (where QRE = Nash), $M$ is constant for a polymatrix game, so the rank of this matrix can be computed just once to extract information about all possible interior equilibria in the game. Furthermore, the Hessian is positive semi-definite over the entire joint strategy space, implying the loss function is convex (see Figure 5 (left) for empirical support). This indicates, by convex optimization theory, 1) all mixed Nash equilibria in polymatrix games form a convex set (i.e., they are connected) and 2) assuming mixed equilibria exist, they can be computed simply by stochastic gradient descent on $\mathcal{L}$. If $M$ is rank-$n\bar{m}$, then this interior equilibrium is unique.

**Complexity**   Approximation of Nash equilibria in polymatrix games is known to be PPAD-hard (Deligkas et al., 2022). In contrast, if we restrict our class of polymatrix games to those with at least one interior Nash equilibrium, our analysis proves we can find an approximate Nash equilibrium in deterministic, polynomial time (Corollary 7). This follows directly from the fact that $\mathcal{L}$ is convex, our decision set $\mathcal{X} = \prod_k \mathcal{X}_k$ is convex, and convex optimization theory admits polynomial time approximation algorithms (e.g., gradient descent). We consider the assumption of the existence of an interior Nash equilibrium to be relatively mild[3], so this positive complexity result is surprising.

---

[3]Marris et al. (2023) shows 2-player, 2-action polymatrix games with interior Nash equilibria make up a non-trivial $1/4$ of the space of possible $2 \times 2$ games.

Also, note that the Hessian of the loss at Nash equilibria is encoded entirely by the polymatrix approximation at the equilibrium. Therefore, approximating the Hessian of $\mathcal{L}$ about the equilibrium (which amounts to observing near-equilibrium behavior (Ling et al., 2018)) allows one to recover this polymatrix approximation (up to constant offsets of the columns which equilibria are invariant to (Marris et al., 2023)).

**Corollary 7** (Approximating Nash Equilibria of Polymatrix Games with Interior Equilibria). *Consider the class of polymatrix games with interior Nash equilibria. This class of games admits a fully polynomial time deterministic approximation scheme (FPTAS).*

*Proof.* Lemma 3 relates the approximation of Nash equilibria to the minimization of the loss function $\mathcal{L}(x)$. By Lemma 1, this loss function attains its minimum value of zero if and only if $x$ is a Nash equilibrium. For polymatrix games, the Hessian of this loss function is everywhere finite and positive definite (Lemma 16), therefore, this loss function is convex. The decision set for this minimization problem is the product space of simplices, therefore it is also convex. Given that we only consider polymatrix games with interior Nash equilibria, we know that our loss function attains a global minimum within this set. By convex optimization theory, this function can be approximately minimized in a polynomial number of steps by, for example, (projected) gradient descent (Boyd and Vandenberghe, 2004). Gradient descent requires computing the gradient of the loss function at each step. From Lemma 14, we see that computing the gradient (at zero temperature) simply requires reading the polymatrix description of the game (i.e., each bi-matrix game $H_{kl}^k$ between players), which is clearly polynomial in the size of the input (the polymatrix description). The remaining computational steps of gradient descent (e.g., projection onto simplices) are polynomial as well. In conclusion, gradient descent approximates a Nash equilibrium in polynomial number of steps (logarithmic if strongly-convex (Mairal, 2013)), each of which costs polynomial time, therefore the entire scheme is polynomial. $\square$

## C EXPERIMENTAL SETUP AND DETAILS

Here we provide further details on the experiments.

### C.1 GAMBIT

The seven methods from the `gambit` (McKelvey et al., 2016) library that we tested on the 3-player and 4-player Blotto games are listed below (with runtimes). Only `gambit-enumpoly` and `gambit-enumpure` are able to return any NE for 3-player Blotto within a 1 hour time limit (and only pure equilibria). And only `gambit-enumpure` returns any NE for the 4-player game.

- `gambit-enumpoly` [73 sec 3-player, timeout 4-player]
- `gambit-enumpure` [72 sec 3-player, 45 sec 4-player]
- `gambit-gnm`
- `gambit-ipa`
- `gambit-liap`
- `gambit-logit`
- `gambit-simpdiv`

### C.2 LOSS VISUALIZATION AND RANK TEST

Figure 5 and claims made in Section 5 analyze several classical matrix games. We report the payoff matrices in standard row-player / column-player payoff form below. All games are then shifted and scaled so payoffs lie in $[0, 1]$ (i.e., first by subtracting the minimum and then scaling by the max).

RPS:

$$\begin{bmatrix} 0/0 & -1/1 & 1/-1 \\ 1/-1 & 0/0 & -1/1 \\ -1/1 & 1/-1 & 0/0 \end{bmatrix}. \tag{130}$$

Chicken:

$$\begin{bmatrix} 0/0 & -1/1 \\ 1/-1 & -3/-3 \end{bmatrix}. \tag{131}$$

Matching Pennies:

$$\begin{bmatrix} 1/-1 & -1/1 \\ -1/1 & 1/-1 \end{bmatrix}. \tag{132}$$

Modified-Shapleys:

$$\begin{bmatrix} 1/-0.5 & 0/1 & 0.5/0 \\ 0.5/0 & 1/-0.5 & 0/1 \\ 0/1 & 0.5/0 & 1/-0.5 \end{bmatrix}. \tag{133}$$

Prisoner's Dilemma:

$$\begin{bmatrix} -1/-1 & -3/0 \\ 0/-3 & -2/-2 \end{bmatrix}. \tag{134}$$

### C.2.1 Loss on Familiar Games

We visualize our proposed loss $\mathcal{L}^\tau$ on two classic 2-player, general-sum games: Chicken, payoff matrix (131), and the Prisoner's Dilemma, payoff matrix (134). Each plot in Figure 5 visualizes the loss at various strategy profiles in probability-space; each strategy profile is represented by each player's probability of playing action 1 of 2 (top row / first column of payoff matrix). Temperature $\tau$ is varied across the plots. Figure 5 repeats this same visualization but in logit-space to better show the equilibria closest to the boundaries.

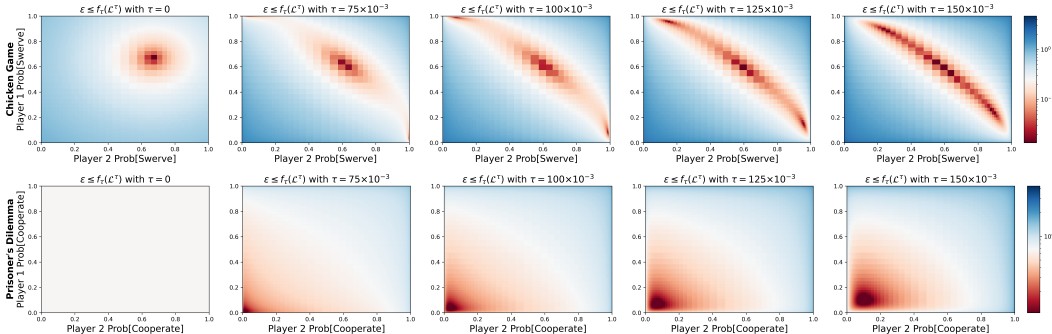

Figure 5: Upper Bound ($\epsilon \le f_\tau(\mathcal{L}^\tau)$) Heatmap Visualization. The first row examines the loss landscape for the classic anti-coordination game of Chicken (Nash equilibria: $(0,1), (1,0), (2/3, 1/3)$) while the second row examines the Prisoner's dilemma (Unique Nash equilibrium: $(0,0)$). For improved visibility, we subtract the offset $\tau \log(m^2)$ from $f_\tau(\mathcal{L}^\tau)$ per Lemma 13, which relates the exploitability at positive temperature to that at zero temperature. Temperature increases for each plot moving to the right. For high temperatures, interior (fully-mixed) strategies are incentivized while for lower temperatures, nearly pure strategies can achieve minimum exploitability. For zero temperature, pure strategy equilibria (e.g., defect-defect) are not captured by the loss as illustrated by the bottom-left Prisoner's Dilemma plot with a constant loss surface.

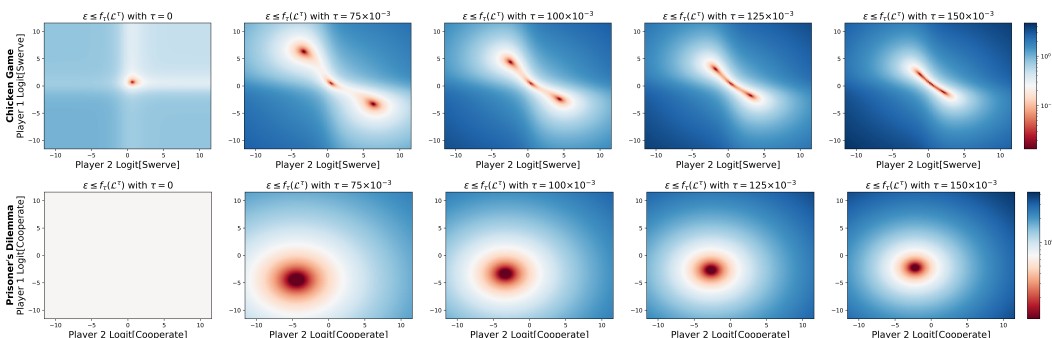

Figure 6: [Figure 5 Repeated in Logit-Space ($\ln(\frac{p}{1-p})$) Rather than Probability-Space ($p$)] Upper Bound ($\epsilon \le f_\tau(\mathcal{L}^\tau)$) Heatmap Visualization. The first row examines the loss landscape for the classic anti-coordination game of Chicken (Nash equilibria: $(0,1), (1,0), (2/3, 1/3)$) while the second row examines the Prisoner's dilemma (Unique Nash equilibrium: $(0,0)$). Temperature increases for each plot moving to the right. For improved visibility, we subtract the offset $\tau \log(m^2)$ from $f_\tau(\mathcal{L}^\tau)$ per Lemma 13, which relates the exploitability at positive temperature to that at zero temperature. For high temperatures, interior (fully-mixed) strategies are incentivized while for lower temperatures, nearly pure strategies can achieve minimum exploitability. For zero temperature, pure strategy equilibria (e.g., defect-defect) are not captured by the loss as illustrated by the bottom-left Prisoner's Dilemma plot with a constant loss surface.

## C.3   SADDLE POINT ANALYSIS

To generate Figure 2, we follow a procedure similar to the study of MNIST in (Dauphin et al., 2014) (Section 3 of Supp.). Their recommended procedure searches for critical points in two ways. The first repeats a randomized, iterative optimization process 20 times. They then sample one these 20 trials at random, select a random point along the descent trajectory, and search for a critical point (using Newton's method) nearby. They repeat this sampling process 100 times. The second approach randomly selects a feasible point in the decision set and searches for a critical point nearby (again using Newton's method). They also perform this 100 times.

Our protocol differs from theirs slightly in a few respects. One, we use SGD, rather than the saddle-free Newton algorithm to trace out an initial descent trajectory. Two, we do not add noise to strategies along the descent trajectory prior to looking for critical points. Thirdly, we minimize gradient norm rather than use Newton's method to look for critical points. Lastly, we use different experimental hyperparameters. We run SGD for 1000 iterations rather than 20 epochs and rerun SGD 100 times rather than 20. We sample 1000 points for each of the two approaches for finding critical points.

## C.4   SGD ON CLASSICAL GAMES

The games examined in Figure 3 were all taken from (Gemp et al., 2022). Each is available via open source implementations in OpenSpiel (Lanctot et al., 2019) or GAMUT (Nudelman et al., 2004).

We compare against several other baselines, replicating the experiments in (Gemp et al., 2022). RM indicates regret-matching and FTRL indicates follow-the-regularized-leader. These are, arguably, the two most popular scalable stochastic algorithms for approximating Nash equilibria. $^{y}\text{QRE}^{auto}$ is a stochastic algorithm developed in (Gemp et al., 2022).

For each of the experiments, we sweep over learning rates in log-space from $10^{-3}$ to $10^{2}$ in increments of 1. We also consider whether to run SGD with the projected-gradient and whether to constrain iterates to the simplex via Euclidean projection or entropic mirror descent (Beck and Teboulle, 2003). We then presented the results of the best performing hyperparameters. This was the same approach taken in (Gemp et al., 2022).

**Saddle Points in Blotto**   To confirm the existence of saddle points, we computed the Hessian of $\mathcal{L}(\boldsymbol{x_{10k}})$ for SGD ($s = \infty$), deflated the matrix by removing from its eigenvectors all directions orthogonal to the simplex, and then computed its top-$(n\bar{m} - n)$ eigenvalues. We do this because there always exists a $n$-dimensional nullspace of the Hessian at zero temperature that lies outside the tangent space of the simplex, and we only care about curvature within the tangent space. Specificaly, at an equilibrium $\boldsymbol{x}$, if we compute $z^{\top}Hess(\mathcal{L})z$ where $z$ is formed as a linear combination of the vectors $\{[x_1, 0, \ldots, 0]^{\top}, \ldots, [0, \ldots, x_n]^{\top}\}$, then each block $\tilde{B}_{kl}$ is identically zero at an equilibrium: $\tilde{B}_{kl}x_l = \sqrt{\eta_k}[I - \frac{1}{m_k}\mathbf{1}\mathbf{1}^{\top}]H_{kl}^k x_l = \sqrt{\eta_k}\Pi_{T\Delta}(\nabla_{x_k}^k) = 0$. By Lemma 16, this implies there is zero curvature of the loss in the direction $z$: $z^{\top}Hess(\mathcal{L})z = 0$.

## C.5   BLIN ON ARTIFICIAL GAME

To construct the 7-player, 2-action, symmetric, artifical game in Figure 4, we used the following coefficients (discovered by trial-and-error):

$$
\begin{bmatrix}
0.09906873 & 0 & 0.23116037 & 0 & 0.62743528 & 0 & 0.19813746 \\
0 & 0.33022909 & 0 & 0.03302291 & 0 & 0.62743528 & 0
\end{bmatrix}.
\tag{135}
$$

The first row indicates the payoffs received when player $i$ plays action 0 and the background population plays any of the possible joint actions (number of combinations with replacement). For example, the first column indicates the payoff when all background players play action 0. The second column indicates all background players play action 0 except for one which plays action 1, and so on. The last column indicates all background players play action 1. These $2n$ scalars uniquely define the payoffs of a symmetric game.

Given that this game only has two actions, we represent a mixed strategy by a single scalar $p \in [0, 1]$, i.e., the probability of the first action. Furthermore, this game is symmetric and we seek a symmetric equilibrium, so we can represent a full Nash equilibrium by this single scalar $p$. This reduces our search space from $7 \times 2 = 14$ variables to $1$ variable (and obviates any need for a map $s$ from the unit hypercube to the simplex—see Lemma 22).

