# OpenReview forum: "Approximating Nash Equilibria in Normal-Form Games via Stochastic Optimization"
_ICLR.cc/2024/Conference — ICLR 2024 oral_

### Official Review · Reviewer_Lp7D · 2023-10-20

**Soundness:** 4 excellent
**Presentation:** 4 excellent
**Contribution:** 4 excellent
**Rating:** 8
**Confidence:** 3

**Summary:**

This paper investigates the problem of solving a Nash equilibrium (NE) inside a normal-form general-sum game (NFG). Unlike previous works, the authors propose a new kind of loss that serves as an upper bound for $\epsilon$ (coefficient of $\epsilon$-Nash). More importantly, the authors show that an unbiased gradient of the loss can be obtained using the symmetrization technique. As a result, the problem of NE can be treated as a stochastic optimization problem, such that some famous algorithms like SGD and BLiN can exploited. Moreover, the authors also consider the case when the NE is not inside the simplex but on its boundary. And they show that this issue can be also handled by optimizing on a refined game with additional bonuses. Finally, empirical studies validate the effectiveness of the proposed method.

**Strengths:**

I have not read the papers about how to solve an NE efficiently before. However, this paper is quite clear in this presentation, such that I have now obtained a whole picture of this problem. The problem setup is clearly introduced. The solution is illuminated step by step, which is clear and intuitive. It is worth mentioning that the authors give several illustration figures (e.g., Figures 1 and 2) to help the readers understand the paper better, which is very good. The experiments are also sufficient and complete. Moreover, this work offers the community a new and principled way of solving NEs efficiently, and many more stochastic optimization algorithms can be leveraged in this field.

**Weaknesses:**

I do not see major weaknesses in this work, though I am not an expert in this field.

**Questions:**

I do not have any questions, as the presentation of this work is quite clear and intuitive, along with some illustrations to help readers understand it well.

---

> ### Author Response · Authors · 2023-11-16
> **Response**
>
> Thank you for your encouraging feedback! One of the goals of this paper is to open up the challenge of approximating Nash equilibria to researchers without a predominant game theory background (and in particular, to those with more of a machine learning / optimization background). We are glad to hear that you valued the step-by-step explanation and accompanying illustrations!

---

### Official Review · Reviewer_rSdL · 2023-10-30

**Soundness:** 3 good
**Presentation:** 3 good
**Contribution:** 3 good
**Rating:** 8
**Confidence:** 4

**Summary:**

The paper proposes a new loss function for approximating Nash equilibria in normal-form games. The key idea is that the loss function can be unbiasedly estimated through Monte Carlo sampling of the joint strategies. This allows the use of stochastic optimization, eliminating the need to read all the payoffs, which would be exponential with respect to the number of players.

The loss function upper-bounds the exploitability of an approximate equilibrium. The authors prove this both theoretically and empirically. This approach enables the scaling of equilibrium computation to larger games than previously thought feasible. Experiments are conducted on games with up to 286 actions.

Two algorithms are explored: vanilla SGD and a bandit-based method. The bandit algorithm comes with theoretical guarantees regarding exploitability.

**Strengths:**

1. The paper proposes a new loss function for approximating Nash equilibria in normal-form games that is amenable to unbiased Monte Carlo estimation. This allows for the use of powerful sample-based stochastic optimization techniques, eliminating the need to read all the payoffs.
2. The loss function upper-bounds the exploitability of an approximate equilibrium. The authors provide theoretical proofs and present empirical results.
3. The proposed approach enables the scaling of equilibrium-finding techniques to larger games than was previously possible. The paper presents experiments on games with up to 286 actions.
4. Two algorithms are analyzed: vanilla SGD and a bandit-based method. The bandit approach offers theoretical guarantees, such as a high probability bound on exploitability.

**Weaknesses:**

1. The loss function only captures fully mixed equilibria. The authors address this by considering quantal-response equilibrium. As a result, a zero loss can only serve as an approximation to a Nash equilibrium.
2. There is limited empirical evaluation on real-world games. Most experiments involve small synthetic games from previous research. It's possible that more complex games may expose certain limitations.
3. SGD encounters issues with saddle points in certain games, which is a common challenge in non-convex optimization.

**Questions:**

1. How well does the approach scale to larger real-world games with structures like symmetry? Are techniques such as grouping actions necessary?
2. Have the authors attempted more advanced optimization methods like Momentum SGD or Adam to address saddle points?
3. What heuristics could be employed to guide equilibrium selection when multiple equilibria exist?

---

> ### Author Response · Authors · 2023-11-16
> **Response**
>
> Thank you for taking the time to carefully review our paper and for your interesting questions.
>
> We agree with each of the weaknesses you describe and hope you are satisfied with how we identified these clearly in our text. We point these out below for convenience.
> - Lemma 14 is composed of two terms. The first term is a positive constant that remains even when our proposed loss is zero. Therefore, yes, as you say, this clearly indicates our loss can only be used to approximate Nash equilibria (hence the title of the paper).
> - The scope of this work is normal-form games, which is the natural starting point for studying more complex game settings and arguably the bedrock of most traditional economic theory. If by real-world games, you mean games like poker (extensive-form) or some video games (Markov games), then yes, we do not cover that in this work. We mention extensive-form games as one of our future goals in the conclusion.
> - Figure 2 presents an empirical analysis of the existence of saddle points as indicated by the colors of points in the scatter plots. We also verify in Appendix K.4 that the Hessian of our loss evaluated at the final iterate of SGD in Figure 3 in both Blotto experiments contains both positive and negative eigenvalues, i.e, the landscape is saddle-shaped.
>
> We now answer your questions:
> 1) The structure of symmetric games can be exploited in our approach. Recall that every symmetric NFG has a symmetric NE. Therefore, we can reduce our search space over joint profiles of size $nm$ to a search over a single player’s mixed strategy of size $m$. Moreover, if you inspect the gradient of our loss in equation (9) which contains a sum over all players $k$, we can simplify this as well to a sum over player $\ell$ and any single player $k \ne \ell$ (scaled by $n-1$). Due to symmetry and the fact that all players play the same background strategy in a symmetric NE, each of the summands is the same, so we only need to retain one. See Figure 4 where we explicitly take advantage of the symmetry of the game to both solve it and visualize it; we present all player strategies as a single 1-d variable ($p$) on the x-axis. If instead of player / payoff-symmetry, you meant action-symmetry, then yes, grouping actions would help to scale our approach when two actions are equivalent.
> 2) We believe this is a very interesting direction for future work. We have tried Adam actually on Blotto (10c, 3f, 4p) by reparameterizing the mixed strategy as a softmax over real-valued logits (vanilla Adam assumes unconstrained optimization). We found that Adam improves on SGD slightly ($\epsilon$ drops from $0.029$ to $0.020$), but we still see a saddle-shape in the landscape when we inspect the loss at Adam’s final step. This suggests Adam helped cope with the saddle point problem, but is not a complete solution. We explicitly decided against including these preliminary results so as to encourage follow-up work by non-convex / deep-learning optimization experts with respect to a variety of non-vanila SGD algorithms such as Adam, Adagrad, saddle-point  avoidant solvers, learning-to-learn approaches, a.o. We believe that the ICLR audience would be the perfect set of experts to take this on.
> 3) A simple heuristic would be to regularize our proposed loss with a second loss term encoding an additional criterion, e.g., maximizing welfare. You could anneal this term over learning to try to find the equilibrium with the highest welfare.

---

> > ### Comment · Reviewer_rSdL · 2023-11-20
> >
> > Thank you for your responses that answered my questions. I will maintain my score.

---

### Official Review · Reviewer_MRwi · 2023-10-30

**Soundness:** 4 excellent
**Presentation:** 4 excellent
**Contribution:** 4 excellent
**Rating:** 8
**Confidence:** 3

**Summary:**

The paper provides a loss function for computation of approximate Nash equilibria in general-sum multiplayer matrix games for which unbiased estimators can be constructed. This allows the exploitation of stochastic optimization algorithms for computation of Nash equilibria in general matrix games. They provide numerical evidence that this approach is competitive with existing approaches.

**Strengths:**

1. The paper is generally well-written.
2. The formulation of the loss function which allows for unbiased sampling is a solid novel technical contribution. As the authors note, this is a step towards enabling computation of Nash equilibria in large-scale settings (in spite of theoretical hardness results), which is quite important for operationalizing game theory in the real world.
3. The paper provides numerical evidence supporting that the loss function is "easy" to optimize in many benchmark games of interest, and that it is competitive with other SOTA approaches.

**Weaknesses:**

1. The empirical section (6.2) could use some additional explanation/discussion. The baselines that are compared against should be explicitly stated in the body text (and cited appropriately) instead of just stating that they are the baselines used in Gemp et al. 2022's simulations.
2. The flow of Section 6 generally could be improved (one suggestion is provided below).

**Questions:**

1. There is a typo in the definition of the projected-gradient (it should be $\Pi_{T\delta^{d-1}}(z) = z - \frac{1}{d} \mathbf{1}^\top z \mathbf{1}$)
2. In Section 6, it is confusing to introduce SGD first and then the bandit algorithm and then immediately have a subsection discussing the bandit algorithm, especially since it is stated that "in the next section, we find it performs well empirically in games previously examined by the literature." Perhaps it makes sense to include the SGD experiments before any discussion of the bandit approach.

---

> ### Author Response · Authors · 2023-11-16
> **Response**
>
> Thank you for your positive review and for your constructive feedback!
>
> We agree, the empirical section 6.2 is quite short and there are several details (in particular, descriptions of the baselines) in Appendix K that we will pull into the main body.
>
> We also appreciate your suggestion regarding flow. Would you agree with a structure such as this?
> - SGD
> - - Discussion
> - - Experiments
> - Bandits
> - - Discusision
> - - Theory
> - - Experiments
>
> And finally, great catch on the typo. Yes, we are missing the “ones vector”.

---

> > ### Comment · Reviewer_MRwi · 2023-11-20
> >
> > Thank you for your response. I think that the structure you have proposed for Section 6 makes sense. I will maintain my score.

---

### Author Response · Authors · 2023-11-16
**General Response**

Thank you to all the reviewers for taking the time to read our paper and provide helpful feedback. We are very encouraged by your words! We were quite pleased to hear that many of you feel this approach is both novel and should be intuitively clear to a non-game theory audience. We hoped precisely for this: that by transforming the fundamental problem of game theory (approximating Nash) into a stochastic optimization problem, we might open the game theory door to world class ML researchers in the ICLR community.

---

### Meta-Review · Area_Chair_QwGz · 2023-12-06

**Metareview:**

There is consensus---and I agree---that this paper proposes an interesting combination of techniques that deserves to be highlighted at the conference, for its somewhat broad appeal and possible future work.

**Justification For Why Not Higher Score:**

NA

**Justification For Why Not Lower Score:**

The paper has a broad appeal to people working in computational game theory. I think it deserves either a spotlight or oral presentation.

---

### Decision · Program_Chairs · 2024-01-16

Accept (oral)